Resource

# The proteomic landscape and temporal dynamics of human and mouse gastruloid development

Riddhiman K. Garge [1,2,3] ✉, Valerie Lynch[1], Rose Fields[1], Silvia Casadei[1,2], Sabrina Best[2], Jeremy Stone [2], Matthew Snyder [2], Connor Kubo [1,3], Arata Wakimoto[3,4,5], Zukai Liu [1,3,5], Chris D. McGann [1], Jay Shendure [1,2,3,5,6] ✉, Lea M. Starita [1,2] ✉, Nobuhiko Hamazaki [1,2,3,4,5] ✉ & Devin K. Schweppe [1,2,5] ✉

The embryo establishes a body plan and primes itself for organogenesis during gastrulation. As gastrulation is challenging to study in vivo, stem-cell-derived 'gastruloids' have emerged as powerful surrogates. Although transcriptomics and imaging have been applied extensively to such embryo models, the dynamics of their proteomes remains largely unknown. Here we apply quantitative proteomics to human and mouse gastruloids at four key stages. We leverage these data to map the expression dynamics of protein complexes, and to nominate cooperative proteins. With matched transcriptome data, we investigate global and stage-specific discordance between the transcriptome and proteome and leverage phosphosite dynamics to nominate kinase–substrate relationships. Finally, we apply co-regulation network analysis to identify genes linked to the Commander complex, the perturbation of which leads to morphological defects in gastruloids. Altogether, our work showcases the potential of applying proteomics to embryo models to advance our understanding of mammalian development in ways challenging through transcriptomics alone.

Gastrulation is a crucial process through which the implanted blastocyst transforms into a three-germ-layer structure, the gastrula[1]. Ethical and practical challenges in obtaining embryos limit our understanding of human gastrulation[2,3]. Conserved aspects of gastrulation can be studied in the mouse, but practical challenges (such as opacity and the cost of genetic manipulation) and notable species differences in morphology, regulators (for example, FGF8 and BMP4) and cell-type origins (for example, primordial germ cells) limit its utility in understanding human development[4].

Stem-cell-derived embryo models are powerful surrogates, and have proliferated in both usage and scope[5]. Gastruloids—one such model—are generated by aggregating hundreds of embryonic stem cells (ESCs) and inducing Wingless-Int (WNT) signalling, which triggers axial elongation and the emergence of all three germ layers[6–8]. With Matrigel, mouse gastruloids form morphological structures resembling their in vivo counterparts, with an elongated neural tube and flanking somites[8,9]. Recently, we demonstrated that an early addition of retinoic acid (RA) in developing human gastruloids yields structures and advanced cell types including a neural crest, neural progenitors, renal progenitors and myocytes ('RA-gastruloids')[10]. Gastruloids can be manipulated, characterized and grown in large numbers[7].

[1]Department of Genome Sciences, University of Washington, Seattle, WA, USA. [2]Brotman Baty Institute for Precision Medicine, Seattle, WA, USA. [3]Seattle Hub for Synthetic Biology, Seattle, WA, USA. [4]Departments of Obstetrics & Gynecology, University of Washington, Seattle, WA, USA. [5]Institute of Stem Cell and Regenerative Medicine, University of Washington, Seattle, WA, USA. [6]Howard Hughes Medical Institute, University of Washington, Seattle, WA, USA. ✉e-mail: rgarge@uw.edu; shendure@uw.edu; lstarita@uw.edu; hamazaki@uw.edu; dkschwep@uw.edu

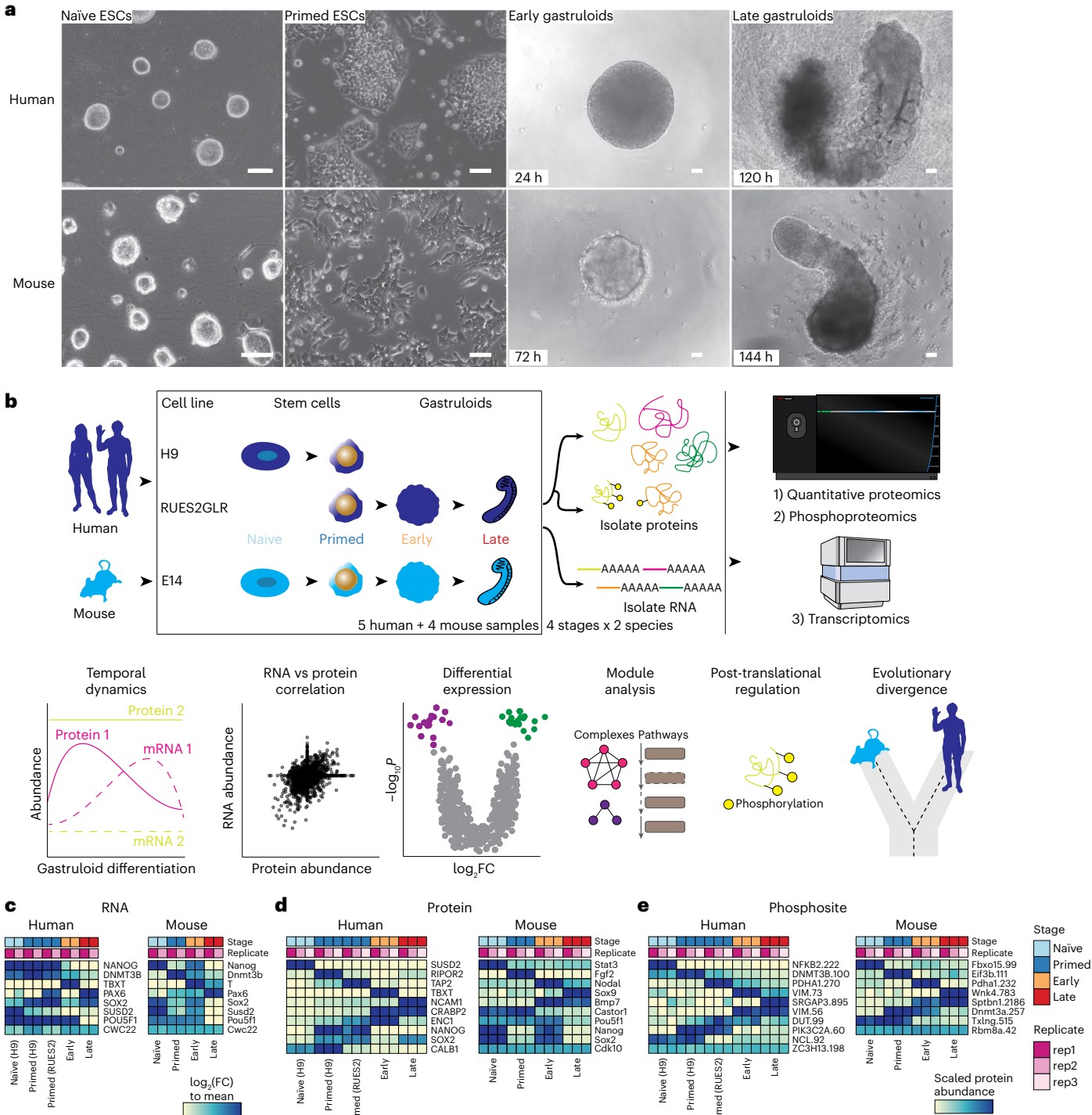

**Fig. 1 | Quantifying the dynamic proteome from ESCs to gastruloids. a**, Representative brightfield images of human RA-gastruloids and mouse gastruloids imaged over the course of their development. Scale bars, 10 μm. The experiments were independently reproduced five times with similar results. **b**, Multi-omics profiling workflow. We sampled two human cell lines (H9 and RUES2-GLR) and one mouse cell line (E14) at the indicated stages.

**c–e**, Representative heatmaps depicting the temporal dynamics of RNAs (**c**), proteins (**d**) or phosphosites (**e**) for selected developmental marker transcripts, proteins or PTMs, respectively, across replicates and stages for both human and mouse. The colour scale for RNAs indicates log₂(fold change) relative to the row mean. The colour scale for protein and phosphorylation data indicates scaled TMTpro reporter ion abundance. FC, fold change.

Several groups, including us, have applied single-cell RNA sequencing (scRNA-seq) to characterize transcriptome dynamics gastruloid development[11,12]. However, RNA is only the messenger. It is proteins that are the workhorses of the cell, and in differentiating gastruloids, proteins form the structures that make emerging germ layers and cell types morphologically and functionally unique. Protein abundances are difficult to estimate from transcriptomics alone[13–18], and studies report varying levels of discordance[19–23]. One study found that transcript abundance accounted for only ~40% of the variance in human protein levels[24]. Moreover, post-translational modifications (PTMs) vastly increase the proteome diversity to more than 10 million proteoforms[25], aspects of identity and function that are entirely

absent from a transcriptomic census. PTMs dynamically regulate the signalling pathways that critically underpin developmental patterning and cell-type specification, for example, WNT, bone morphogenetic protein (BMP) and fibroblast growth factor (FGF)[26]. Yet, few studies have characterized the proteome in early mammalian developmental contexts, and, to our knowledge, none in human post-implantation embryos or gastruloids[13,27].

In this Article we describe the generation of a foundational resource to understand the temporal dynamics of gastrulation using high-throughput quantitative mass spectrometry to profile proteins and phosphosites across four key stages of gastruloid differentiation. We map the dynamics of hundreds of known protein complexes and identify additional proteins whose temporal profiles correlate with specific complexes, suggesting cooperative relationships during early development. With experimentally matched RNA-seq data, we identify temporal and pathway-specific discordance between the transcriptome and proteome. We map the dynamics of thousands of phosphosites, predict stage-specific kinase activities across gastruloid development, and observe that MAPKAPK2 regulates pluripotency exit in gastruloids. Finally, we leverage co-regulatory protein networks to establish roles for DPYSL4 and PRKACB in gastruloid development. Altogether, our work lays the groundwork for bridging transcriptomic and proteomic views of early mammalian development.

## Results

### Quantifying the dynamic proteome from ESCs to gastruloids

We profiled the dynamics of RNAs, proteins and phosphosites in human RA-gastruloids[10] and conventional mouse gastruloids[9] corresponding to four stages of gastruloid differentiation: pre-implantation 'naïve' ESCs, post-implantation 'primed' ESCs, post-symmetry-breaking 'early' gastruloids and anterior–posterior elongation/patterning 'late' gastruloids (Fig. 1a and Extended Data Fig. 1a). We analysed two human ESC lines (H9 and RUES2-GLR) to assess inter-cell-line variation[28,29] (Fig. 1b). All data were analysed in biological duplicate (transcriptomics) or triplicate (proteomics, phosphoproteomics) (Extended Data Fig. 1b–e). Replicates for each data type were also tightly grouped by principal components analysis (PCA). In human gastruloids, generally, PC1 separated naïve H9 ESCs from other samples, and PC2 broadly correlated with developmental progression. In mouse, PC1 generally separated late gastruloids from other samples, and PC2 once again resolved developmental progression (Extended Data Fig. 1f).

We quantified 7,352 human and 8,699 mouse proteins (Extended Data Fig. 1b and Supplementary Table 1), and measured proteins from all 34 annotated subcellular locations[30] (Extended Data Fig. 1g). The pluripotent markers *NANOG* and *POU5F1* were highly abundant in ESCs, while mesendoderm marker *TBXT*[31] and neural tube marker *PAX6*[32] were abundant in early- and late-stage gastruloids, respectively (Fig. 1c). Stage specificity was observed for proteins such as naïve epiblast marker SUSD2[33] in naïve H9 cells, while TBXT and NCAM were specific to early- and late-stage gastruloids, respectively (Fig. 1d). Interestingly, retinoic-acid binding protein CRAPBP2[34] was detected only in human samples after the addition of retinoic acid[35]. For mouse *Sox2*/Sox2, we observed consistent dynamics for messenger RNA (mRNA) and protein abundance (Fig. 1c,d). Protein levels for the pluripotency marker SOX2 dropped in early gastruloids before increasing in late gastruloids. SOX2 endogenously tagged with mCitrine confirmed that this pattern was driven by neural-cell populations (neural progenitors, neural crest and neural tube; Fig. 1c,d and Extended Data Fig. 1h). By quantitative phosphoproteomics[17,36], we also mapped the temporal dynamics of human and mouse phosphosignalling (Fig. 1e). Phosphorylation of the methyltransferases DNMT3B (Ser100, human) and Dnmt3a (Thr257, mouse) decreased during gastruloid development, potentially related to previous reports of DNA hypomethylation in ground-state pluripotency and increased methylase activity during differentiation[26,37–41]. Compared to recent

mouse gastruloid datasets[42], our work quantified 3,290 additional mouse proteins (65% more) and 2,303 additional homologous human proteins (46% more) (Extended Data Fig. 2a,b). Strong overlap with gastruloid and embryonic proteome datasets[42–44] support the interpretation that we had sampled biologically relevant temporal protein changes (Extended Data Fig. 2c,d). The increased depth of the proteome sampled over the course of gastruloid differentiation also enabled temporal co-regulatory analysis at the level of proteins, complexes and phosphosignalling.

### Time-resolved proteomics reveals coherent shifts across gastruloid development

To identify proteins with similar temporal dynamics, we merged the human and mouse proteomic datasets by orthology and subjected them to hierarchical clustering (Fig. 2a). Focusing on ten protein sets with similar dynamics across both species ('clusters'), Gene Ontology (GO) analyses[45] identified significantly enriched cell division and DNA repair (cluster 1), mitochondria and aerobic respiration (cluster 2), RNA biogenesis (cluster 3), cilia and pattern specification (cluster 4), small-molecule metabolism (cluster 6), extracellular matrix (ECM) organization (cluster 7) and tube development (cluster 8) (Fig. 2b and Supplementary Table 2). These enrichments suggest that the proteins that underlie these biological processes are coordinated during gastrulation.

Across adjacent timepoints in each species we identified thousands of differentially abundant proteins (DAPs; Extended Data Fig. 3a,b). Owing to cell-line differences, we refrained from directly comparing naïve H9 cells to the other stages. However, naïve H9 cells tended to exhibit a high number of both DAPs (3,499 DAPs comparing naïve and primed states of pluripotency) and differentially expressed transcripts (DETs; Extended Data Fig. 3a). SUSD2, whose expression marks pre-implantation epiblasts in human blastocysts, was detected only in naïve ESCs, and SOX2 and NANOG were enriched in primed ESCs (Fig. 3c). When compared to primed ESCs, DAPs in naïve ESCs were enriched for proteins involved in ECM organization, and primed cells were enriched for proteins involved in nucleotide metabolism. Comparing primed RUES2-GLR ESCs to early human RA-gastruloids, we identified 3,207 DAPs, including SOX2 enrichment in primed ESCs, and TBXT and CDX2 enrichment in early human RA-gastruloids. DAPs upregulated in early gastruloids mapped to actin filament organization and cytoskeletal processes, and DAPs downregulated mapped to mitochondrial processes (Extended Data Fig. 3d). Comparing early versus late human RA-gastruloids, we identified 767 DAPs, including downregulation of TBXT, caudal axial progenitors marker WNT8A and presomitic mesoderm marker TBX6, and upregulation of advanced cell-type markers including PAX3 (dorsal somites and neural tube), SOX1 and SOX2 (neural tube) and cardiomyocytes (MEIS1) (Fig. 2c).

To identify the cell types driving bulk proteomic observations, we compared our dataset with existing gastruloid scRNA-seq datasets[10]. We focused on seven proteins with characteristic upregulation in early gastruloids (TBXT, WNT8A, TBX6, APLNR), late gastruloids (SOX2, PAX3) or both (NEBL) (Extended Data Fig. 5a). In early gastruloids, TBXT was predominantly expressed in neuromesodermal progenitors (NMP) and axial mesoderm, and WNT8A and TBX6 were enriched in NMP, nascent mesoderm and primitive streak populations. APLNR (mesoderm development[46]) was broadly expressed across mesodermal lineages, suggesting that both nascent and emergent mesoderm populations contribute to its bulk protein profile. In late gastruloids, SOX2 was specifically expressed in neural progenitors and neural tube cells. PAX3 expression was primarily driven by neural and somite populations. Interestingly, NEBL protein levels in late gastruloids tended to reflect expression in cardiac cell populations, whereas in early gastruloids it lacked clear cell-type specificity (Extended Data Fig. 5b,c). SOX2 and ZIC2 were highly correlated at the protein level, and their scRNA-seq profiles suggested both were expressed in neural cell types. Upon

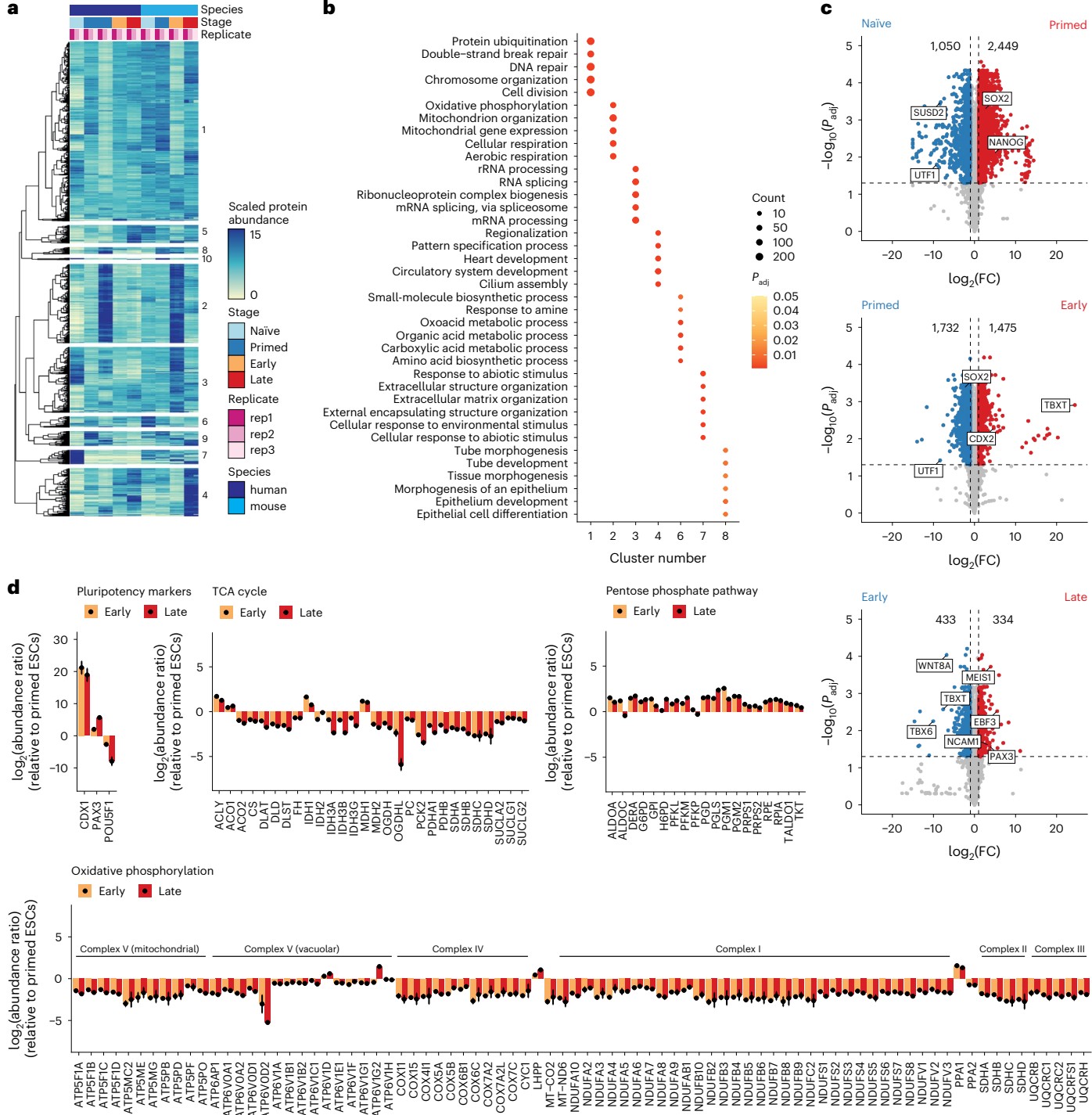

**Fig. 2 | Time-resolved proteomics reveals biologically coherent shifts across gastruloid development. a**, Heatmap depicting the temporal dynamics of protein expression across human and mouse gastruloid differentiation samples and replicates. Labels on the right of the heatmap indicate cluster number. **b**, Dotplot showing the top GO-terms enrichment across clusters. Clusters 5, 9 and 10 did not have significantly enriched GO terms. Significance was determined using a one-sided hypergeometric test. The colour scale indicates the Benjamini–Hochberg (BH)-adjusted $P$ values (correcting for multiple hypothesis testing). Size of dots corresponds to the number of proteins associated with a particular GO term. **c**, Volcano plots of the protein expression changes across consecutive

stages of human gastruloid differentiation, where the $x$ axes represent the $\log_2$(fold change) between two adjacent timepoints and the $y$ axes represents the negative $\log_{10}$ of the BH-adjusted $P$ value (correcting for multiple hypothesis testing). Numbers indicate the counts of differentially abundant proteins in each condition. Significance was calculated using the standard $t$-test. **d**, The $\log_2$(protein abundance ratio) of early (yellow) or late (red) gastruloids compared to primed human ESCs (RUES2-GLR) for proteins associated with pluripotency and central metabolism including the TCA cycle, the pentose phosphate pathway and oxidative phosphorylation. Mean abundance ratios (from three biological replicates) are indicated with dots, and error bars represent s.d.

immunostaining, ZIC2 was in nuclear bodies as previously reported[30], and colocalized with SOX2 to neural cell types. These observations suggest that our data capture at least some cell-type-specific expression patterns for major lineages.

Comparing H9 versus RUES2-GLR human primed ESCs, we detected 3,047 DAPs (Extended Data Fig. 3b). Although both cell lines expressed characteristic primed ESC markers (for example, SOX2 and NANOG), the DAPs largely mapped to mitochondrial processes

(respiration, oxidative phosphorylation), which are upregulated in primed RUES2-GLR relative to primed H9 ESCs. Conversely, DAPs upregulated in primed H9 ESCs were enriched for cytoskeletal processes and translation (Extended Data Fig. 3c). This comparison reinforces that substantial differences exist between widely used human ESCs[47].

The proteomes of primed RUES2-GLR ESCs were highly enriched for mitochondrial processes relative to RUES2-GLR early gastruloids (Extended Data Fig. 3c,d), suggesting that these processes are downregulated over the course of gastruloid differentiation. To determine whether this downregulation was specific to a subset of mitochondrially mediated metabolic pathways, we compared primed human ESCs with early and late gastruloids (all RUES2-GLR-derived) broken down by pathway. Intriguingly, we observed highly consistent levels of downregulation of mitochondrial proteins involved in the tricarboxylic acid (TCA) cycle and oxidative phosphorylation, and upregulation of proteins involved in the pentose phosphate pathway. Within oxidative phosphorylation, this consistency extended to individual protein complexes (Fig. 2d). Thus, the levels of mitochondrial machinery appear highly coordinated during gastruloid differentiation, consistent with studies of metabolic complexes during mammalian ageing[17]. Downregulation of mitochondrial activity was also observed in H9 early gastruloids, despite lower OxPhos protein levels in H9 primed ESCs (Extended Data Fig. 3e,f).

Across mouse gastruloid development, we observed similar numbers of DAPs (Extended Data Fig. 3g) with expected stage-specific patterns; for example, pluripotency markers Sox2 and Nanog were enriched in naïve mESCs compared to primed mESCs. Similarly, mesenchymal cell marker Bmp7 was enriched in early mouse gastruloids compared to primed mESCs, and endoderm marker Sox17 was enriched in late gastruloids compared to their early counterparts. To analyse conserved protein expression dynamics, we compared fold changes across stage transitions for orthologous human and mouse proteins. We observed modest positive correlation in the naïve to primed ($r_{Pearson} = 0.17$) and early to late ($r_{Pearson} = 0.5$) transitions, but strong anticorrelation in the primed to early transition ($r_{Pearson} = -0.8$). This anticorrelation was driven by the aforereferenced elevated levels of mitochondrial proteins in primed RUES2-GLR ESCs, whose metabolic state better matches early mouse gastruloids than primed mESCs (Extended Data Fig. 3h).

Despite species-specific protocol differences (for example, the tenfold lower number of starting cells for mouse gastruloids), the downregulation of oxidative phosphorylation primed ESCs to early human gastruloids is mirrored in early to late mouse gastruloids (Extended Data Fig. 4a,b). Furthermore, these trends in early versus late mouse gastruloids reproduce (providing independent confirmation), extend (by showing homologous patterns in human gastruloids) and add resolution to (by profiling more proteins) similar observations by Stelloo and colleagues[42] in mouse gastruloids (Extended Data Fig. 4c–e).

## Co-regulation analysis maps cooperative protein associations to protein complexes and pathways

Given that proteins belonging to shared modules (for example, oxidative phosphorylation) were coherently regulated across gastruloid development (Fig. 2d), we explored co-regulation among members of specific pathways or complexes. Co-regulation analysis, that is, calculating pairwise correlations of protein abundances across samples, can elucidate coordinated protein functions such as macromolecular complexes and biochemical pathways[48–52]. Correlated and anticorrelated edges within the resulting networks can reveal effects including direct protein interactions[53], signalling cascades[54,55] and cell-state-specific roles[56]. Proteome-based coexpression has been shown to outperform transcriptome-based coexpression for predicting gene function[57]. Consistent with this, pairwise correlations of glycolysis and TCA-cycle genes in our data revealed coherent intra-pathway correlations and inter-pathway anticorrelations at the protein level that were not recapitulated at the RNA level (Extended Data Fig. 6a).

We calculated correlations ($r_{Pearson}$) between all 19.6 million possible pairs of the 6,261 proteins that were successfully quantified in 18 primed ESC or gastruloid samples. Proteins within known complexes were generally highly correlated, for example, TUBG1 and TUBGCP2, which constitute the γ-tubulin ring complex[58], while TUBG1 was anticorrelated with the ATPase ATP1A1. Across all pairs, we observed a bimodal distribution of $r_{Pearson}$, but a similar analysis was not seen in permuted control samples (Fig. 3a,b).

We focused on pairs that were either strongly correlated ($r_{Pearson} \geq 0.95$) or anticorrelated ($r_{Pearson} \leq -0.95$) at a false discovery rate (FDR) of 1% (Fig. 3c). The resulting network consisted of 5,681 nodes (proteins) and 489,417 significant correlations or edges, of

---

**Fig. 3 | Co-regulation analysis maps cooperative protein associations to known protein complexes and pathways. a**, Scatterplots comparing abundances across selected protein pairs across samples. **b**, Distribution of $r_{Pearson}$ based on observed (top) and permuted (bottom) data. The observed distribution was obtained by calculating $r_{Pearson}$ across all possible protein–protein pairs. Permuted distributions were generated by randomly sampling 50,000 protein pairs after randomly shuffling their respective timepoints ten times each before calculating $r_{Pearson}$. Colours indicate strongly correlated (≥0.95; blue) or anticorrelated (≤−0.95; red) edges. **c**, Distribution of protein edge counts across the trimmed correlation network. On average, each protein in the network participated in 174 edges ± 195 edges. **d**, Ratio of enrichment for the annotated edges in the correlation network ('observed network') compared to the expected edge annotation frequencies across Gene Ontology biological process (GOBP), cellular component (GOCC), molecular function (GOMF), localization, pathways, protein–protein interactions (PPIs, BioPlex) or protein complexes. Specifically, we calculated the enrichment for annotated edges as the fraction of annotated edges per category in the observed correlation network divided by the fraction of annotated edges among all possible edges involving the 5,227 proteins in the correlation network. The expected frequency of annotated edges was calculated by generating all possible pairs from 5,227 human proteins (UniProt, July 2024) and computing the number of pairs explained by each functional category. **e**, Network analysis identifies known associations between proteins for BMP1 and RPL7A. **f,g**, Network structure of the 26S proteasome (**f**) and citric acid cycle pathway (**g**). Magenta nodes indicate known complex members annotated either from CORUM or EMBL ComplexPortal for protein complexes, or from BioCarta,

KEGG, the Protein Interaction Database (PID), Reactome and WikiPathways (WP) for biochemical pathways. Blue edges indicate positive correlations between nodes, and red edges indicate anticorrelations. **h,i**, Cooperative proteins are highly correlated with members of established protein complexes, including the NuA4 chromatin remodelling complex (**h**) and the Chaperonin-containing T (TRiC/CCT) complex (**i**). Magenta nodes indicate subunits of a given complex, and orange nodes indicate cooperative proteins, that is, proteins with profiles correlated to proteins constituting a particular protein complex. Cooperative node sizes indicate the negative $\log_{10}$ of the BH-adjusted $P$ value after computing significance from a one-sided Fisher's exact test to determine the cooperative association of a protein to a particular module. Blue edges indicate correlated edges, and orange edges link cooperative proteins to members of a particular module. **j**, Bioplex interaction network of the TRiC/CCT complex. Orange nodes are cooperative proteins with profiles correlated to proteins found in the TRiC/CCT complex. Grey edges indicate BioPlex evidence. **k**, Histogram of protein complexes (*x* axis) and their respective numbers of cooperative proteins (*y* axis). **l**, Heuristic to identify shared cooperative proteins between complexes. **m**, Heatmap depicting a subset of shared cooperative proteins across manually curated EMBL ComplexPortal protein complexes, namely exosomes, SWI/SNFs, ATAC remodellers, nucleosome remodellers (NuRDs) and histone acetyltransferase (HAT) and deacetylase (HDAC) complexes. The heatmap is coloured by Jaccard similarity coefficients calculated from overlapping sets of cooperative proteins between protein complex pairs, and clustered using Euclidean distances with average linkage.

which 62% were positively and 38% negatively correlated (Fig. 3c, Extended Data Fig. 6b,c and Supplementary Table 3). We trimmed our network to 5,227 proteins by retaining only the canonical isoforms detected in our datasets, and validated positively correlated edges by mapping the resulting network onto the databases cataloguing known gene ontologies[45], subcellular localizations[30], biochemical pathways[59–63], protein–protein interactions[64] and protein complexes[65,66]. The proportion of annotated edges that were positively

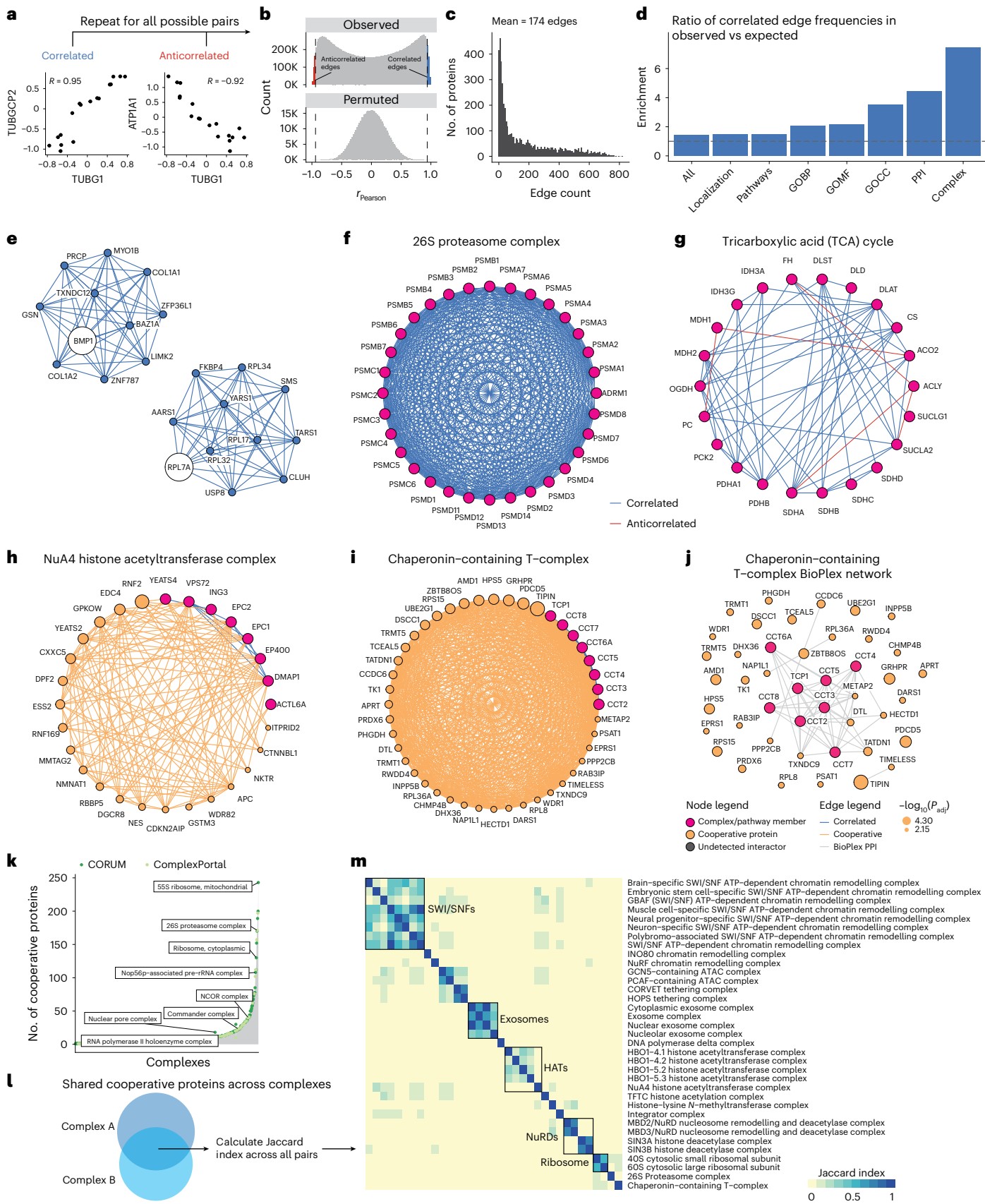

correlated edges varied by database, for example, 73–92% for proteins with shared GO annotations, subcellular localization or pathway databases, but 93% for proteins previously reported to interact, and 97% for proteins belonging to the same complex (Extended Data Fig. 6d and Supplementary Table 4).

In the trimmed network, 37.8% of positively correlated edges were explained by at least one established annotation, a 1.4-fold enrichment over the 26.7% of all possible edges involving these 5,227 proteins that are annotated in these databases (Extended Data Fig. 6e,f). This was consistent with previous studies that attributed 34–42% of protein correlation network edges to previous annotations. Notably, those studies also required 41–375 different cell lines to generate co-regulation networks[53,56]. Moreover, our network's edges were only modestly enriched for shared subcellular localization (1.5-fold), but were strongly enriched for annotated protein–protein interactions (4.5-fold) and shared membership in a protein complex (7.4-fold) (Fig. 3d).

We leveraged the untrimmed network to positively map protein pairs to specific developmental genes or protein complexes (Extended Data Fig. 6f,g). Anecdotally, many known protein–protein interactions were recovered. For example, BMP1, a metalloprotease involved in ECM formation and procollagen processing[67], was highly correlated with collagens COL1A1 and COL1A2, whereas RPL7A, a large ribosomal subunit member, was highly correlated with other large ribosomal subunit members and transfer RNA synthetases (AARS1, TARS1, YARS1) involved in translation (Fig. 3e).

To investigate whether the correlation network recovered known protein complexes, we focused on 1,357 complexes from CORUM[65] or ComplexPortal[66] with 3+ subunits represented in our correlation network. An average of 80% of complex members were represented among the 5,681 proteins in the network (Extended Data Fig. 6h). Within the 26S proteasome, 29 of 33 (88%) proteins were represented, with 87% of all possible edges detected, and 100% of edges were positively correlated (Fig. 3h). Similar trends were observed for core metabolic modules, including in the citric acid cycle, for which 90% of edges connecting pathway members were positively correlated (Fig. 3g).

Beyond recovering known protein–protein relationships (37.8% of filtered network, Fig. 3e), we nominated potential developmentally associated relationships. Many of these are potentially driven by cell states unique to gastruloid development relative to common workhorse cell lines[53,56]. Drawing from previous proteomics studies[50,64], we defined a protein cooperativity metric to enrich the first-degree neighbours of complexes and pathways, termed 'cooperative edges', connecting cooperative proteins (Methods). We reasoned that if members of a complex were withheld from our analysis, our cooperative edge mapping framework should recover their association to the remaining protein complex network. For example, when ribosomal proteins were divided into 60S and 40S subunits, three large ribosomal subunit members (RPL5, RPL13A, RPL32) were among the top five cooperative hits for the 40S subunit (Extended Data Fig. 6i).

We identified 1,385 cooperative proteins associating with 218 ComplexPortal complexes[66] and 1,944 cooperative proteins associating with 524 CORUM complexes[65] (Supplementary Table 5). The number of cooperative proteins per complex was not correlated with the number of complex subunits (Fig. 3h–k and Extended Data Fig. 6j) or the number of complexes with which a given protein was cooperatively associated (Extended Data Fig. 6k). When comparing cooperative protein–complex relationships with protein–protein interaction databases[64], 1,610 cooperative edges (involving 18.5% of cooperative proteins) were annotated as interactors (Extended Data Fig. 6l). For example, in the Chaperonin-containing T (CCT) complex, five (13%) of the 36 most significantly cooperative proteins were BioPlex interactors, and nine (25%) were BioGrid interactors (Fig. 3i,j).

We reasoned that complexes with shared cooperative proteins might inform these proteins' functional roles. Jaccard similarity coefficients between pairs of complexes (Fig. 3i) revealed network structures

among overlapping cooperative protein sets (Fig. 3m). For example, exosome and histone acetyltransferase complexes each had discrete sets of cooperative proteins that overlapped with one another but not with other complexes. The 40S and 60S ribosomal subunits shared extensive cooperative protein overlap with each other and also with the 26S proteasome and the Chaperonin-containing TCP-1 complex. Additionally, SWItch/Sucrose Non-Fermentable (SWI/SNF) complexes shared cooperative proteins among themselves, with a subset also overlapping with tethering complexes, the ATAC coactivator[68], and histone methyltransferase complexes (Fig. 3m and Supplementary Table 6).

## Gastruloid stages and gene modules exhibit varying degrees of RNA–protein discordance

Previous studies across biological contexts have reported varying extents of concordance between mRNA and protein levels[15,16,18,53,69]. With experimentally matched bulk RNA-seq data, we assessed the extent to which transcript abundances were predictive of protein levels in developing gastruloids. Our transcriptome data confirmed the expected temporal trends and stage-specific markers (Fig. 1d). Of note, HOX genes[70] turned on with gastruloid induction in both species, both at the early stage in human gastruloids and the late stage in mouse gastruloids (Extended Data Fig. 7a).

RNA–protein abundances for 6,010 matched genes were modestly correlated, consistent with previous work[53] (mean $r_{Pearson}$ = 0.39; Fig. 4a and Supplementary Table 7). When highly correlated or anticorrelated ($|r_{Pearson}| \geq 0.75$), RNA–protein relationships were stratified by broad gene classes[71–74]; for example, genes associated with transcription tended to be positively correlated, while those associated with the ribosome tended to be anticorrelated (Extended Data Fig. 7b,c). Within GO biological processes, genes exhibiting positive RNA–protein correlation were enriched for cytoskeletal and organ morphogenesis terms, suggesting that RNA levels are a reasonable proxy for protein abundance for these processes (Fig. 4b and Supplementary Table 7). Protein complexes involved in transcription (for example, the SOX2–OCT4 complex, CTNNB1–EPCAM–FHL2–LEF1 complex and the mRNA decapping complex) and signalling pathways (WNT, MAPK) tended to be positively correlated (Fig. 4d,e).

At the level of GO biological processes as well as shared subcellular localization (Human Protein Atlas[30]), mitochondrial genes, particularly those involved in aerobic respiration, tended to have anticorrelated RNA and protein levels (Fig. 4b,c). This trend was driven by mitochondrial protein complexes (for example, Complex I) and pathways of central metabolism (for example, oxidative phosphorylation) (Fig. 4d,e and Supplementary Table 8). In the case of Complex I, previous work in HeLa cells[75] has demonstrated that proteins in this complex are rapidly degraded post-translationally, suggesting that these systems are regulated in a similar fashion during gastruloid development.

We next sought to better understand the relationship between RNA and protein abundance as a function of developmental stage. Across all genes within each stage, early mouse gastruloids exhibited substantially lower RNA–protein correlation than all other human or mouse stages ($r_{Pearson}$ = 0.26; Extended Data Fig. 7e). We defined a metric of discordance between RNA and protein measurements—the $\log_2$-transformed ratio of the average fold change of a protein to its corresponding RNA—at a given stage of gastruloid development (Methods). Discordance values close to 0 indicate comparable levels of RNA and protein, while positive discordance implies the protein is more abundant than its corresponding transcript and vice versa. Focusing on mouse gastruloids, Gata6 discordance was high at the naïve ESC stage (higher than expected protein, given RNA levels), whereas in late gastruloids, Gata6 protein–RNA discordance was low. In contrast, SOX2 transcript and protein abundance remained relatively consistent over time (Fig. 4f).

Overall, we observed varying discordance profiles across mouse gastruloid development (Fig. 4g and Extended Data Fig. 7e) and

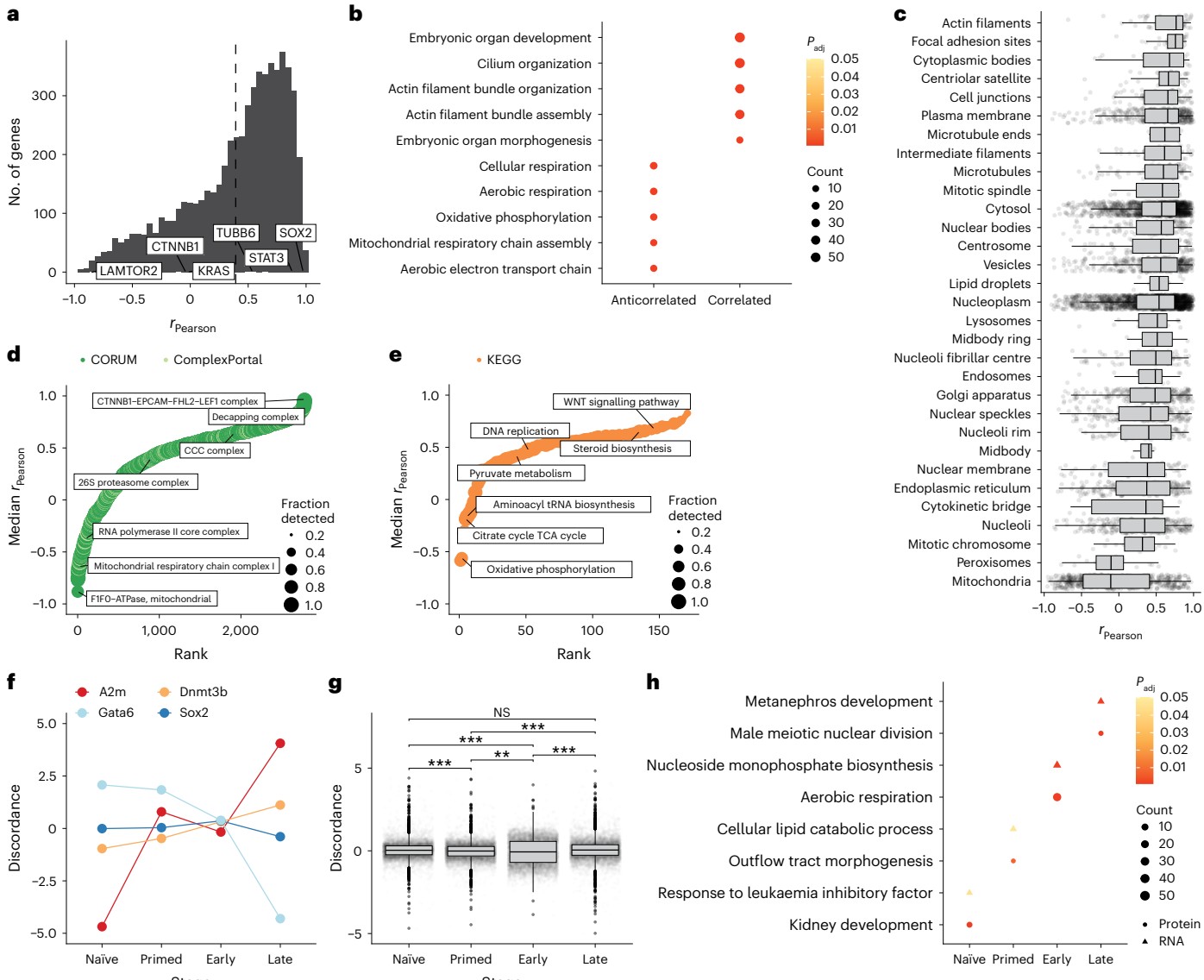

**Fig. 4 | Gastruloid stages and gene modules exhibit varying degrees of RNA–protein discordance. a**, Histogram of correlations ($r_{Pearson}$) between protein and RNA expression for all genes detected at the transcript and protein level in our samplings of human and mouse gastruloid development. The dashed line indicates the mean $r_{Pearson}$ across all genes. Representative genes with varying extents of correlation are highlighted. **b**, GO-term dotplot highlighting GO-defined biological processes exhibiting high RNA–protein correlation ($r_{Pearson} \geq 0.75$) or anticorrelation ($r_{Pearson} \leq -0.75$). The colour scale indicates the $P$ value adjusted using the BH procedure (correcting for multiple hypothesis testing), and sizes of dots indicate the number of genes detected within each term. **c**, Boxplot depicting the distribution of protein–RNA correlation ($x$ axis) across 6,010 genes as a function of subcellular location ($y$ axis). **d,e**, Rank plots of median RNA–protein $r_{Pearson}$ across protein complexes (**d**) or biochemical pathways (**e**). Colours indicate databases from which the module sets were

curated. **f**, Representative examples of RNA–protein discordance profiles (for any given gene, the mean across replicates is shown) for various stages. **g**, Boxplot depicting the distributions of RNA–protein discordances (for any given gene, the mean across biological replicates is shown). Boxplots show the median (centre line), 25th–75th percentiles (box), 1.5× the interquartile range (line; end points signify maxima and minima). Mean protein and RNA abundance were calculated from three and two biological replicates, respectively, for various mouse stages. Significance was determined using a standard $t$-test. (NS, not significant; \*\*$P < 0.01$, \*\*\*$P < 0.001$. **h**, Dotplot highlighting the biological processes significantly enriched in genes exhibiting protein-abundant (circles; discordance $\geq 1$) or RNA-abundant (triangles; discordance $\leq -1$) RNA–protein discordance. The colour scale indicates the $P$ value adjusted using the BH procedure (correcting for multiple hypothesis testing), and sizes of dots indicate the number of genes detected within each term.

applied GO enrichment analysis to genes with absolute discordance ratios greater than 1 (that is, protein either more or less abundant than expected, given RNA levels) across each developmental stage. In early mouse gastruloids, discordance tended to be driven by mitochondrial and metabolic processes (Fig. 4h and Supplementary Table 9). At the complex level, median RNA–protein discordance distributions were centred at 0 across developmental stages (Extended Data Fig. 7g). We next compared the fold changes of RNA and proteins between two temporally adjacent stages to delineate when discordance emerges or

resolves (Extended Data Fig. 7h). Most complexes had no significant differences in discordance between stages (for example, the core Mediator complex; Extended Data Fig. 7g,h). However, 12% (33/279) of complexes exhibited significantly different RNA and protein fold changes between early and late gastruloid stages, including cytoplasmic and mitochondrial ribosomal subunits, intraflagellar transport complex B and Complex I (Extended Data Fig. 7f,h).

Finally, we assessed whether the protein levels of transcription factors (TFs) could adjudicate potential targets (Extended Data Fig. 8a).

We focused on Sox2, Sox3, Tfap2c and Gata6, which exhibit distinct patterns of stage-specific protein expression in mouse gastruloids (Extended Data Fig. 8b). Transcripts for established targets of each of these TFs were upregulated in a corresponding pattern, for example, *Nanog* with Sox2, *Top2a* with Sox3, *Dppa3* with Tfap2c, and *Sox17* with Gata6 (Extended Data Fig. 8c)[76–81]. Although each of these TFs has thousands of targets according to the database TFlink[82], the RNA levels of only a subset of these are well-correlated with the TF's protein levels in our data ($r_{Pearson} \geq 0.9$), for example, 582 for Sox2 (3.4% of its targets), 122 for Sox3 (2.6% of its targets), 218 for Tfap2c (1.5% of its targets) and 347 targets for Gata6 (6.6% of its targets) (Extended Data Fig. 8d). These correlated targets were enriched for distinct biological processes: SMAD signalling, heart development and embryonic morphogenesis for Gata6; lysosome organization, autophagy and Leukemia Inhibitory Factor (LIF) response for Sox2; mitochondrial translation and RNA processing for Sox3 (Extended Data Fig. 8e). Given Sox2's elevated levels in naïve ESCs and early-stage gastruloids, we asked how discrete these sets were and if the same downstream targets were upregulated at both stages. Of 245 naïve-stage Sox2 targets and 298 early-stage Sox2 targets, 69 were enriched in both stages (Extended Data Fig. 8f). Naïve-stage targets were enriched for response to LIF, while early-stage targets were enriched for processes associated with cell adhesion, placenta development and meiosis (Extended Data Fig. 8g). Downstream targets of these four TFs were also enriched for protein–protein interactions (Extended Data Fig. 8h,i), suggesting that among large numbers of putative targets[82], these subsets would be good candidates for additional investigation in differentiating gastruloids.

## Quantitative phosphoproteomics reveals kinase activities across gastruloid development

Developmental programs are largely driven by signalling pathways that are regulated via phosphorylation[26]. We mapped the change in post-translational states of proteins across gastruloid development (Figs. 1a,b and 5a–d, Extended Data Fig. 9a,b and Supplementary Table 10). Human and mouse phosphosites were correlated with their protein abundances (human, median $r_{Pearson} = 0.71$; mouse, median $r_{Pearson} = 0.84$) and included residues of known stem-cell markers (Extended Data Fig. 9c,d). For example, phosphorylation of T35 and S207 on UTF1 decreased markedly through gastruloid development[39,83] (Fig. 5b). Immunofluorescence confirmed that H2AX S140 phosphorylation dynamically changes across human gastruloid development (Fig. 5e). H2AX S140 phosphorylation was highest in RUES2 primed ESCs, lower in H9 primed ESCs, and markedly reduced in early gastruloids before increasing again in late gastruloids (Fig. 5f). We further confirmed our ability to decipher the temporal dynamics of phosphosignalling by profiling mouse gastruloids treated with Chiron,

a GSK3 kinase inhibitor that activates WNT[84–86]. Gsk3a-activating phosphorylation at Y279 was inversely correlated with Chiron treatment, reflecting Chiron-dependent perturbation of Gsk3a activity during mouse gastruloid induction (Extended Data Fig. 9e). Additionally, kinase–substrate enrichment analysis[87–90] identified reduced activity of GSK3B and DYRK2 during gastruloid development[42] and increased inhibitory N-terminal phosphorylation of GSK3B[91–93].

Based on the role of phosphosignalling in key developmental transcriptional programs, we mapped phosphosites on the pluripotency markers POU5F1, SOX2 and NANOG, curated from previous studies[40,41] (Supplementary Table 11). Fourteen proteins were shared targets of POU5F1, SOX2 and NANOG and had phosphosites that exhibited temporal changes over the course of gastruloid development (Fig. 5c). For example, compared to naïve ESCs, DPPA4 phosphorylation (T215) was more abundant in primed ESCs; however, S570 and T514 on DPYSL2 tended to have more total phosphorylation in early and late gastruloids. DPPA4 is a known marker of pluripotency[94], whereas DPYSL2 is associated with nervous system development[95]. TCF20, a transcriptional co-activator associated with neurodevelopmental disorders, displayed two distinct phosphosite patterns, with residues S1522 and S1671 peaking in primed ESCs and correlating with pluripotency factors NANOG, POU5F1 and SOX2, whereas S574 was most abundant in early and late gastruloids when pluripotency factor abundance was low (Fig. 5d).

Conserved human and mouse phosphosites, including those on DYPSL2 and DNMT3B, exhibited highly consistent profiles across gastruloid differentiation. Notably for DNMT3B, conserved S100 phosphorylation was in a region important for DNA binding[96,97]. Contrastingly, HSP90AB1 S255 and RPS6KB1 S447 displayed species-specific phosphosite dynamics (Extended Data Fig. 9f). Kinase–substrate analysis predicted temporally dependent MAPKAPK2 phosphorylation of ZFP36L1 at Ser92 and PRKCI phosphorylation of ECT2 Thr359 (Fig. 5g–i). ZFP36L1, a downstream target of NANOG, peaked in early gastruloids (Fig. 5h), with an inverse relationship to NANOG abundance. ZFP36L1 Ser92 phosphorylation was correlated with the predicted activity of MAPKAPK2 (Fig. 5g,h). ZFP36L1 Ser92 may play a role in stabilizing ZFP36L1 levels and is associated with the degradation of pluripotency factors[98,99], and Ser92 phosphorylation correlated with MAPKAPK2 activity. Given the roles of ZFP36L1 in embryonic development[100], we hypothesized that MAPKAPK2 may play functional roles in symmetry-breaking and body-axis formation. In the presence of its inhibitor, MK2in1 (Extended Data Fig. 9i), gastruloids failed to elongate and displayed multi-axis morphology with the majority of late gastruloid cells expressing SOX2 (Fig. 5j–m). The elevation of SOX2 levels began after 48 h (Extended Data Fig. 9i) and continued until the end of gastruloid induction. Thus, coupled with previous

**Fig. 5 | Quantitative phosphoproteomics reveals kinase activities across gastruloid development. a**, The temporal dynamics of phosphorylated peptides across human gastruloid development. Rows indicate phosphosites, and columns sample type. The colour scale indicates the scaled TMTpro reporter ion abundance of individual phosphopeptides. **b**, Ridgeplots depicting the characteristic phosphorylation states within a given protein. **c**, Venn diagrams depicting the detection of phosphorylated proteins that are targets of pluripotency factors SOX2, POU5F1 and NANOG. The gene sets were curated from ref. 40. **d**, Phosphosites associated with downstream targets of pluripotency factors. The *y* axis indicates the log$_2$(abundance ratio) of early (yellow) or late (red) gastruloids to primed RUES2-GLR ESCs. Mean abundance ratios are indicated with dots, and error bars represent the s.d. calculated from three biological replicates. **e**, Bar plots of scaled H2AX pS139/pS140 abundance changes across human ESCs and gastruloid developmental stages. Mean abundance ratios are indicated with dots, and error bars represent the s.d. calculated from three biological replicates. **f**, Validation of the differential phosphorylation state of pS139/pS140 (red) in primed H9 (left) and RUES2-GLR (right) ESCs. The blue channel indicates DAPI. Scale bars, 25 μm. **g**, Heatmap

depicting the *z*-scores of kinase–substrate enrichment analysis. **h**, Representative examples of temporal phosphosite dynamics in comparison to their respective proteins and cognate kinases. The colour scale indicates the abundance *z*-score. ECT2 T359 was correlated with PRKCI, and ZFP36L1 S92 was strongly correlated with both MAPKAPK2 and AKT1. **i**, Network of kinases (circles) connecting to their substrates (rectangles). Pairs are annotated from PhositePlus. Edge colours indicate correlated (blue) or anticorrelated (red) relationships (absolute $r_{Pearson} \geq 0.5$) between kinase and substrate phosphosite nodes. **j**, Representative images of 120-h gastruloids cultured with DMSO (left) and the MAPKAPK2 inhibitor MK2in1 (right). Fluorescent images depict SOX2-mCitrine expression. BF, brightfield. Scale bars, 200 μm. **k**, Fraction of multi-axis gastruloids when treated with DMSO (control) and MK2in1 (MAPKAPK2 inhibitor). Fractions were calculated from 16 independent gastruloid observations for each condition. **l,m**, Boxplots depicting the differences in gastruloid area (**l**) and fraction of SOX2+ cells (**m**) when treated with DMSO (*n* = 9) and 10 μM MK2in1 (*n* = 15). Boxplots show the median (centre line), 25th–75th percentiles (box), 1.5× the interquartile range (line; end points signify maxima and minima). Significance was determined using a two-sided standard *t*-test.

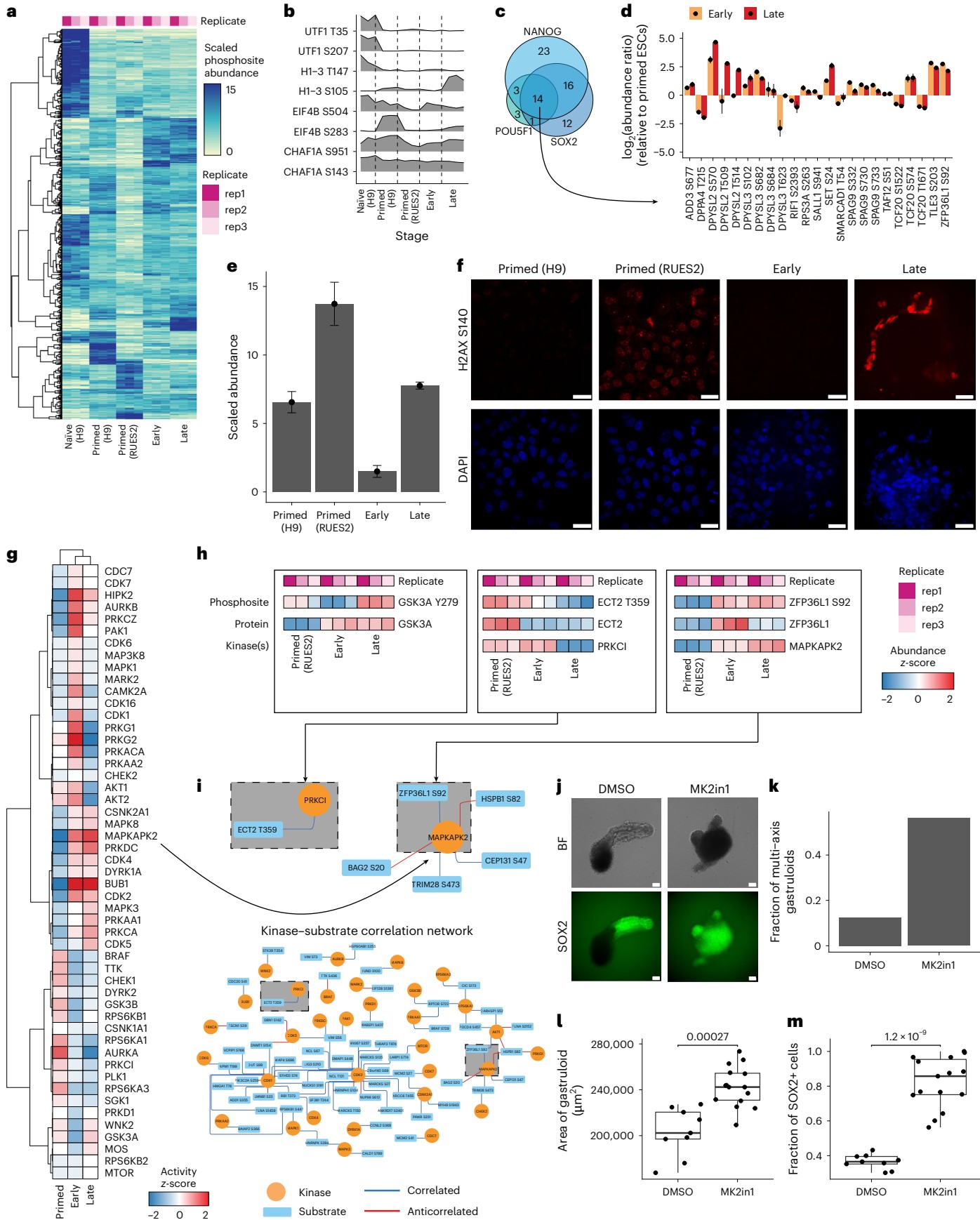

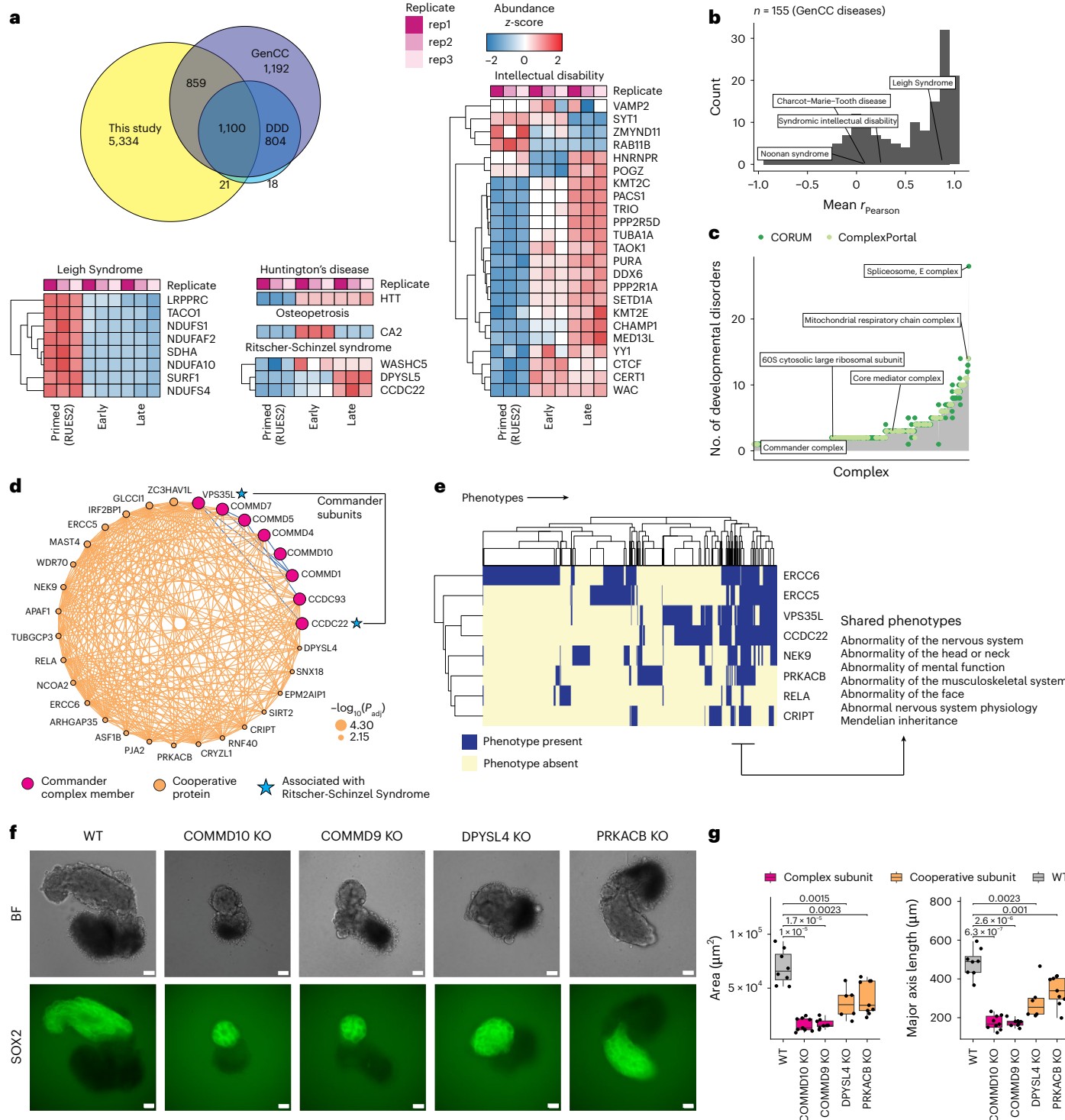

**Fig. 6 | Co-regulatory networks of protein dynamics in gastruloids link to shared phenotypes and developmental disorders. a**, Overlap of proteins detected across our dataset, GenCC and DDD. The heatmaps correspond to the temporal abundance changes of human proteins (rows) associated with a specific developmental disorder across human gastruloid development (columns). **b**, Distribution of mean Pearson correlation coefficients across GenCC disease sets. Only diseases with ≥2 genes are plotted. Mean $r_{Pearson}$ was calculated by averaging Pearson correlation coefficients for detected pairs of proteins in our dataset. **c**, Histogram of the number of developmental disorders associated (*y* axis) with genes comprising protein complexes (*x* axis). Complexes were associated with an average of 2.95 ± 2.68 developmental diseases. **d**, Co-regulation network of Commander complex subunits. Size of orange nodes indicates the

significance of cooperative association (−$\log_{10}$ of the adjusted *P* value; Fisher's exact test; Methods). Proteins associated with developmental disorders (blue stars) are linked to the Commander complex co-regulation network. **e**, Heatmap depicting the extent of shared phenotypic overlap (columns) across genes (rows) in the Commander subnetwork. **f**, Representative images of WT (wild-type), COMMD10 KO, COMMD9 KO, DPYSL4 KO and PRKACB KO gastruloids. *n* = 48 gastruloids per genotype. Scale bars, 200 μm. **g**, Boxplots comparing the area and major axis length of wild-type versus genetically perturbed gastruloids. Significance was determined using a two-sided standard *t*-test (*n* ≥ 8 for each genetic knockout). Boxplots show the median (centre line), 25th–75th percentiles (box) and 1.5× the interquartile range (line; end points signify maxima and minima).

work[40,41], phosphoproteome analyses identified potential routes of post-translational control of developmental chromatin regulators and TFs.

## Co-regulatory protein networks in gastruloids link shared phenotypes and developmental disorders

To investigate the temporal dynamics of proteins linked to developmental disorders, we intersected our dataset with the Gene Curation Coalition (GenCC)[101] and Deciphering Developmental Disorders (DDD)[102] databases. We quantified 1,980 proteins (27%) with at least one disease association in at least one of these databases (Fig. 6a and Supplementary Table 12). Anecdotally, genes linked to the same disease tended to be co-regulated across gastruloid development. For example, genes associated with Leigh Syndrome, a congenital early-onset neurological disorder associated with mitochondrial dysfunction, tended to be upregulated in primed ESCs, whereas genes linked with broad intellectual disability mostly showed increased abundance during the gastruloid stages (Fig. 6a). More broadly, proteins associated with the same GenCC disease class tended to be positively co-regulated (average $r_{Pearson}$ = 0.46; Fig. 6b).

Mapping disease-associated genes onto known protein complexes can inform their molecular roles in developmental disorders. There were 461 developmental disease-associated genes contributing to 217 ComplexPortal and 631 CORUM complexes (Supplementary Table 12), with the spliceosome E complex and mitochondrial respiratory Complex I associated with the most developmental disorders (Fig. 6c). Leveraging our co-regulatory analysis heuristic (Fig. 3a and Extended Data Fig. 6j) and protein interaction data from BioPlex and BioGrid, we identified 232 and 180 edges linking cooperative disease proteins to CORUM and ComplexPortal complexes, respectively (Extended Data Fig. 10a). Thus, functional proteomics can assign molecular functions for disease-associated genes, and nominate candidates in developmental contexts[103,104]. We illustrate this with examples involving Leigh Syndrome and Ritscher–Schinzel Syndrome.

Leigh syndrome is an early-onset mitochondrial neurometabolic disorder impacting the central nervous system[105]. Protein levels of 51 Leigh Syndrome-associated genes detected in our data were highly correlated with one another (Fig. 6b; mean $r_{Pearson}$ = 0.87). In our co-regulation network, they clustered with genes associated with central metabolism (for example, Complex I, ATP synthase) and were significantly enriched in an oxidative phosphorylation subnetwork ($P < 9.6 \times 10^{-15}$, Fisher's exact test, Extended Data Fig. 10b).

Ritscher–Schinzel syndrome is a developmental disorder characterized by abnormal craniofacial, cerebellar and cardiovascular malformations, classically associated with WASHC5 and CCDC22, and more recently with VPS35L and DPYSL5[106–111]. These four proteins were positively correlated (mean $r_{Pearson}$ = 0.78), with CCDC22 and VPS35L, clustering within a co-regulation network involving the Commander complex (Fig. 6c)[112], which is involved in the endosomal recycling of proteins[113]. Perturbations of Commander subunits COMMD9 and COMMD10 in mice have been previously linked to severe developmental defects and embryonic lethality[114,115]. Although all 16 Commander subunits were detected (Extended Data Fig. 10c), our co-regulation network contained eight subunits and 23 cooperative proteins (Fig. 6d).

Of the 31 proteins in the Commander co-regulatory network, seven had GenCC disease associations. We hypothesized that cooperative disease-associated proteins in the Commander network would share similar phenotypic features. Using gene–phenotype associations in the Monarch database[116], eight proteins in the Commander co-regulatory network had associations that clustered into shared sub-phenotypes (Fig. 6f). Unsurprisingly, the Ritscher–Schinzel syndrome genes *CCDC22* and *VPS35L* shared highly similar phenotypes. Broadly, Commander co-regulatory proteins exhibited overlapping phenotypic characteristics, including abnormalities of the nervous system, musculoskeletal system and in mental function (Fig. 6f).

Based on the Commander co-regulatory network, we perturbed two Commander subunits (COMMD9, COMMD10) and two co-regulatory proteins (DPYSL4, PRKACB) in human ESCs, and generated gastruloids from them. COMMD9 and COMMD10 knockouts failed to elongate with gastruloid induction, resulting in abnormal neural-tube morphology. DPYSL4 knockouts phenocopied these defects (Fig. 6f). Perturbation of PRKACB also resulted in gastruloids with reduced areas, although the reduction in major axis length was less pronounced. Across knockouts, gastruloids had reduced areas and a pronounced reduction in major axis lengths (Fig. 6g and Extended Data Fig. 10d).

## Discussion

We have described an integrated proteomic, transcriptomic and phosphoproteomic resource profiling both mouse and human gastruloids, an increasingly widely used model of early mammalian development. Although the numbers of in vitro models of embryogenesis continue to expand and are increasingly characterized with single-cell genomics, only recently have they been phenotyped at the protein level. For example, a recent study applied mass spectrometry to map temporal protein dynamics across stages of mouse gastruloid development, yielding insights into germ-layer proteomes and phosphorylation states[42]. However, this study was restricted to conventional mouse gastruloids. Here, we have extended multi-omic approaches to a human model of gastrulation to enable cross-species comparisons and explore additional developmental states spanning pre- and post-implantation to gain a more comprehensive view of gastrulation.

Metabolically, TCA-cycle proteins tended to be upregulated in primed human ESCs relative to late gastruloids, whereas glycolytic proteins showed the opposite trend, consistent with previous studies demonstrating the metabolic shift to glycolysis in post-implantation embryos[117,118]. Notably, while transcriptomic studies suggest a bivalent metabolic state in epiblast cells[118], primed RUES2-GLR cells had elevated oxidative phosphorylation protein levels compared to both late gastruloids and primed H9 cells. However, oxidative phosphorylation was downregulated in gastruloids from both cell lines, suggesting that metabolic shifts underlie early gastruloid development. The elevated levels of glycolysis at later gastruloid stages are consistent with previous studies linking glycolysis to somite formation, occurring in human RA-gastruloids from 96 to 120 h post induction[119–121]. Future profiling of neural or somite organoids may reveal how metabolic states shape or are shaped by differentiation.

Our data enabled comparison of protein dynamics and conservation across human and mouse gastruloid development. Although late gastruloids were modestly correlated across species, key developmental markers displayed conserved patterns of expression. Pluripotency markers (POU5F1, NANOG, CDH1) were decreased in late gastruloids relative to their stem-cell progenitors. Conversely, ZEB2 (epithelial–mesenchymal transition[122]), SOX9 (neural crest), CDX2 (caudal axial stem cells) and MEIS1 (cardiomyocytes) all increased. Conserved upregulated processes include cell differentiation, organ morphogenesis and heart/muscle development, while conserved downregulated processes include amino-acid metabolism and transport. This conservation was evident despite substantial protocol differences (for example, starting cell number, induction timing), suggesting that these are robustly conserved features.

Surprisingly, primed RUES2-GLR proteomes were most similar to early mouse gastruloids, driven by mitochondrial protein upregulation and suggesting RUES2-GLR cells may already be primed towards gastrulation at the protein level. This highlights potential species-specific differences in staging. So, although our study is a starting point for cross-species comparisons, more work is needed to understand the extent of cell line-specific and species-specific differences, that is, through more continuous temporal sampling and computational staging between species[10].

In gastruloids, we observe moderate correlation between transcript and protein abundances, with a clear discordance in mitochondrial oxidative phosphorylation genes but not for WNT signalling and steroid biosynthesis. Our findings align with studies mapping RNA–protein relationships in developmental contexts[16,19–23] and highlight the need to study multiple biomolecular layers—for example, the transcriptome, proteome and their interactions—during development. The discordance of oxidative phosphorylation genes in both human and mouse gastruloid development suggests that post-transcriptional regulation of metabolic machinery is evolutionarily conserved during early lineage specification, and the heightened discordance at the earliest stages points to a developmental window of active proteome remodelling during cell fate transitions. Applying ribosome profiling[123] could disentangle translational control from protein turnover as a driver of these discordances, and matched single-cell proteomics and transcriptomics[124] would enable cell-type resolution of these effects.

Using phosphoproteomics, we have identified MAPKAPK2 as a regulator of human gastruloid development, a role not previously characterized outside of cancer and stress response contexts. Our results implicate this kinase in symmetry-breaking and pluripotency exit during human gastrulation, and highlight how phosphoproteomics data can reveal post-translational regulators of early human development that are invisible to transcriptomic approaches alone.

We mapped the co-regulation of hundreds of protein complexes and pathways during gastruloid development. From co-regulatory networks, we identified cooperative proteins associating with complexes, suggesting developmental roles in gastrulation, including chromatin remodellers (SWI/SNFs), histone methyltransferases (SIN3A/SIN3B), and acetyltransferases (HBO complexes). Our work highlights co-regulatory networks as hypothesis generators for understudied genes, particularly those related to disease. Focusing on the Commander complex (linked to Ritscher–Schinzel syndrome[125,126]), perturbations to co-regulated proteins DPYSL4 (associated with neurite initiation and dendrite growth of hippocampal neurons[127,128]) and PRKACB (associated with neural-tube defects[129]) produced similar morphological phenotypes as Commander subunit knockouts. These results support network-based predictions as powerful starting points for understanding gene function in gastrulation. More broadly, this study offers scalable, protein-focused approaches extending beyond nucleic acid-centric assays to phenotype gastruloids and other embryo models[5].

Although our work offers insight into gastruloid development, several limitations merit emphasis. First, finer temporal sampling would enhance the resolution of the developmental dynamics. Second, although we quantified ~7,500 human and ~8,700 mouse proteins—a substantial portion of the observable proteome[130]—targeted workflows could improve coverage of low-abundance developmental proteins. Third, bulk measurements lack cell-type resolution, which might benefit from characterization with fluorescence activated cell sorting[42] or single-cell proteomics. Fourth, although co-regulatory protein networks provide strong starting points for inferring gene function in development, they are correlative. Integrating structural modelling and interactome mapping would improve hypotheses for validation. Fifth, the absence of standardized mammalian gastruloid techniques makes it difficult to distinguish species-specific variation from protocol-specific variation. Profiling of gastruloids under different conditions (for example, varying starting cell numbers[131,132]) and benchmarking against in vivo models may be necessary to identify the effects of protocol variation and establish the physiological importance of gastruloid-derived proteomic signatures. Species comparisons would further benefit from harmonization of mouse and human protocols.

Finally, gastruloids remain imperfect surrogates for embryogenesis, and future multi-omic studies will be needed to advance these models to better understand embryonic development.

## Online content

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

# Methods

## Ethics statement
All research conducted in this work, including the induction and cellular and/or molecular analysis of both mouse gastruloids and human RA-gastruloids, was reviewed and approved by Embryonic Stem Cell Research Oversight of the University of Washington (E0047-001). This work was performed in compliance with the principles laid out in the International Society for Stem Cell Research Guidelines for Stem Cell Research and Clinical Applications of Stem Cells[133]. No experiments involving human embryos and gametes were performed in this study. Both human and mouse gastruloids were cultured for no longer than five days after induction.

## Mouse cell lines
The E14Tg2a cell line was obtained from C. Schroeter (Max Planck Institute).

## Mouse naïve ESC culture
Mouse naïve ESCs were maintained in 2iLif medium[85] containing 3 μM CHIR99021 (Millipore Sigma, SML1046), 1 μM PD0325901 (Stemcell Technologies, 72184) and 1,000-U-ml⁻¹ LIF (Millipore, ESG1107) and passaged with TrypLE (Thermo, 12604021) every other day onto new wells, which were coated with 0.01% poly-L-ornithine (Millipore Sigma, P3655-10MG) and 300-ng-ml⁻¹ laminin (Corning, 354232).

## Mouse EpiLC differentiation
Mouse EpiLC differentiation was performed as previously described[134]. Briefly, $1 \times 10^5$ mouse naïve ESCs were seeded onto a well on a 12-well plate, which was coated with human plasma fibronectin (Thermo, 33016015) in EpiLC differentiation medium (N2B27 + 20-ng-ml⁻¹ ActivinA + 12-ng-ml⁻¹ bFGF + 1% KnockOut serum replacement (KSR)). The medium was changed a day after seeding. Day-2 EpiLCs were dissociated with TrypLE (Thermo, 12604021) and sampled.

## Mouse gastruloid induction
Mouse gastruloid induction was performed as previously described[9]. Briefly, mESCs cultured in 2iLiF medium were dissociated with TrypLE, and 300 cells were seeded into U-bottomed, non-adherent 96-well plates in N2B27 medium and kept for 48 h at 37 °C in a 5% $CO_2$ incubator. After 48 h, 150 μl of N2B27 containing 3 μM CHIR99021 was added to each well. At 72 and 96 h, 150 μl of medium was replaced with fresh N2B27 medium lacking CHIR99021. Mouse gastruloids were sampled at 72 and 144 h after induction.

## Human cell lines
Pluripotent stem cell lines, hESCs (RUES2-GLR), were gifted by A. Brivanlou (Rockefeller University). Chemically reset (cR) H9 naïve and primed cells were kindly gifted by A. Smith (University of Exeter).

## Human naïve ESC culture
Chemically reset (cR) H9 naïve hESCs were propagated in N2B27 with PXGL (P-1mM PD0325901, 2 mM X- XAV939, G- 2 mM Gö 6983 and L- 10-ng-ml⁻¹ L-human LIF) on irradiated mouse embryonic fibroblast (MEF) feeders as described previously[33,135,136]. Y-27632 and Geltrex (0.5 ml per cm² of surface area; Thermo Fisher Scientific, A1413302) were added during re-plating. To remove MEF cells, cells were passaged on Geltrex-coated wells at 1 μl cm⁻² and were repeatedly passaged by dissociation with Accutase (Biolegend, 423201) every 3–5 days for five successive passages.

## Human primed ESC culture
Human primed ESCs were cultured in StemFlex (Thermo, A3349401) on Geltrex (Thermo, A1413201) and were routinely passaged using StemPro Accutase (Thermo, A1110501) to new Geltrex-coated wells, as recommended by the manufacturer. For the first 24 h after passaging,

hESCs were cultured in StemFlex with 10 μM Rho Kinase inhibitor Y-27632 (Sellek, S1049) to prevent apoptosis.

## Human RA-gastruloid induction
Human RA-gastruloids were induced as described previously[10]. Briefly, ~2 × 10⁴ hESCs were plated onto a single well of a vitronectin-coated 12-well dish (Gibco, A14700) in Nutristem hPSC XF medium (Biological Industries, 05-100-1 A) in the presence of 10 μM Y-27632. After 24 h, the medium was replaced with NutriStem containing 5 μM Y-27632. At 48 h the medium was replaced with Nutristem containing 4 μM CHIR (Millipore, SML1046). At 72 h, the medium was replaced with NutriStem containing 4 μM CHIR and 500 nM RA (Millipore Sigma, R2625). Pre-treated cells were detached using StemPro Accutase, dissociated into a single-cells suspension, then 4,000 cells were inserted per well of a U-bottom-shaped 96-well plate with 50 μl of Essential 6 medium (Thermo, A1516401) containing 1 μM CHIR and 5 μM Y-27632. At 24 h, 150 μl of Essential 6 medium was added to each well. At 48 h, 150 μl of the medium was removed with a multichannel pipette, and 150 μl of Essential 6 medium containing 5% Matrigel and 100 nM RA was added and maintained at 37 °C and 5% $CO_2$ until 120 h. Human gastruloids were sampled at 24 and 120 h after induction.

## Perturbation experiments
**Genetic perturbations in ESCs.** Genetic perturbations in RUES2-GLR ESCs were performed as previously described using CRISPR-Cas9 RNA–protein complexes[10]. In brief, equal molar amounts of crRNA and tracrRNA (IDT; Supplementary Table. 13) were hybridized by heating at 95 °C for 5 min in a thermal cycler and cooling to room temperature for 10–20 min. AltR-Cas9 protein (IDT, 1081058) was added to the hybridized crRNA–tracrRNA mixture to assemble Cas9 ribonucleoproteins.

RUES2-GLR ESCs were dissociated with StemPro Accutase, the activity of which was quenched with DMEM-F12 nutrient mix supplemented with 10 mM Y-276322. For each perturbation, 200,000 cells were collected by centrifugation at 250$g$ for 5 min. Cells were resuspended in 20 μl of nucleofection buffer (16.4 μl Nucleofector solution + 3.6 μl supplement) provided in the P3 Primary Cell 4D-Nucleofector X kit S (Lonza, V4XP-3032). Ribonucleoproteins (3 μl) and 0.5 μl of AltR-Cas9 electroporation enhancer (IDT, 1075915) were added to cells before transferring them into 16-well Nucleocuvette strips and electroporated with the CA-137 nucleofection program. The nucleofected cells were transferred to a 12-well plate that contained NutriStem or StemFlex with 10 mM Y-27632 and, after 24 h, the medium was replaced with NutriStem without Y-27632. Cells were maintained until they reached 50–70% confluence. The electroporated cells were then transferred onto 0.5-μg-cm⁻² vitronectin-coated 12-well plates before proceeding with RA-gastruloid induction steps as described above.

**Chemical perturbations in gastruloids.** Stocks (10 mM) were prepared by resuspending MK2in1 (HY-12834, MedChemExpress) in dimethyl sulfoxide (DMSO). MAPKAPK2 perturbations were performed by inducing RA-gastruloids in the presence of 10 μM MK2in1 added on day 0 and replenished on day 2.

## Immunostaining of ESCs and gastruloids
The ESCs were fixed and stained as described previously. Briefly, ESCs were cultured on Matrigel with StemFlex or mTeSR+ in glass-bottomed 12-well plates (Cellvis, P12-1.5H-N). The cells were washed three times with phosphate-buffered saline (PBS) before a 30-min fixation in 4% paraformaldehyde. The cells were then washed three times with PBS before permeabilizing with 0.1% Triton X-100 (in PBS) for 30 min, then they were stained with primary antibodies diluted to the recommended working concentrations in Cell Painting Buffer[138] (1× Hanks' balanced salt solution, 1% bovine serum albumin and 0.01% sodium azide) with 0.75% Triton-X-100 for 1 h while shaking. The cells were

then washed three times in PBS with Tween 20 (PBST; 0.2% Tween-20) and stained with secondary antibodies (diluted 1:500 or 1:1,000 in Cell Painting Buffer) for 1 h while shaking in the dark. The cells were washed three times with PBST and kept in the dark after staining. Cells were imaged in UltraPure saline sodium citrate (SSC) (Thermo Fisher Scientific, 15557044).

Gastruloids were fixed and stained as previously described[10]. Briefly, the gastruloids were fixed overnight in 4% paraformaldehyde at 4 °C. The following day, they were washed three times for 1 h each with PBST and incubated in blocking buffer (PBS containing 0.1% bovine serum albumin and 0.3% Triton X-100) overnight at 4 °C. Primary antibodies were then applied, diluted in blocking buffer to working concentrations as per the manufacturer's recommendations, and incubated overnight at 4 °C. Stained gastruloids were washed with washing buffer (PBS containing 0.3% Triton X-100), stained with secondary antibodies (diluted either 1:500 or 1:1,000 in blocking buffer) and 4′,6-diamidino-2-phenylindole (DAPI; diluted 1:1,000) overnight at 4 °C in the dark. The following day, the gastruloids were washed in blocking buffer and mounted in SlowFade gold antifade mountant (S36936, Thermo Fisher Scientific). The antibodies used in this study are listed in Supplementary Table 14. All samples were analysed with a Nikon Eclipse Ti2 confocal microscope (Supplementary Table 15) and analysed using Fiji[139] and the Python sci-kit-image[140]. When comparing pixel intensities across images, we normalized fluorescence intensities (for example, antibody) to that of DAPI (defined as normalized fluorescence).

### RNA-seq analysis
**Sample preparation.** Each stage consisted of two biological replicates collected within the same experimental batch to minimize batch effects. Approximately 0.5 million cells per replicate were collected across mouse and human cells across the four gastruloid developmental stages. DNA and RNA from each sample were isolated using the Qiagen AllPrep DNA/RNA kit (Qiagen, 80204). Approximately 500 ng of total RNA was used as input for library preparation. mRNAs were isolated using the NEBNext Poly(a) mRNA magnetic isolation module (NEB, E7490) and prepared for sequencing using the NEBNext UltraII RNA Library Prep Kit for Illumina (NEB, E7770).

**Sequencing and data analysis.** Concentrations of cDNA libraries across all samples were estimated using a Qubit system (Invitrogen) and/or visualized by a TapeStation (Agilent) to ensure standard ranges for library sizes. All libraries were dual-indexed with eight nucleotide indexes using NEBNext Multiplex Oligos for Illumina (Index Primers Set 1) and were sequenced on NextSeq 2000 (Illumina) either by the 2x150-bp or 2x50-bp configuration.

Basecall files were converted to fastq formats using bcl2fastq (Illumina) and demultiplexed on the i5 and i7 indexes. FastQC was performed to estimate the quality of the reads. Adapter trimming and filtering for low-quality reads was performed using Trimmomatic v0.39[141], either in paired-end or single-end mode, trimming low-quality reads (<2) at the ends and applying a four-base sliding window across reads, retaining those with average quality above 15. Depending on the species, trimmed reads were then aligned using STAR[142] to either the human GRCh38 or mouse GRCm39 reference assemblies. Human samples had an average unique mapping rate of 85.1%, while those of the mouse samples were 73.13%. Finally, count matrices for each species were generated with the bam files using FeatureCounts with default parameters.

### Mass spectrometry data collection
**Sample preparation.** For each stage analysed, we collected 1–2.5 million cells per replicate across four gastruloid developmental stages. To mitigate batch effects, all replicates from each developmental time point were collected together within the same batch. Stem cells across each stage were collected from culture plates by enzymatic dissociation

using Accutase (StemCell Technologies, 07920). As each gastruloid was cultured in a single well of a 96-well U-bottomed plate, gastruloids were first pooled together to reach the 2.5 million cell number and gently centrifuged at 500g for 5 min to remove growth media followed by Accutase treatment to dissociate the gastruloids. Once dissociated, Accutase treatment for both gastruloid and stem-cell samples was quenched by addition of a wash buffer consisting of either StemFlex or mTeSR+ along with rock inhibitor (Y-27632). Finally the cells were washed twice with PBS to remove cell debris, lysed cells and Matrigel from the samples. The samples were finally stored at −80 °C after aspirating the PBS, before proceeding to protein isolation.

Cell pellets were thawed on ice and resuspended in lysis buffer (8 M urea, 250 mM 4-(2-hydroxyethyl)piperazine-1-propanesulfonic acid (EPPS) pH 8.5, 50 mM NaCl, Roche protease inhibitor cocktail, Roche PhosSTOP). The cell pellets were homogenized using a 21-G needle to syringe pump lysate. Lysates were cleared by centrifugation at 21,130g at 4 °C for 30 min. Supernatants were placed in clean microcentrifuge tubes and a BCA assay (Pierce) was performed to determine protein concentrations. Lysate containing 25 μg of protein material for biological triplicates at each point of gastrulation were reduced and alkylated with 5 mM dithiothreitol (DTT) for 30 min at room temperature and 20 mM iodoacetamide (IAA) for 1 h in the dark at room temperature. The IAA reaction was then quenched with 15 mM DTT. Single-pot solid-phase sample preparation (SP3)[143] using Sera-Mag SpeedBeads was performed to desalt the reduced and alkylated samples. An on-bead protein digestion was performed by adding LysC at a 1:100 ratio (protease:protein) overnight (16–24 h) on a thermocycler at room temperature, then adding trypsin at a 1:100 ratio for 6 h at 37 °C at 900 r.p.m. TMTpro was used to label each sample at a 2.5:1 ratio of TMTpro reagents to the peptide mixtures for each sample. Samples were left at room temperature for 1 h for TMTpro labelling, and the labelling efficiency was verified to be >99% for lysines and >97% for N termini. The labelling reaction was quenched with 5% hydroxylamine diluted to a concentration of 0.3% for 15 min at room temperature. Samples were then placed on a magnetic rack to aggregate SP3 beads, and labelled peptide supernatants from each sample were pooled. The pooled sample was then partially dried down using a speed-vac instrument, and 10% formic acid was added to bring the pH of the pooled sample to below 3 for desalting. The pooled sample was desalted using a Sep-Pak C18 cartridge (Waters), then dried completely.

**Phosphoproteomics sample preparation.** The pooled sample was resuspended in 94 μl of 80% acetonitrile and 0.1% trifluoroacetic acid for $Fe^{3+}$-nitrilotriacetic acid (NTA) magnetic bead phosphopeptide enrichment[144]. Next, 100 μl of 75% acetonitrile/10% formic acid was added to a clean microcentrifuge tube, and the $Fe^{3+}$-NTA magnetic beads were washed twice with 1 ml of 80% acetonitrile and 0.1% trifluoroacetic acid and the supernatant removed. After the final wash, the peptides, in 94 μl of 80% acetonitrile and 0.1% trifluoroacetic acid, were added to the tube with the washed beads. The sample was vortexed and incubated for 30 min on a thermoshaker (250 r.p.m., 25 °C). After the incubation period, the sample was washed three times with 200 μl of 80% acetonitrile and 0.1% trifluoroacetic acid, and all flowthrough was saved in a clean microcentrifuge tube, as it contains non-phosphorylated peptides. We then added 100 μl of 50% acetonitrile and 2.5% $NH_4OH$ to elute phosphorylated peptides from the magnetic beads, then the sample was transferred to a tube with 100 μl of 75% acetonitrile and 10% formic acid. The phosphopeptide-enriched sample was dried immediately using a speed-vac and resuspended in 100 μl of 5% formic acid. A C18 stage tip was used to desalt the phosphopeptide-enriched sample. The sample was transferred to a mass spectrometry (MS) insert vial, which was placed within a microcentrifuge tube. The sample was placed in a freezer at −80 °C for 30 min, then dried completely in a speed-vac. The sample was resuspended in 10 μl of 2% formic acid and 5% acetonitrile within the MS insert vial.

**Total proteomics sample preparation.** The saved flowthrough was dried using the speed-vac, resuspended in 500 µl of 5% formic acid, then a Sep-Pak C18 cartridge (Waters) was used to desalt the sample. The flowthrough sample was dried completely in the speed-vac after desalting. The flowthrough sample was resuspended and neutralized in 1 ml of 10 mM ammonium bicarbonate/90% acetonitrile and again dried completely in the speed-vac. It was then resuspended in 115 µl of 10 mM ammonium bicarbonate and 5% acetonitrile, then 110 µl were transferred to a sample vial. High-pH reverse-phase high-performance liquid chromatography (HPLC) fractionation was performed on the flowthrough sample using an Agilent 1200 HPLC system. After HPLC fractionation[145], the fractions were dried in the speed-vac, resuspended in 100 µl of 5% formic acid, and cleaned using a C18 stage tip. Eluate from each stage-tipped fraction was placed in an MS insert vial and dried in vial. Fractions were then resuspended in 5 µl of 2% formic acid/5% acetonitrile within the MS insert vial.

## Mass spectrometry data acquisition
**Proteomics.** All analyses were performed using an Orbitrap Eclipse Tribrid mass spectrometer (Thermo Fisher Scientific), in-line with an Easy-nLC 1200 autosampler (Thermo Fisher Scientific). The peptides underwent separation using a 15-cm-long C18 column with 75-µm inner diameter, with a particle size of 1.7 µm (IonOpticks). Each fraction collected from the off-line fractionation was analysed using a 90-min gradient of 2% to 26% acetonitrile in 0.125% formic acid with a flow rate of 500 nl min$^{-1}$. The MS1 resolution was set to 120,000 with a scan range of 400–2,000 $m/z$, a normalized automatic gain control (AGC) target of 200%, and a maximum injection time of 50 ms. The field asymmetric waveform ion mobility spectrometry (FAIMS) voltage was cycled through activation at constant compensation voltages (CVs) of −40 V, −60 V and −80 V. MS2 scans were collected with an AGC target of 200%, maximum injection time of 50 ms, isolation window of 0.5 $m/z$, collision-induced dissociation (CID) collision energy of 35% (10-ms activation time), and 'rapid' scan rate. SPS-MS3[137] scans were triggered based on the real-time search (RTS) filter[36]. Briefly, RTS was run by searching species-specific UniProt protein databases (downloaded April 2023) for mouse (taxid: 10090) and human (taxid: 9606) with static modifications for carbamidomethylation (57.0215) on cysteines and TMTpro acylation (304.2071) on peptide N termini and lysines, variable modification of oxidation (15.9949) on methionines, one missed cleavage, and a maximum of three variable modifications per peptide. Scan parameters of the SPS-MS3 were set to collect data on 10 SPS ions at a resolution of 50,000, AGC target of 400%, maximum injection time of 150 ms, and a higher-energy collisional dissociation (HCD) normalized collision energy of 45%.

**Phosphoproteomics.** Duplicate injections (4 µl) were analysed on an Orbitrap Eclipse Tribrid mass spectrometer (Thermo Fisher Scientific) along with an Easy-nLC 1200 autosampler (Thermo Fisher Scientific). The peptides underwent separation using a 15-cm-long C18 column with a 75-µm inner diameter, with a particle size of 1.7 µm (IonOpticks). Each fraction was analysed using a 90-min gradient of 2% to 26% acetonitrile in 0.125% formic acid, with a flow rate of 400 nl min$^{-1}$. The MS1 scan resolution was set to 120,000 with a scan range of 400–1,800 $m/z$, a normalized AGC target of 200%, and a maximum injection time of 50 ms. The FAIMS voltage was cycled between compensation voltages of −40 V, −60 V and −80 V. MS2 scans were collected with an AGC target of 250%, maximum injection time of 35 ms, isolation window of 0.5 $m/z$, CID-multistage activation (MSA) collision energy of 35% (10-ms activation time), with additional activation of the neutral loss mass of n-97.9763, and the 'rapid' scan rate. For SPS-MS3 scans[137] a resolution of 50,000, AGC target of 300%, maximum injection time of 86 ms and HCD normalized collision energy of 45% were applied.

## Proteomic and phosphoproteomic data analysis
All supporting scripts have been generated using standard open-source software, packages and code, and are available from https://github.com/bbi-lab/Temporal-Gastrulomics. All processed data are available through the web application at https://gastruloid.brotmanbaty.org/.

**Peptide spectral matching.** Raw files were searched against the relevant annotated proteome from UniProt (Human, October 2020; Mouse, March 2021). Sequences of common contaminant proteins and decoy proteins were added to the UniProt FASTA file to also be searched. The Comet search algorithm[146] was used to match peptides to spectra with the following parameters: 20-ppm precursor tolerance, fragment_tolerance of 1.005, tandem mass tag (TMTpro) labels (304.207145) on peptide N termini and lysine residues, alkylation of cysteine residues (57.0214637236) as static modifications, and methionine oxidation (15.9949146221) as a variable modification. Phosphoproteomics runs were also searched for phosphorylation as a variable modification on serine, threonine and tyrosine residues (79.9663304104). Peptide-spectrum matches were filtered to a 1% FDR using a linear discriminant analysis[36]. Proteins were filtered to a FDR of 1% using the rules of protein parsimony and the protein picker methods[147]. For quantitation, Peptide-spectrum matches were required to have a summed TMTpro reporter ion signal-to-noise ration of ≥100 (ref. 137).

**Differential protein expression analysis.** DAPs between developmental gastruloid stages were identified as follows. For each protein, we calculated the log$_2$ ratios of mean abundance across two given timepoints and computed their $P$ values using a two-sided standard $t$-test. We corrected for multiple hypothesis testing by adjusting the $P$ values using the Benjamini–Hochberg (BH) procedure. We classified proteins as DAPs if they had an absolute fold change of greater than 2 and BH-adjusted $P$ value of <0.05 between two given timepoints.

**Protein module analysis.** All quantified proteins were mapped onto known TFs (curated from the Transcription Factor Database[148]), protein complexes (curated from CORUM[65] and EMBL ComplexPortal[66]), biochemical pathways (curated from BioCarta[60], KEGG[61], PID[63], Reactome[62] and WikiPathways[59]), subcellular localization (curated from Human Protein Atlas[30,149]) and Gene Ontology (GO) terms[45]. For biochemical pathways and complexes, we filtered module sets to those where we detected more than two members. With respect to subcellular locations, if a protein in the Human Protein Atlas was listed as localized to multiple regions in its main subcellular location, we considered each location as unique. We avoided searching our data against overly broad descriptions of GO terms by filtering for terms containing fewer than or equal to 150 genes and greater than two members detected in our data. All mappings were based on UniProt annotations[72,150] unless otherwise stated.

**Correlation network construction and network analysis.** We first intersected the human and mouse protein datasets and used 6,261 proteins that were observed across the shared timepoints within a cell line, that is, primed ESCs, early and late gastruloids. We normalized each protein's abundance in a given replicate to its respective species geometric mean and log$_2$-transformed values for subsequent analysis, unless otherwise stated. To construct our correlation network, we first calculated the Pearson correlation coefficients ($r_{Pearson}$) across all 19,596,930 possible pairs of proteins. As we had already calculated $r_{Pearson}$ across all possible pairs of proteins, we permuted sample labels across our dataset to generate the null distribution of correlation coefficients. Given the relatively lower number of timepoints sampled and the strong bimodal distribution of Pearson correlation coefficients, we stringently filtered the network edges with BH-adjusted $P < 0.01$ and absolute $r_{Pearson} \geq 0.95$. This step filtered the network down to

489,417 (301,561 correlated and 187,856 anticorrelated) pairs, but was strongly enriched for protein–protein interactions, macromolecular complexes and biochemical pathways, and was used for subsequent network analysis.

**Edge annotation in the correlation network.** We considered seven major annotations as literature evidence for any given edge: (1) protein–protein interaction, (2) belonging to the same protein complex or (3) biochemical pathway, (4) GO biological process, (5) GO molecular function, (6) GO cellular component or (7) subcellular location. Protein complex annotations were obtained from CORUM[65] (downloaded 12 September 2022) and ComplexPortal[66] (downloaded 7 January 2024). Annotated gene sets for pathways[59–63] and GO[45] were downloaded from the Molecular Signatures Database[151]. Protein localization annotations were curated from the Human Protein Atlas[30,149]. Networks were illustrated using the igraph R package or Cytoscape[152].

**Bioinformatic identification of cooperative protein interactions.** We searched all nodes in our correlation network against known complexes and pathways that consisted of at least three subunits. We adapted a previously described approach[64] and used Fisher's exact test to compute statistical enrichment of cooperative complexes with established modules. For each protein complex or pathway module, we tested its neighbouring proteins (first-degree edges) for significant association with a particular module, and termed those 'cooperative proteins'. For each protein tested, we first counted the number of edges that it shared with the established module, then we counted the number of edges that linked the module to other proteins (excluding the candidate protein) in the network. We next counted the number of edges the candidate protein had to the rest of the correlation network (that is, excluding the module of interest). Finally, we counted the number of edges that were not associated with the candidate protein nor the module of interest. These edge counts were used to compute statistical significance using Fisher's exact test. We independently repeated this test for all 6,261 proteins against 1,357 known protein complexes and select metabolic pathways. The $P$ values obtained were adjusted for multiple hypothesis testing using the BH procedure, and only cooperative proteins with adjusted $P$ values of <0.05 were considered significant.

**Comparison of RNA and protein abundance analysis.** Global RNA–protein correlations were calculated using all nine observations of transcripts and proteins across mouse and human gastruloid development. To ensure stringent analysis, we filtered for genes detected in both species for downstream analysis. Pseudocounts of 1 were added to filtered count matrices and were converted to transcripts per million. Mean transcript and protein abundances were converted to $\log_2$(fold change) ratios to their respective species geometric mean. For every gene, we calculated the per-gene RNA–protein correlation ($r_{Pearson}$) using a vector of abundances across nine samples. GO-term enrichment of biological processes in correlated and anticorrelated genes was performed using ClusterProfiler[153]. We intersected the 6,010 genes detected across both datasets with the Human Protein Atlas[30] for subcellular locations, CORUM[65] and ComplexPortal[66] for protein complexes, and KEGG for biochemical pathways[61]. To measure the extent of correlation of transcripts and RNAs within the mouse timepoints, we calculated the ratio of protein to RNA mean fold changes across each timepoint. In summary, a discordance of 0 implied that the protein and RNAs were highly correlated, and discordance less than 0 implied that the RNAs were more abundant than protein levels and vice versa. Discordance scores for protein complexes were calculated by taking the median protein–RNA correlation across constituent members. To prevent averaging pairs of proteins, we only considered complexes where more than two proteins were detected in our data. Transcriptional signatures of stage-specific mouse TFs were detected as follows. First, we calculated the Pearson correlation comparing TF protein abundances to all observed transcripts. We subset the resulting correlation matrix to identify protein–transcript pairs with high correlation ($r_{Pearson} \geq 0.9$) and used TFLink[82] to select only transcripts that were annotated as targets of specific TFs. We confirmed that identified TF targets displayed similar temporal regulation to their upstream TF by comparing target transcript abundance at each stage to determine the maximum transcript abundance.

**Phosphoprotein and kinase analysis.** For differential expression testing and analysis, in every pairwise comparison, $\log_2$ ratios for all quantified phosphosites were calculated following subtraction of the $\log_2$ ratios of the corresponding proteins to identify protein-independent phosphorylation changes. Kinase–substrate pairs were curated from PhosphositePlus[89]. Human kinases were annotated using KinMap[154]. For kinase substrate prediction and enrichment analysis, for each phosphosite, we first calculated the $\log_2$(fold change) ratio to the row mean (across all samples), subtracted the corresponding protein $\log_2$(fold change) ratios, and used that as input into the KSEA app[88] with a minimum substrate cutoff of ≥2 to calculate $z$-scores for the kinases. Kinase–substrate pairs with absolute $r_{Pearson} \geq 0.5$ were visualized as a network using Cytoscape[152].

**Statistics and reproducibility.** No statistical methods were used to predetermine sample sizes, but our sample sizes are similar to those reported in previous work[42]. No experimental data were excluded from the analyses. Sequencing and spectrometry data exclusion criteria are outlined in the Methods, including filtering out the substandard reads and spectra, following general practices in genomics and proteomics. Human RA-gastruloids and mouse conventional gastruloids used in the experiments were randomly selected from each timepoint before sample preparation. The investigators were not blinded to allocation during experiments and outcome assessment.

### Use of AI-based tools
We disclose that manuscript refinement and proofreading were supported by the AI-based tools Claude (Opus 4.6 and Sonnet 4.6) and ChatGPT (GPT-4o and GPT-4.5). AI-based tools were not used for conceptual development, initial manuscript drafting or building figures.

### Reporting Summary
Further information on research design is available in the Nature Portfolio Reporting Summary linked to this Article.

## Data availability
Sequencing data that support the findings of this study have been deposited in the Gene Expression Omnibus (GEO) under accession code GSE273813. Proteomics datasets have been deposited and are available at the ProteomeXchange Consortium under accession code PXD054460. Other data supporting the findings of this study are available from the corresponding author on reasonable request. Source data are provided with this paper.

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

## Acknowledgements

We thank D. Calderon, C. Qiu, J.-B. Lalanne, A. Keith and S. Fayer at the University of Washington, as well as the rest of the members of the Shendure and Starita labs, in particular for critical insights, discussions and feedback. We thank M. Yang, C. Xu, V. Browning, E. Nichols and K. Partington for assistance, reagents and advice related to microscopy and imaging. We also thank A. Rajaraman and K. Drew (University of Illinois Chicago) for advice and feedback related to network analyses and mapping of protein complexes. R.K.G. acknowledges support from a Washington Research Foundation postdoctoral fellowship. D.K.S. acknowledges support from the NIH/NIGMS (R35GM150919), Washington Research Foundation, the W. M. Keck Foundation, an Andy Hill CARE Distinguished Researcher Award, a Cancer Consortium New Investigator Award, and the Pew Charitable Trusts. R.K.G., S.C., S.B. and L.M.S. were supported by the National Human Genome Research Institute (NHGRI; 1RM1HG010461). J. Shendure is an Investigator of the Howard Hughes Medical Institute and acknowledges support from the Paul G. Allen Frontiers Group (Allen Discovery Center for Cell Lineage Tracing) and the Brotman Baty Institute for Precision Medicine.

## Author contributions

R.K.G. and N.H., in consultation with D.K.S., conceived the study. R.K.G. and N.H. performed stem cell and gastruloid experiments with assistance from Z.L., C.K. and A.W. R.K.G. and N.H. performed the transcriptomics experiments with assistance from S.C. and S.B. V.L., R.F. and C.D.M. performed the proteomics and phosphoproteomics experiments. R.K.G. computationally analysed the data with support from N.H., M.S., D.K.S. and J. Shendure. R.K.G., N.H., V.L., D.K.S., L.M.S. and J. Shendure wrote the manuscript. J. Stone and R.K.G. built the web interface. D.K.S., N.H., L.M.S. and J. Shendure oversaw the experiments and data analyses.

## Competing interests

J. Shendure is on the scientific advisory board, a consultant and/or a co-founder of Prime Medicine, Guardant Health, Camp4 Therapeutics, Phase Genomics, Adaptive Biotechnologies, Sixth Street Capital, Pacific Biosciences, Cellular Intelligence and 10x Genomics. D.K.S. is a consultant and/or collaborator with ThermoFisher Scientific, AI Proteins, Genentech and Matchpoint Therapeutics. The remaining authors declare no competing interests.

## Additional information

**Extended data** is available for this paper at https://doi.org/10.1038/s41556-026-01937-5.

**Correspondence and requests for materials** should be addressed to Riddhiman K. Garge, Jay Shendure, Lea M. Starita, Nobuhiko Hamazaki or Devin K. Schweppe.

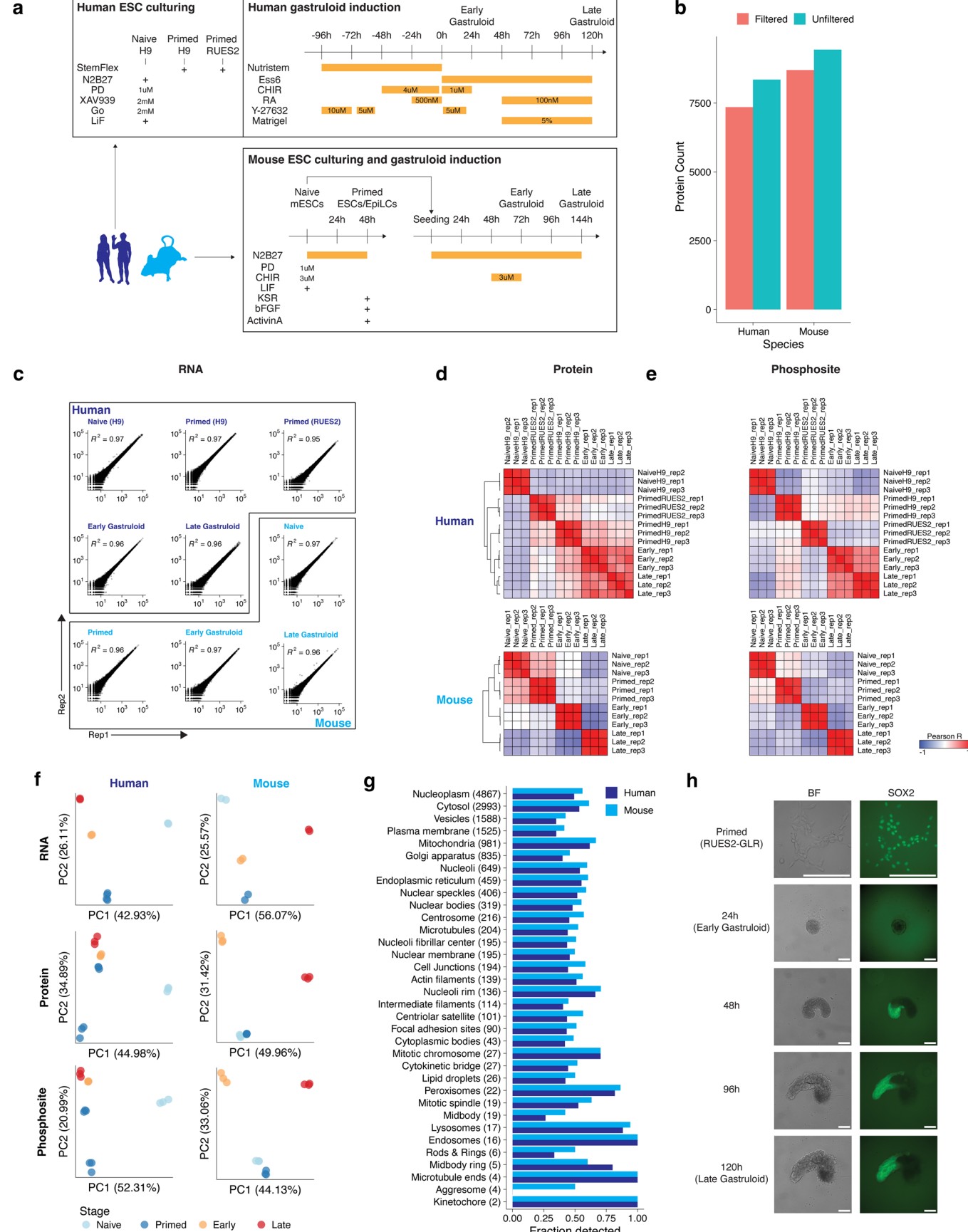

**Extended Data Fig. 1 | See next page for caption.**

**Extended Data Fig. 1 | Mapping the dynamics of gastruloid development using multi-omics. (a)** Timeline and conditions for human and mouse ESC culturing and gastruloid induction. **(b)** Total numbers of proteins quantified across all human or mouse samples. Protein identifications were filtered to a 1% FDR and required summed TMTpro reporter ion signal-to-noise ratios >100 for quantitation. **(c)** Scatterplots comparing the RNA counts between biological replicates for each sample. **(d,e)** All-by-all sample similarity matrices of pairwise Pearson correlation coefficients ($r_{Pearson}$) calculated from summed protein **(d)** or phosphosite intensities **(e)** across human (top) or mouse (bottom) samples. Biological replicates were highly correlated across each data type (RNA: $r > 0.98$; protein: $r > 0.93$; phosphosite: $r > 0.97$). **(f)** PCA plots of PC1 vs. PC2 using RNA (top), protein (middle) or phosphosite (bottom) data across human (left) or mouse (right) samples. **(g)** Fraction of proteins assigned to each of 34 subcellular localizations by the Human Protein Atlas that were successfully detected here. Numbers within brackets indicate the total numbers of proteins within each class shown. **(h)** Representative images highlighting the morphology (left) and the SOX2-mCitrine expression across the stages of gastruloid development. Scale bar: 200 μm. The experiments were independently reproduced five times with similar results.

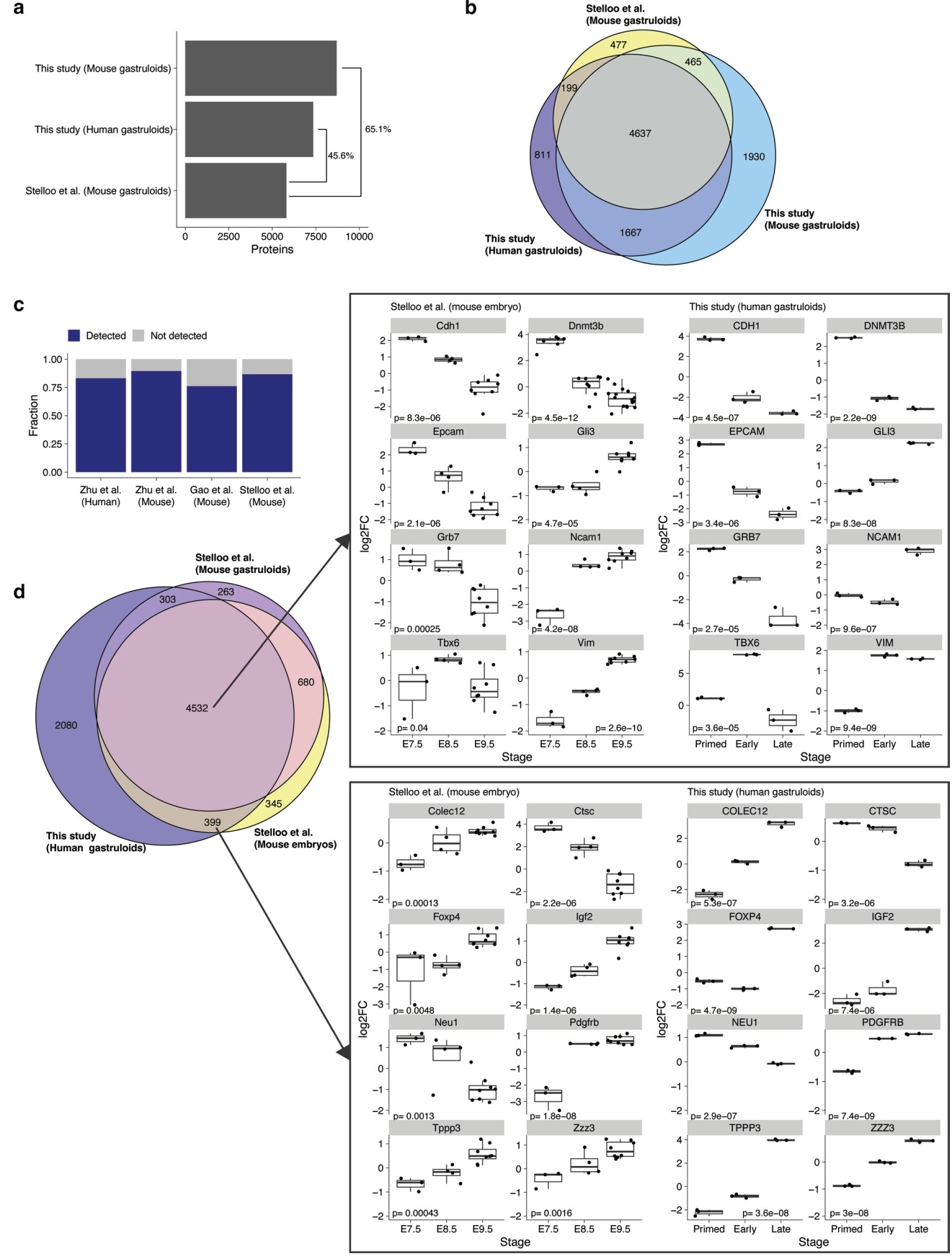

**Extended Data Fig. 2 | See next page for caption.**

**Extended Data Fig. 2 | Quantitative proteomics of human gastruloids expands protein coverage and recapitulates temporal trends observed during gastruloid differentiation. (a)** Barplot of the number of proteins quantified in human and mouse gastruloids [this study] vs. mouse gastruloids [Stelloo et al.[42]]. **(b)** Venn diagram showing intersection of proteins detected in human gastruloids [this study], mouse gastruloids [this study], and mouse gastruloids [Stelloo et al.[42]]. **(c)** Proportion of the 7,352 proteins detected in human gastruloids [this study] also detected in published human and mouse embryo proteomics datasets[42-44]. **(d)** Left: Venn diagram showing intersection of proteins detected in human gastruloids [this study], mouse gastruloids [Stelloo et al.[42]], and mouse embryos [Stelloo et al.[42]]. Right: Comparison of temporal trends of selected proteins in mouse embryos (E7.5, E8.5, and E9.5; n >=3 for each stage) [Stelloo et al.[42]] vs. human gastruloid timepoints (primed, early, and late; n = 3 for each stage) [this study]. Significance computed through one-way ANOVA.

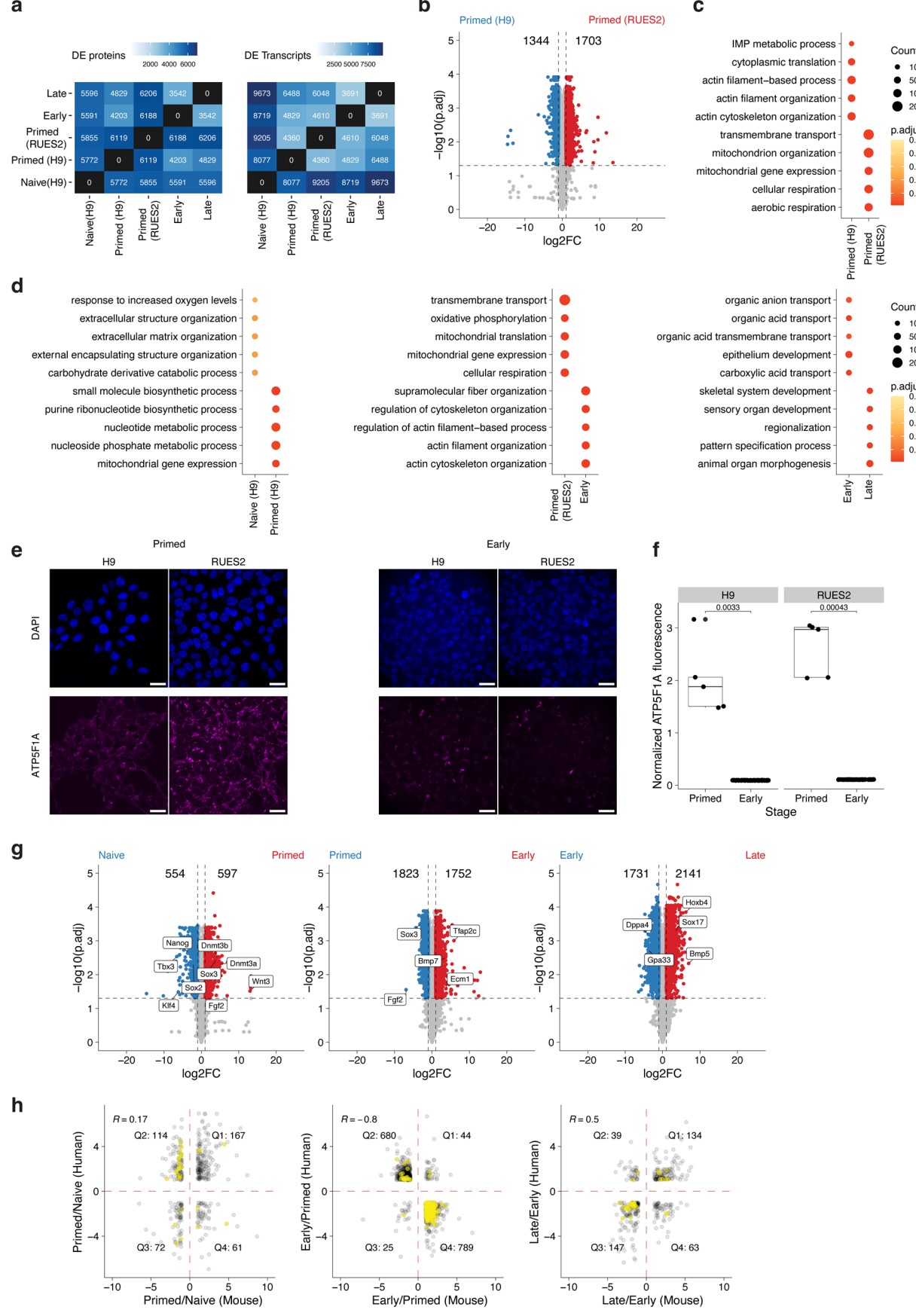

**Extended Data Fig. 3 | See next page for caption.**

**Extended Data Fig. 3 | Mapping differentially expressed biological processes across gastruloid development. (a)** Heatmaps indicating the number of differentially expressed proteins (left) and transcripts (right) between pairs of human samples. Differentially abundant proteins (DAPs) and differentially expressed transcripts (DETs) determined by those having absolute $\log_2$ fold change >=1 and BH-adjusted p-value < 0.05. **(b)** Volcano plot depicting the DAPs between primed H9 vs. primed RUES2-GLR ESCs where x-axis represents the log2 fold change between two adjacent timepoints and y-axis represents the negative log10 of the Benjamini–Hochberg-adjusted p-value (correcting for multiple hypothesis testing). Significance determined using the two-sided standard t-test. **(c)** Dot plot indicating the GO terms enriched in DAPs between primed H9 vs. primed RUES2-GLR ESCs. **(d)** Dot plots indicating the GO terms enriched in DAPs between adjacent stages of human samples. Color scales for dot plots indicate the BH-adjusted p-value and sizes of dots indicate the number of genes detected within each term. Significance determined using a one-sided hypergeometric test. **(e)** Representative ATP5F1A fluorescence images of RUES2 and H9 primed ESCs (left) and early gastruloids (right). Blue channel indicates DAPI, Scale bar: 25 μm. **(f)** Boxplots of normalized ATP5F1A immunofluorescence intensity. Significance determined using two-sided standard t-test. (n = 5 for primed ESCs and n = 60 for early gastruloid timepoints). Boxplots show the median (centre line), 25th–75th percentiles (box), 1.5x the interquartile range (line; end points signify maxima and minima). **(g)** Volcano plots depicting the DAPs between adjacent stages of mouse samples, where x-axis represents the $\log_2$ fold change between two adjacent timepoints and y-axis represents the negative $\log_{10}$ of the BH-adjusted p-value (correcting for multiple hypothesis testing). Significance determined using the two-sided standard t-test. **(h)** Scatter plots comparing mouse and human proteomes across adjacent stages. Comparisons were filtered to proteins with an absolute $\log_2$ fold change >=1 across both species. Mitochondrial proteins highlighted in yellow.

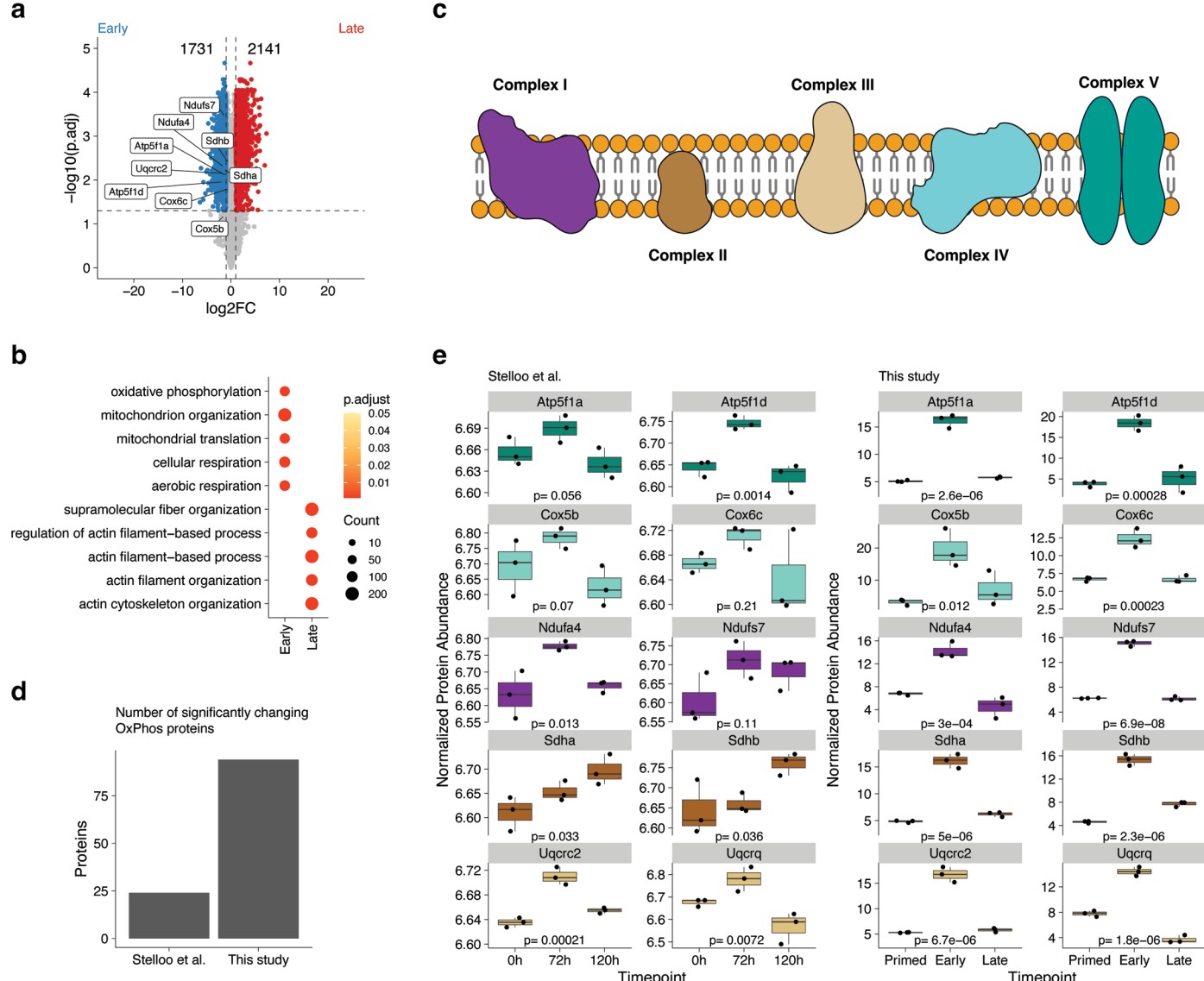

**Extended Data Fig. 4 | Temporal protein profiling recapitulates systematic downregulation of OxPhos over mouse gastruloid differentiation. (a)** Volcano plots depicting the DAPs between early (blue) and late (red) mouse gastruloids, where x-axis represents the $\log_2$ fold change between the two timepoints and y-axis represents the negative $\log_{10}$ of the BH-adjusted p-value (correcting for multiple hypothesis testing). Significance determined using the two-sided standard t-test. Labels indicate OxPhos subunits. **(b)** Dot plots indicating the GO terms enriched in DAPs between early and late mouse gastruloids. Color scales for dot plots indicate the BH-adjusted p-value and sizes of dots indicate the number of genes detected within each term. Significance determined using a one-sided hypergeometric test. **(c)** Schematic of OxPhos complexes. **(d)** Number of significantly changing OxPhos proteins in this study and Stelloo et al.[42]. **(e)** Comparison of the temporal dynamics of OxPhos proteins in mouse gastruloid development between this study (left) and Stelloo et al.[42] (right); n = 3 biological replicates per timepoint. Significance determined using one-way ANOVA.

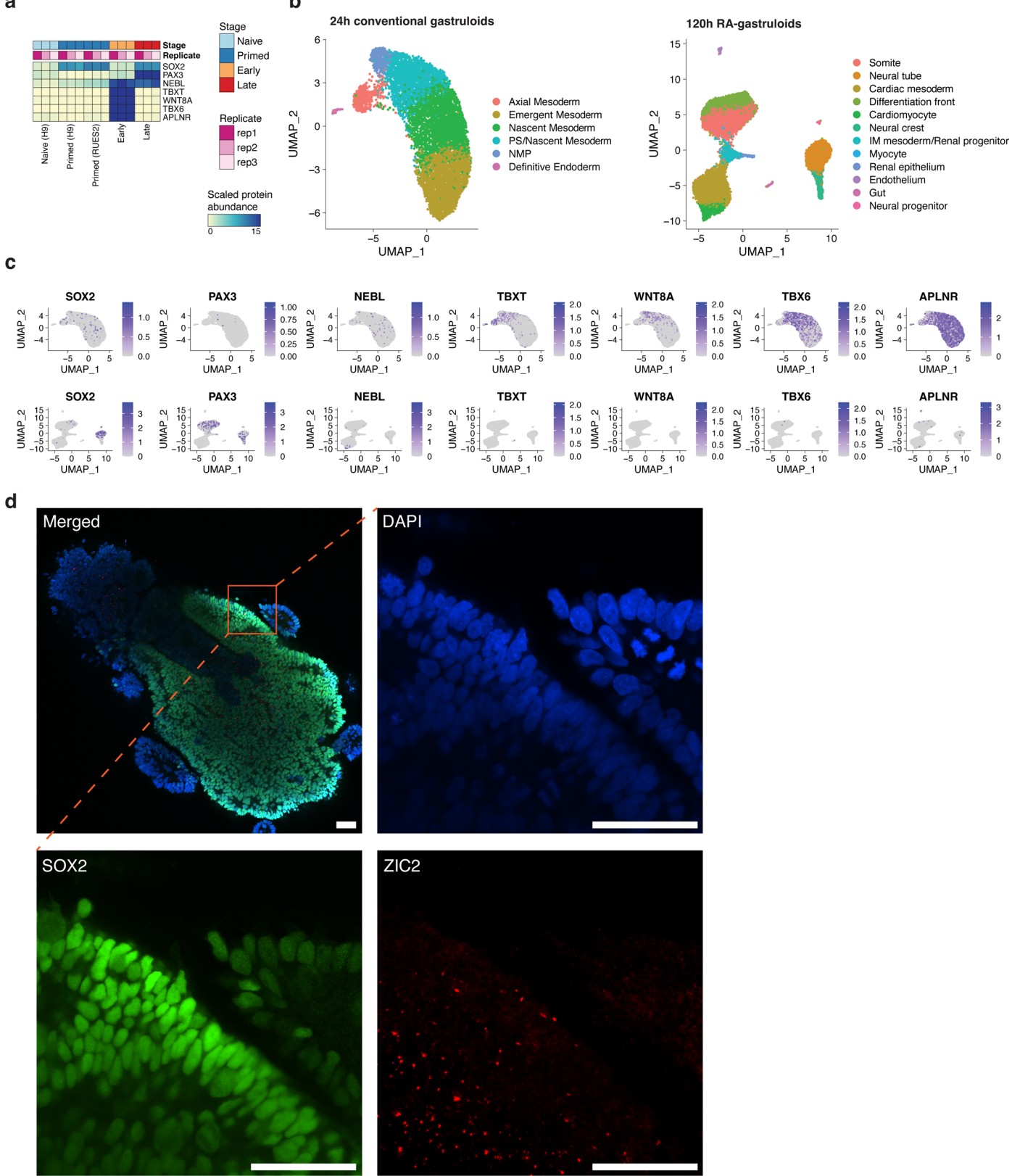

**Extended Data Fig. 5 | Mapping cell types contributing to bulk proteomic observations. (a)** Heatmap depicting the temporal profiles of SOX2, PAX3, NEBL, TBXT, WNT8A, TBX6, and APLNR. Color scale for protein data indicates scaled TMTpro reporter ion abundance. **(b)** UMAP projection of scRNA-seq profiles from 24-hour (Early) and 120-hour (Late) time points of human gastruloids. Colors in each cell type indicate the cell type **(c)** Normalized expression of SOX2, PAX3, NEBL, TBXT, WNT8A, TBX6, and APLNR from human gastruloids at 24 (top row) and 120 (bottom row) hours of development. **(d)** Immunostaining of 120 hour RA-gastruloids reveals coexpression of SOX2 (green) and ZIC2 (red) in neural tube cells. Blue channel indicates DAPI stain. Scale bar 50 μm.(n = 3/3 gastruloids displayed similar results).

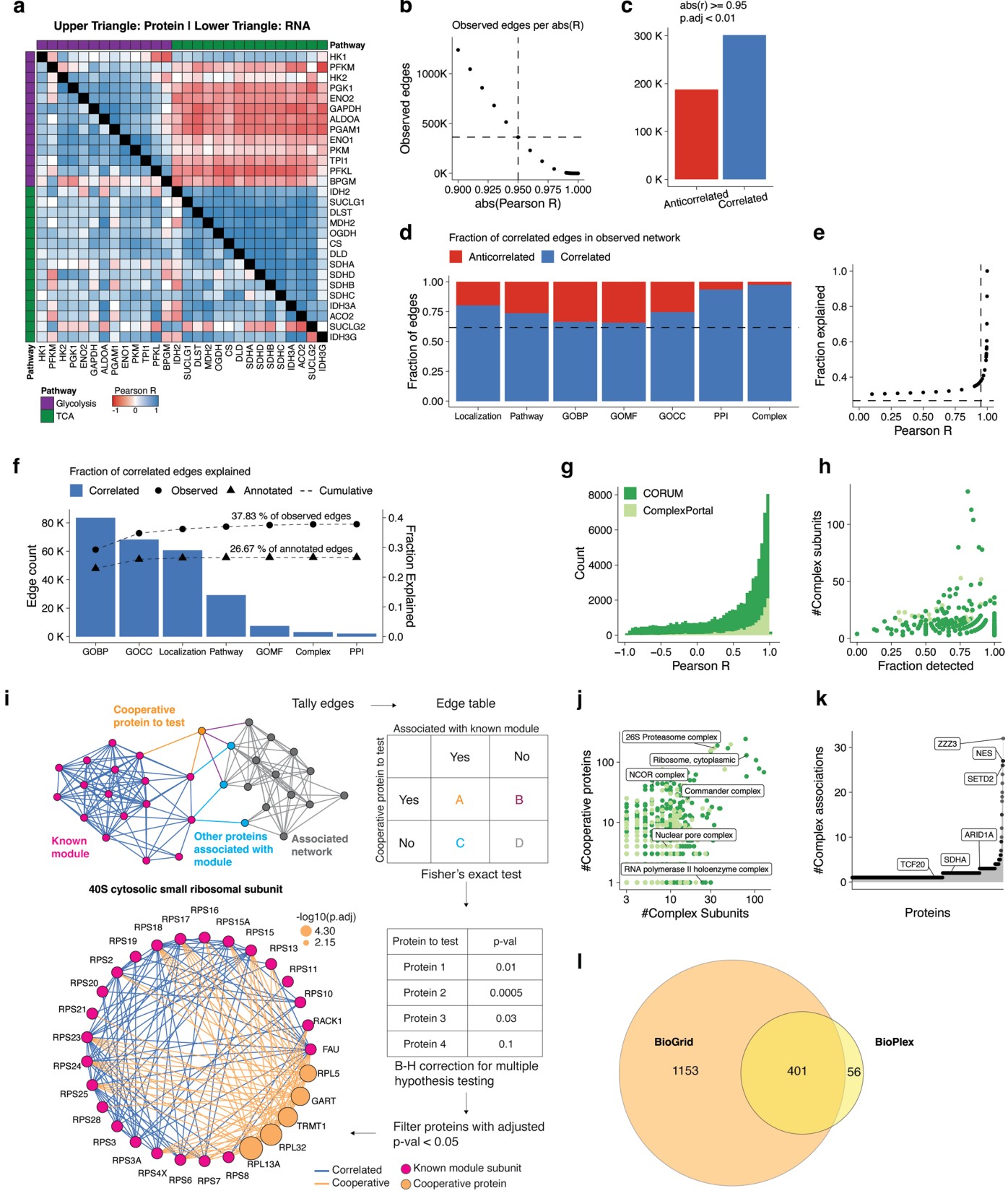

**Extended Data Fig. 6 | See next page for caption.**

**Extended Data Fig. 6 | Mapping pairwise protein co-regulation onto known protein modules identifies cooperative protein associations across gastruloid development. (a)** Heatmap displaying pairwise Pearson correlation coefficients among glycolysis (purple) and TCA cycle (green) genes. The upper triangle represents protein-level correlations and the lower triangle represents RNA-level correlations. Black cells denote self-comparisons along the diagonal. **(b)** Number of observed edges (y-axis) in the correlation network as a function of absolute $r_{Pearson}$ (x-axis). **(c)** Summary of correlated and anticorrelated edges in the network. **(d)** Fraction of correlated and anticorrelated edges stratified by GOBP, GOCC, Localization, Pathway, GOMF, BioPlex PPI, and Complex databases. Dashed line indicates the fraction of positively correlated edges in the trimmed network. **(e)** Fraction of correlated edges in the trimmed network explained by at least one database (y-axis) as a function of $r_{Pearson}$ (x-axis). Horizontal and vertical dashed lines respectively indicate the fraction of edges explained in annotated network and $r_{Pearson}$ at 0.95. **(f)** Summary of correlated edges in the trimmed network explained by shared membership in a Gene Ontology biological process (GOBP), cellular component (GOCC), molecular function (GOMF), localization, pathway, protein-protein interaction (BioPlex) or protein complex. Dotted line indicates the cumulative number of edges explained within the observed (circles) and annotated (triangle) networks. **(g)** Distribution of $r_{Pearson}$ for protein pairs in CORUM and ComplexPortal complexes. **(h)** Fraction of complexes detected in correlation network (x-axis) versus size of protein complex (y-axis). Dots colored by database used to curate the protein complexes. **(i)** Workflow to map cooperative proteins associated with detected modules. **(j)** Number of cooperative proteins detected (y-axis) as a function of complex size (x-axis). **(k)** The number of annotated ComplexPortal complexes that were found to be cooperative with each individual protein in the correlation analysis (x-axis), *for example* ZZZ3 was assigned as a cooperative protein to 32 ComplexPortal protein complexes. **(l)** Venn diagram indicating the number of cooperative proteins with physical protein-protein interaction evidence to at least one subunit of their associated complex.

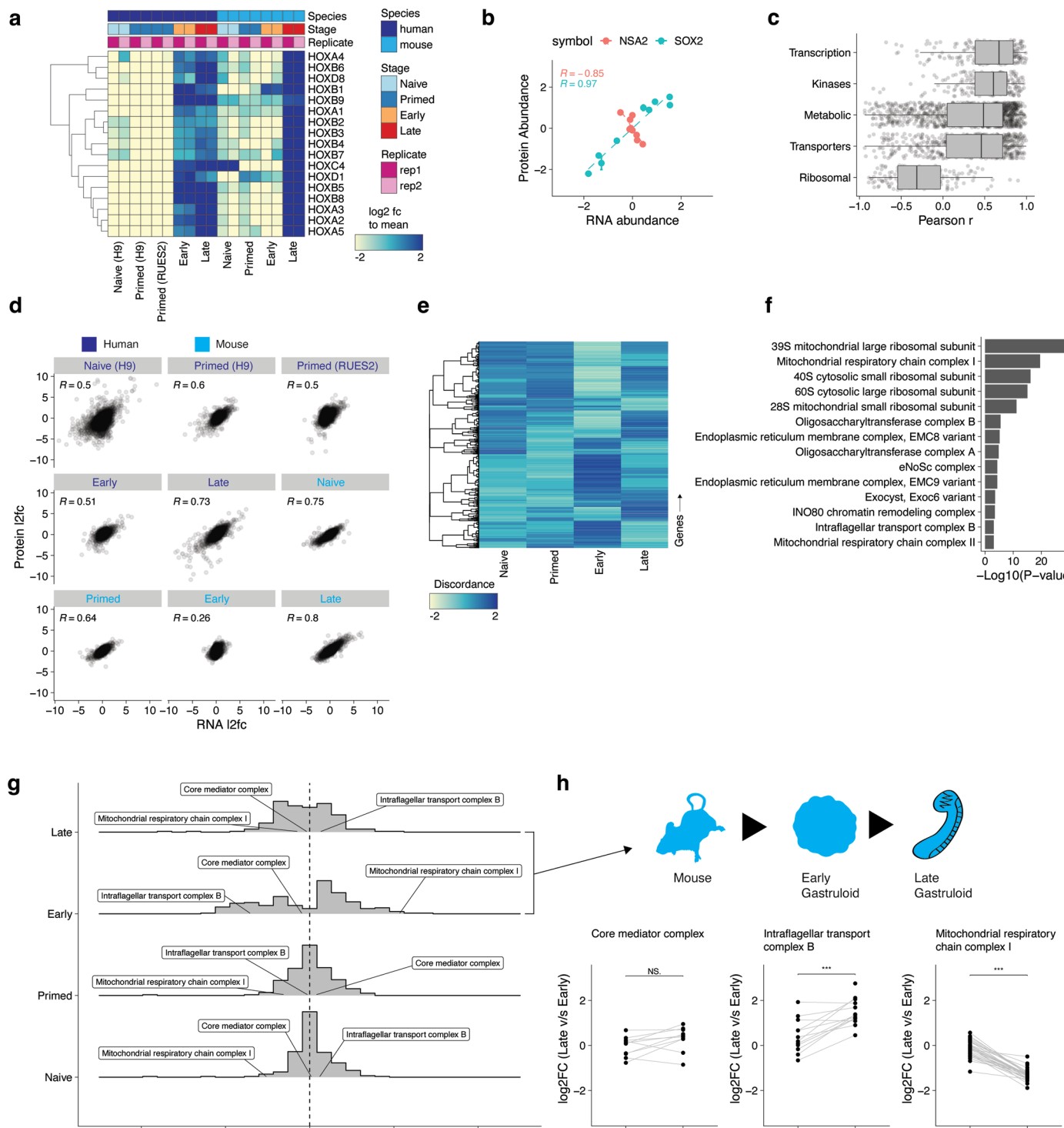

**Extended Data Fig. 7 | Identifying patterns of RNA–protein discordance across gastruloid development. (a)** The temporal dynamics of the HOX gene expression cluster. Rows indicate genes, while columns signify samples. Color scale represents the log2 fold change of transcripts normalized to each sample's respective species mean. **(b)** Representative examples of RNA vs. protein abundance correlation for SOX2 (red) and LAMTOR2 (teal). **(c)** Distributions of RNA-protein correlations ($r_{Pearson}$) for 6,010 genes grouped by protein class (curated from Human Protein Atlas). **(d)** Scatterplot of RNA (x-axis) and protein (y-axis) abundance across stages of mouse or human gastruloid development. **(e)** Hierarchical clustering of patterns of RNA-protein discordance ratios across

genes during mouse gastruloid development. **(f)** Protein complexes whose RNA abundances differ significantly from their protein abundances when comparing early vs. late mouse gastruloids. Significance determined two-sided standard t-test. **(g)** Median RNA-protein discordances of members of protein complexes at each stage of mouse gastruloid development. **(h)** Comparison of the RNA and protein $\log_2$-scaled fold changes between early vs. late mouse gastruloids in the Mediator complex (left), intraflagellar transport complex B (middle), and mitochondrial Complex I of the oxidative phosphorylation pathway. Significance testing on RNA and protein distributions was performed using a two-sided standard t-test (NS. denotes not significant; *** denotes p < 0.001).

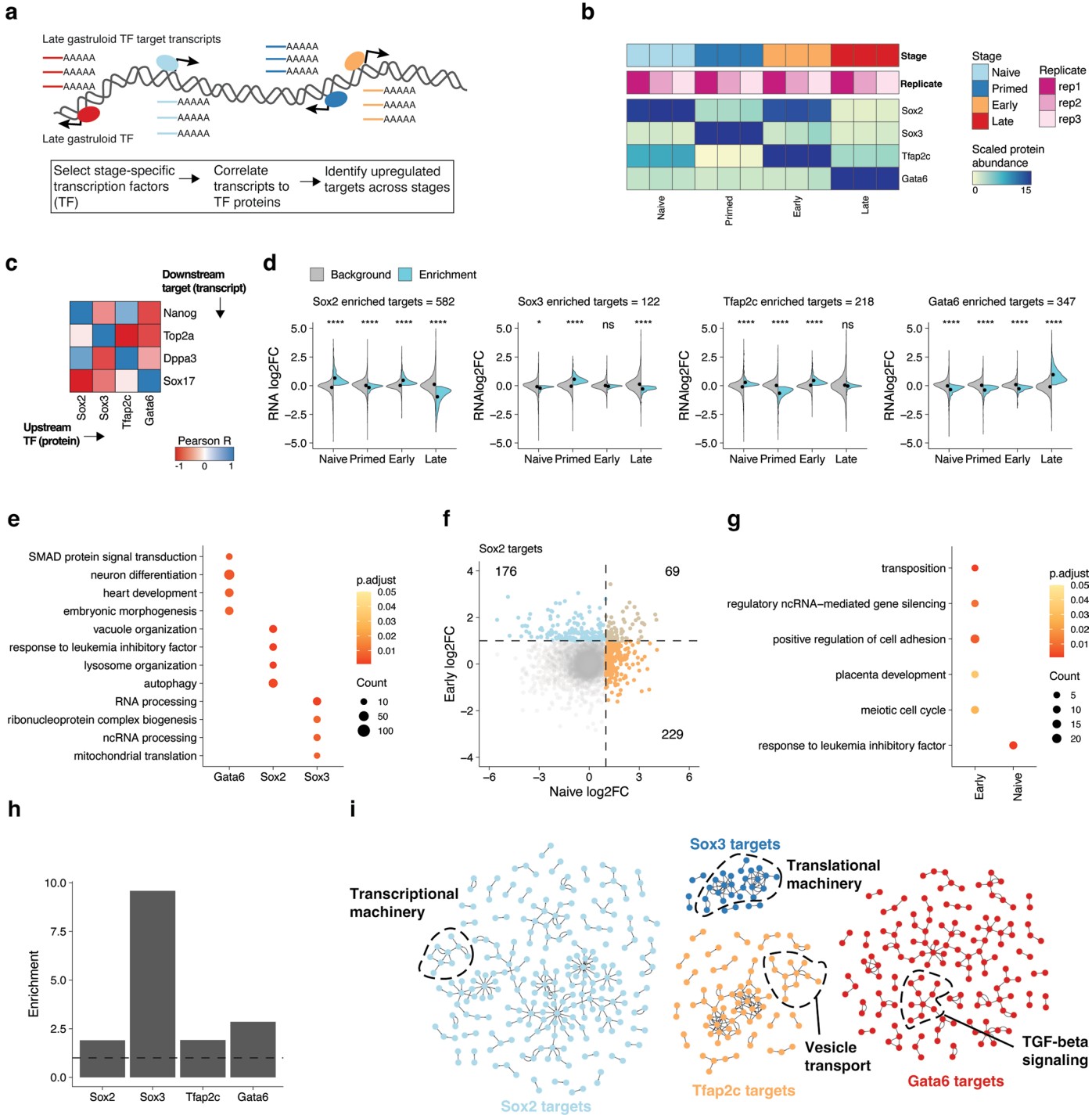

**Extended Data Fig. 8 | Sample-matched temporal multi-omic profiling of mouse gastruloid differentiation reveals stage-specific upregulation and biological commonalities of downstream transcription factor targets.**
(a) Workflow for identifying putative downstream targets of stage-specific transcription factors. (b) Stage-specific protein expression of Sox2, Sox3, Tfap2c and Gata6. (c) Representative heatmap depicting the $r_{Pearson}$ correlation coefficients of transcription factor protein abundance (columns) to downstream target transcripts (rows). (d) RNA abundance distributions (y-axes) of target transcripts to aforementioned transcription factors (top). Colors indicate the enriched (cyan) or background (gray) target transcripts to the corresponding transcription factor. Significance estimated using ANOVA (n.s. denotes not significant; * denotes $p < 0.05$; **** denotes $p < 1.3e-8$). (e) Dotplot highlighting the biological processes significantly enriched in downstream targets of Gata6, Sox2, and Sox3. Color scale indicates the p-value adjusted for multiple hypothesis testing using the Benjamini–Hochberg procedure and sizes of dots indicate the number of genes detected within each term. Significance calculated using a one-sided hypergeometric test. (f) Scatterplot comparing levels of downstream Sox2 targets in naïve stage ESCs and early-stage gastruloids. Naïve and early-stage enriched targets colored in orange and blue respectively while brown points indicate enriched Sox2 targets upregulated in both. (g) Dotplot highlighting the biological processes significantly enriched in downstream targets of Sox2. Color scale indicates the p-value adjusted for multiple hypothesis testing using the Benjamini–Hochberg procedure and sizes of dots indicate the number of genes detected within each term. Significance calculated using a one-sided hypergeometric test. (h) Enrichment of BioPlex protein-protein interactions across Sox2, Sox3, Tfap2c, and Gata6. Dotted line indicates the background rate (i) Network representation of protein-protein interactions in the enriched targets of Sox2, Sox3, Tfap2c, and Gata6.

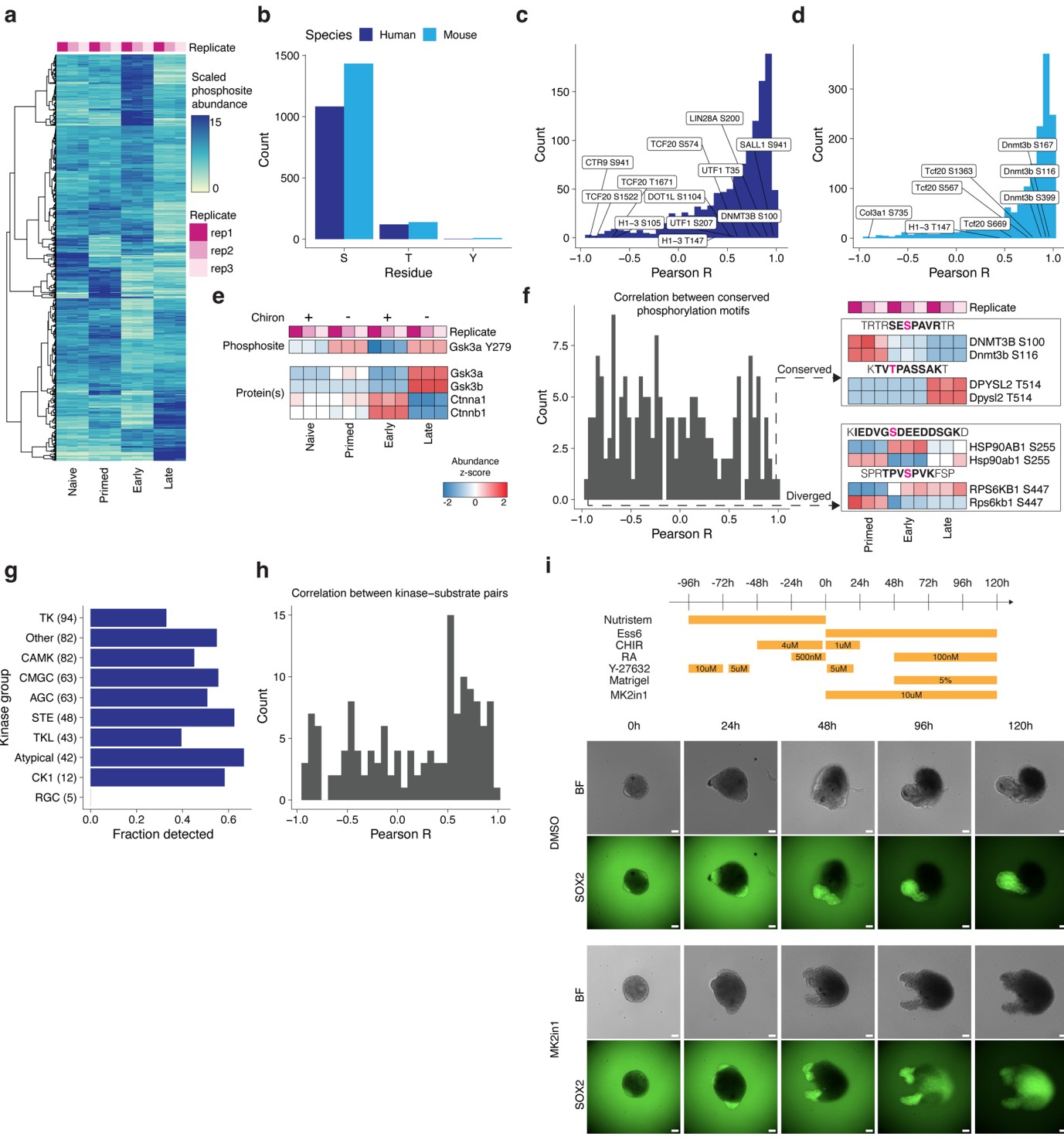

**Extended Data Fig. 9 | Mapping phosphorylation states across gastruloid development.** (a) The temporal dynamics of phosphorylated peptides across mouse gastruloid development. (b) Number of phosphorylated sites (y-axis) identified per amino-acid residue (x-axis). (c,d) Distribution of Pearson correlation coefficients ($r_{Pearson}$) from comparing the abundances of phosphosites to their respective (c) human or (d) mouse proteins. (e) Effects of temporal Chiron treatment on protein and/or phosphorylation dynamics of Gsk3a, Gsk3b, Ctnna1, and Ctnnb1. (f) Distribution of $r_{Pearson}$ computed from comparing temporal abundances of conserved phosphorylation motifs between human and mouse (left). Representative tile plots of conserved and diverged phosphosite profiles

across motifs shared between humans and mice (right). Detected peptide (bold) and phosphorylated residue (magenta) are highlighted above each tile plot. (g) Proportion of human protein kinases detected by kinase group. Kinase annotations curated from KinMap explorer[154]. (h) Histogram of $r_{Pearson}$ between human kinase–substrate pairs detected across human gastruloid development. Pairs curated from PhosphositePlus. (i) Protocol and timecourse of RA-gastruloid development when treated with DMSO (top row) and MAPKAPK2 inhibitor, MK2in1 (bottom row). Fluorescence images indicate the expression of SOX2-mCitrine. Ess6, Essential 6 media; CHIR9, CHIR99021; RA, Retinoic Acid. Scale bar: 200 μm. The experiments were independently reproduced five times with similar results.

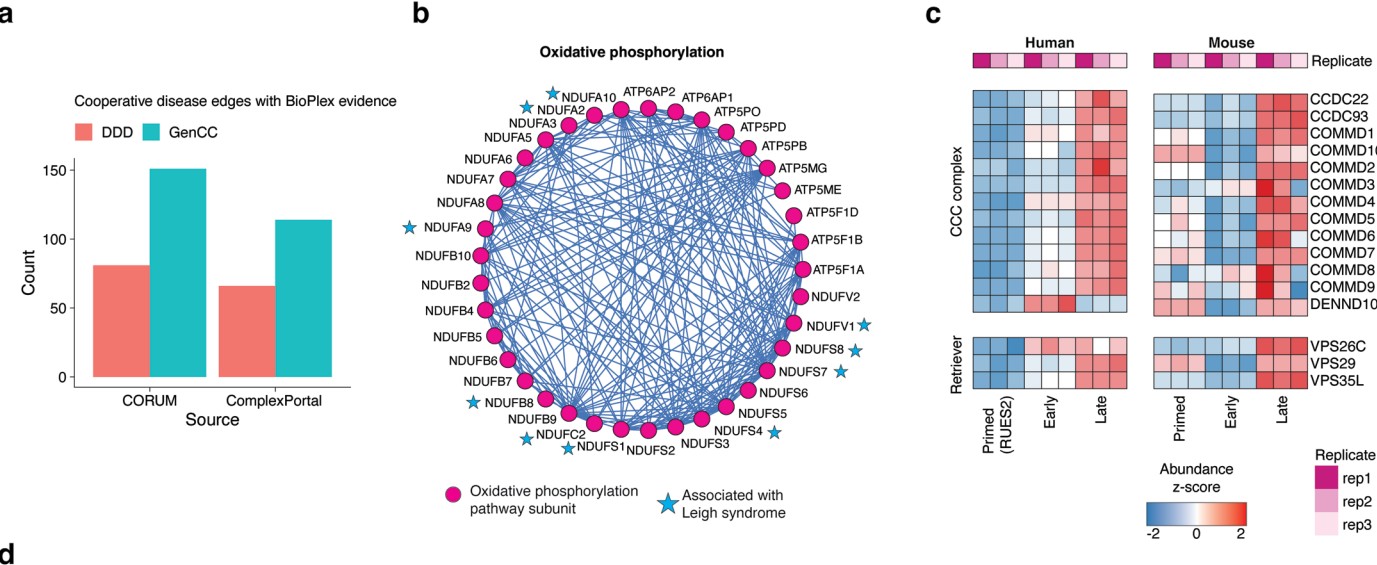

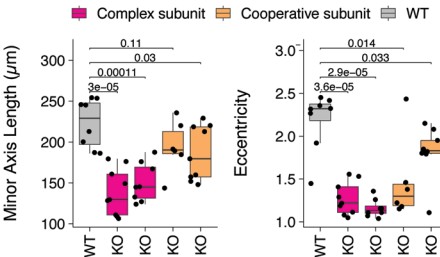

**Extended Data Fig. 10 | Mining co-regulatory protein networks to nominate disease gene candidates. (a)** Number of cooperative disease proteins with physical evidence in BioPlex or BioGrid to known protein complexes**. (b)** Oxidative phosphorylation co-regulatory network (pathway associations curated from WikiPathways). Proteins associated with Leigh syndrome (blue stars) were enriched in the oxidative phosphorylation co-regulation network (Pathway curated from WikiPathways). **(c)** Temporal protein profiles of the

Commander complex across human and mouse gastruloid development. **(d)** Boxplots comparing the distributions of minor axis length (left) and major-to-minor axis ratio (Eccentricity, right) of wild-type and perturbed gastruloids (n >= 8 for each genetic knockout). Significance determined using two-sided standard t-test. Boxplots show the median (centre line), 25th–75th percentiles (box), 1.5x the interquartile range (line; end points signify maxima and minima). Significance determined using two-sided standard t-test.

Jay Shendure
Lea Starita
Nobuhiko Hamazaki
Devin Schweppe

# Reporting Summary

## Statistics

For all statistical analyses, confirm that the following items are present in the figure legend, table legend, main text, or Methods section.

| n/a | Confirmed | |
|---|---|---|
| ☐ | ☒ | The exact sample size (*n*) for each experimental group/condition, given as a discrete number and unit of measurement |
| ☐ | ☒ | A statement on whether measurements were taken from distinct samples or whether the same sample was measured repeatedly |
| ☐ | ☒ | The statistical test(s) used AND whether they are one- or two-sided<br>*Only common tests should be described solely by name; describe more complex techniques in the Methods section.* |
| ☒ | ☐ | A description of all covariates tested |
| ☐ | ☒ | A description of any assumptions or corrections, such as tests of normality and adjustment for multiple comparisons |
| ☐ | ☒ | A full description of the statistical parameters including central tendency (e.g. means) or other basic estimates (e.g. regression coefficient) AND variation (e.g. standard deviation) or associated estimates of uncertainty (e.g. confidence intervals) |
| ☐ | ☒ | For null hypothesis testing, the test statistic (e.g. *F*, *t*, *r*) with confidence intervals, effect sizes, degrees of freedom and *P* value noted<br>*Give P values as exact values whenever suitable.* |
| ☒ | ☐ | For Bayesian analysis, information on the choice of priors and Markov chain Monte Carlo settings |
| ☒ | ☐ | For hierarchical and complex designs, identification of the appropriate level for tests and full reporting of outcomes |
| ☐ | ☒ | Estimates of effect sizes (e.g. Cohen's *d*, Pearson's *r*), indicating how they were calculated |

*Our web collection on statistics for biologists contains articles on many of the points above.*

## Software and code

Policy information about availability of computer code

| | |
|---|---|
| Data collection | Basecall files were converted to fastq formats using bcl2fastq (Illumina) and demultiplexed on the i5 and i7 indexes. Adapter trimming was performed using Trimmomatic v0.39. Depending on the species, trimmed reads were then aligned using STAR to either the human GRCh38 or mouse GRCm39 reference assemblies. Count matrices were then generated with bam files using FeatureCounts.For proteomics data, raw files were searched against the relevant annotated proteome from Uniprot. Comet search algorithm was utilized to match peptides to spectra. Peptide-spectrum matches (PSMs) were filtered to a 1% false discovery rate (FDR). Proteins were filtered to an FDR of 1%. For quantitation, PSMs were required to have a summed TMT reporter ion signal-to-noise ≥100.For further detail, please see the Methods section. |
| Data analysis | The code used to analyze the data, and generate figures are deposited in the following GitHub repository- https://github.com/bbi-lab/Temporal-Gastrulomics<br>The following freely available R packages were used for data analysis:<br>Tidyverse- https://cran.r-project.org/web/packages/tidyverse/index.html<br>Scales- https://cran.r-project.org/web/packages/scales/index.html<br>ggrepel- https://cran.r-project.org/web/packages/ggrepel/index.html<br>ggrastr- https://cran.r-project.org/web/packages/ggrastr/index.html<br>cowplot- https://cran.r-project.org/web/packages/cowplot/index.html<br>RColorBrewer- https://cran.r-project.org/web/packages/RColorBrewer/index.html<br>ggplot2- https://cran.r-project.org/web/packages/ggplot2/index.html<br>ggridges- https://cran.r-project.org/web/packages/ggridges/index.html<br>ggpmisc- https://cran.r-project.org/web/packages/ggpmisc/index.html<br>GGally- https://cran.r-project.org/web/packages/GGally/index.html |

```
psych- https://cran.r-project.org/web/packages/psych/index.html
ggpubr- https://cran.r-project.org/web/packages/ggpubr/index.html
reshape2- https://cran.r-project.org/web/packages/reshape2/index.html
umap- https://cran.r-project.org/web/packages/umap/index.html
Seurat- https://cran.r-project.org/web/packages/Seurat/index.html
EnvStats- https://cran.r-project.org/web/packages/EnvStats/index.html
pheatmap- https://cran.r-project.org/web/packages/pheatmap/index.html
dendsort- https://cran.r-project.org/web/packages/dendsort/index.html
amap- https://cran.r-project.org/web/packages/amap/index.html
Matrix- https://cran.r-project.org/web/packages/Matrix/index.html
igraph- https://cran.r-project.org/web/packages/igraph/index.html
DeSeq2- https://bioconductor.org/packages/release/bioc/html/DESeq2.html
conflicted- https://cran.r-project.org/web/packages/conflicted/index.html
```

For manuscripts utilizing custom algorithms or software that are central to the research but not yet described in published literature, software must be made available to editors and reviewers. We strongly encourage code deposition in a community repository (e.g. GitHub). See the Nature Portfolio guidelines for submitting code & software for further information.

# Data

Policy information about availability of data

All manuscripts must include a data availability statement. This statement should provide the following information, where applicable:
- Accession codes, unique identifiers, or web links for publicly available datasets
- A description of any restrictions on data availability
- For clinical datasets or third party data, please ensure that the statement adheres to our policy

DATA AVAILABILITY
RNA-seq data have been deposited to the Gene Expression Omnibus (GEO) database with the identifier GSE273813. Mass spectrometry proteomics data have been deposited to the ProteomeXchange Consortium via the PRIDE partner repository with the dataset identifier PXD054460. Reviewers can access these data through PRIDE using the account details: Username: reviewer_pxd054460@ebi.ac.uk; Password: 4lQJ5v6pqvGs.

CODE AVAILABILITY
All supporting scripts and code have been deposited onto the following repository at https://github.com/bbi-lab/Temporal-Gastrulomics. All processed data are available through the web application at: https://gastruloid.brotmanbaty.org/.

No experimental data were excluded from the analyses. Sequencing and spectrometry data exclusion criteria is outlined in the Methods, including filtering out the substandard reads and spectra, following general practices in genomics and proteomics.

# Research involving human participants, their data, or biological material

Policy information about studies with human participants or human data. See also policy information about sex, gender (identity/presentation), and sexual orientation and race, ethnicity and racism.

| | |
|---|---|
| Reporting on sex and gender | N/A |
| Reporting on race, ethnicity, or other socially relevant groupings | N/A |
| Population characteristics | N/A |
| Recruitment | N/A |
| Ethics oversight | N/A |

Note that full information on the approval of the study protocol must also be provided in the manuscript.

# Field-specific reporting

Please select the one below that is the best fit for your research. If you are not sure, read the appropriate sections before making your selection.

☒ Life sciences ☐ Behavioural & social sciences ☐ Ecological, evolutionary & environmental sciences

For a reference copy of the document with all sections, see nature.com/documents/nr-reporting-summary-flat.pdf

# Life sciences study design

All studies must disclose on these points even when the disclosure is negative.

| | |
|---|---|
| Sample size | No statistical methods were used to pre-determine sample sizes but the sample sizes used in this study are comparable to those reported in previous publications (please see main text for references) |
| Data exclusions | Data exclusion criteria for transcriptomic and proteomic assays and datasets are outlined in the methods section, including the filtering metrics, all of which are in line with general practices in the field. |
| Replication | All gastruloid samples were collected within the same experimental batch to avoid confounding issues from batch effects. Samples profiled using proteomics were cultured in triplicate while samples profiled using transcriptomics were cultured in duplicate. All other experiments were repeated in triplicate. |
| Randomization | Human RA-gastruloids and mouse conventional gastruloids were randomly selected within each timepoint before multi-omics phenotyping. |
| Blinding | The investigators were not blinded to allocation during experiment and outcome assessment. |

# Reporting for specific materials, systems and methods

We require information from authors about some types of materials, experimental systems and methods used in many studies. Here, indicate whether each material, system or method listed is relevant to your study. If you are not sure if a list item applies to your research, read the appropriate section before selecting a response.

### Materials & experimental systems

| n/a | Involved in the study |
|---|---|
| ☐ | ☒ Antibodies |
| ☐ | ☒ Eukaryotic cell lines |
| ☒ | ☐ Palaeontology and archaeology |
| ☒ | ☐ Animals and other organisms |
| ☒ | ☐ Clinical data |
| ☒ | ☐ Dual use research of concern |
| ☒ | ☐ Plants |

### Methods

| n/a | Involved in the study |
|---|---|
| ☒ | ☐ ChIP-seq |
| ☒ | ☐ Flow cytometry |
| ☒ | ☐ MRI-based neuroimaging |

## Antibodies

| | |
|---|---|
| Antibodies used | Phospho-Histone H2A.X (Ser139) (20E3) Rabbit Monoclonal Antibody 5763. Dilution- 1:50<br>Anti-ATP5A antibody ab14748. Dilution- 1:1000<br>Anti-SOX2 Antibody AB5603. Dilution- 1:100<br>Donkey anti-Mouse IgG (H+L) Highly Cross-Adsorbed Secondary Antibody, Alexa Fluor™ 647 A-31571. Dilution- 1:500<br>Donkey anti-Rabbit IgG (H+L) Highly Cross-Adsorbed Secondary Antibody, Alexa Fluor™ Plus 488 A32790. Dilution- 1:500<br>Donkey anti-Rabbit IgG (H+L) Highly Cross-Adsorbed Secondary Antibody, Alexa Fluor™ Plus 555 A32794. Dilution- 1:500 |
| Validation | Donkey anti-Mouse IgG (H+L) Highly Cross-Adsorbed Secondary Antibody, Alexa Fluor™ 647 A-31571 Validated in HeLa cells by manufacturer https://www.thermofisher.com/antibody/product/Donkey-anti-Mouse-IgG-H-L-Highly-Cross-Adsorbed-Secondary-Antibody-Polyclonal/A-31571<br>Donkey anti-Rabbit IgG (H+L) Highly Cross-Adsorbed Secondary Antibody, Alexa Fluor™ Plus 488 A32790 Validated in HEK293 cells by manufacturer https://www.thermofisher.com/antibody/product/A32790.html<br>Donkey anti-Rabbit IgG (H+L) Highly Cross-Adsorbed Secondary Antibody, Alexa Fluor™ Plus 555 A32794 Validated in A549 cells by manufacturer https://www.thermofisher.com/antibody/product/Donkey-anti-Rabbit-IgG-H-L-Highly-Cross-Adsorbed-Secondary-Antibody-Polyclonal/A32794 |

## Eukaryotic cell lines

Policy information about cell lines and Sex and Gender in Research

| | |
|---|---|
| Cell line source(s) | The RUES2-GLR line was provided by Dr. Ali H. Brivanlou (Rockefeller University). Naive and primed H9 ESCs were kindly provided by Dr. Austin Smith (University of Exeter). Mouse ESC line E14Tg2a was obtained from Dr. Christian Schroeter (Max Planck Institute). |
| Authentication | Activities of three markers SOX2, TBXT, and SOX17 (tagged with mCitrine, mCerulean, and tdTomato respectively) were monitored using fluorescence microscopy. |
| Mycoplasma contamination | Cell lines used in this study were not tested for Mycoplasma contamination |

| Commonly misidentified lines<br>(See ICLAC register) | No commonly misidentified cell lines were used. |

## Plants

| Seed stocks | N/A |
| Novel plant genotypes | N/A |
| Authentication | N/A |

