## [Peer Review File · Nature Cell Biology]

The proteomic landscape and temporal dynamics of human and mouse gastruloid development

Corresponding Author: Dr Devin Schweppe

Version 0:

Decision Letter:

Dear Dr Schweppe,

I am writing on behalf of my colleague Stelios Lefkopoulos, who is currently out of the office.

Your manuscript "The proteomic landscape and temporal dynamics of mammalian gastruloid development", has now been seen by 3 referees, who are experts in gastruloids (referee 1); embryonic development and embryo models (referee 2); and proteomics and genomics (referee 3), and whose comments are pasted below. In light of their advice, we regret that we cannot offer to publish the study in Nature Cell Biology.

As you will see, although the reviewers find this work interesting, they raise serious concerns that question the conceptual advance that these findings represent over previous work, and the strength of the data and of the novel conclusions that can be drawn at this stage.

Although we cannot publish your paper, it may be appropriate for another journal in the Nature Portfolio. If you wish to explore the journals and transfer your manuscript please use our manuscript transfer portal. You will not have to re-supply manuscript metadata and files, unless you wish to make modifications. For more information, please see our http://www.nature.com/authors/author_resources/transfer_manuscripts.html?WT.mc_id=EMI_NPG_1511_AUTHORTRANSF&WT.ec_id=AUTHOR manuscript transfer FAQ page.

We are very sorry that we could not be more positive on this occasion, but we thank you for the opportunity to consider this work.

With kind regards,
Angela Parrish

Angela R Parrish, PhD
Locum Senior Editor
Nature Cell Biology

Reviewers' comments:

Reviewer #1 (Remarks to the Author):

In this manuscript, Garge et al. conduct a dual transcriptomics and (phospho)proteomics time course study in both human and mouse stem cell-based embryo models, from naïve through gastrulation stages. Using two primed human embryonic stem cell lines (H9 and RUES2-GLR), the authors also address differences in the proteomic state of different human ESC lines, finding the RUES2-GLR to be more enriched for proteins involved in respiration, including oxidative phosphorylation. Through co-regulation analysis, the authors recapitulate established protein-protein interactions and then begin to assign unknown proteins to complexes. The authors then ask whether transcript expression profiles can be predictive of protein abundance across different species and stages, finding that correlations (and anti-correlations) align with specific biological processes. Interestingly, the authors then uncover that developmental transcription factor expression profiles correlates with only a very small number of total targets. Extending the investigation to phosphoproteomics, the authors were able to capture expected protein phosphorylation states at specific developmental stages, as well as potentially new kinase-substrate regulation e.g. MAPKAPK2 mediated phosphorylation of ZFP36L leading to NANOG degradation. Finally, focusing on different neurological syndromes, the authors recapitulate disease associated protein complexes whilst identifying potential additional complexes members.

Overall, the work is conducted to a high standard and this significant study provides a powerful reference dataset for the community, from early through to late gastrulation stage mammalian development. The text is clearly written and the data presented in a simple and accessible manner. My overall recommendation is “to publish” provided the following comments/concerns are addressed by the authors

1. While the manuscript is being evaluated as a “resource” article, it will definitely be strengthened if the authors emphasised the conceptual advances that were made through this work. Specifically there are two interesting results that authors should discuss more/elaborate on better.

1.1 One of the striking observations that the authors make is that there is surprisingly little discordance at the protein level between human and mouse gastruloids in this study, particularly in the WNT and MAPK signalling pathways and metabolic pathways. This is interesting because the mouse and human gastruloid protocols used have a 10-fold difference in starting cells number, ultimately generating gastruloids of substantially different sizes. Recent works (<https://doi.org/10.1242/dev.204870>) indicate that starting cell number is a major factor in the developmental trajectory of a gastruloid (doi.org/10.1101/2024.12.12.628151, doi.org/10.1101/2024.12.23.630037). Could the authors comment on this observation, or provide further analysis to investigate this further?

1.2 The finding that TF abundance correlates with only a small percentage of the total TF target abundance is very interesting. For the small number of targets that do correlate with a given TF - is it possible to identify any commonality between them? Are they in protein complexes, or associated complexes?

2. While there is a generally high standard of validation through known transcript-protein expression profiles, could the authors comment on the low (<50% for mouse samples) unique mapping rate in the RNA-seq data.

3. The authors mention that “over human gastruloid development, proteins in the TCA cycle tend to be upregulated in primed ESCs as compared to late gastruloids, and late gastruloids display elevated levels of glycolytic proteins.” Can they discuss these findings in light of recent works on glycolytic activity in mouse gastruloids and embryos (DOI: [10.1016/j.stem.2025.04.005](https://doi.org/10.1016/j.stem.2025.04.005) and DOI: [10.1038/s41586-024-08044-1](https://doi.org/10.1038/s41586-024-08044-1)).

4. This study shows that the primed RUES2-GLR cells show an enrichment for oxidative phosphorylation components but this is specifically downregulated in early gastruloid stages. Can the authors rule out that this is not a result of the changing aggregate size, where larger gastruloids limit the diffusion of metabolites at the core (related to point 1.1 above). To address this, could the authors show (a) generate human gastruloids from H9 cells (which do not show the same enrichment for OxPhos proteins) and assess potential downregulation of oxidative phosphorylation between primed and early gastruloid stages (e.g. Via immunofluorescence or western blot); and/or c) generate RUES2-GLR gastruloids from 400 rather than 4000 cells to test if the downregulation of certain metabolic proteins is still observed. (d) Also whether the same down regulation occurs in mouse gastruloids which are seeded from 10x fewer cells?

5. Could the authors speculate on why early gastruloid mouse samples specifically showed low RNA-protein expression profile correlation?

6. Along the lines of point 1.2 above, in Extended Data Fig. 5h, Sox2 displays two peaks in abundance, during both naïve and early gastruloid stages. Do the same Sox2 targets correlate with TF abundance at these developmental distinct stages?

Reviewer #1 (Remarks on code availability):

DOI or URL: <https://github.com/bbi-lab/Temporal-Gastrulomics>

I have not reviewed the code myself but have accessed the The authors Github page. The authors have done a terrific task at making the data and the codes readily accessible to the community

Reviewer #2 (Remarks to the Author):

In this study, Garge et al. profiled the proteomic and phosphoproteomic landscapes of mammalian gastruloid development. They generated proteomic and phosphoproteomic data and integrated them with publicly available scRNA-seq datasets to characterize the progression from naïve ESCs to late stage gastruloids in both human and mouse models. Their results highlight dynamic changes at both the protein and phosphorylation levels during gastruloid formation, reveal RNA-protein discordance, and identify conserved phosphorylation sites between species. However, given the limited novelty relative to recent publications and the insufficient insights currently offered by the study, I cannot support the publication of this manuscript in Nat Cell Biol.

Major issues:

1. Stelloo et al. (PMID: 38754429) recently profiled the gastrulation process in mouse embryos and gastruloids using multilayered proteomics. Their study combined bulk proteomics, phosphoproteomics, and single-cell proteomics, and included more detailed time points throughout gastrulation. Although the authors integrated transcriptomic data and included human gastruloids, these additions are incremental and do not offer substantial new insights. Overall, I do not find that this work surpasses the previous one either technically or conceptually.

2. The scRNA-seq data lack coverage of the naïve and primed stages, which limits the ability to perform comprehensive co-analysis of the transcriptome and proteome. As a result, only two gastruloid stages are available for comparison.

3. As a developmental stage during which the anteroposterior axis is established, bulk proteomic profiling may obscure critical spatial information related to body axis formation.

4. Although gastruloids are valuable in vitro models for studying mammalian gastrulation, this study relied exclusively on them to investigate normal embryonic development, without incorporating natural embryos as authentic references (at least in the case of mouse) or in vivo data to validate the findings.

5. The authors identified numerous phosphosites on key proteins that are dynamically regulated during gastruloid formation. It would be valuable to validate some functionally relevant phosphosites, as this could guide future investigations in the field.

Minors:

1. In extended data Fig 1d and 1e revealed a negative correlation between naïve and primed H9 ESCs, as well as a positive correlation between early and late samples in human proteomic and phosphoproteomic data. In contrast, mouse data showed a positive correlation between naïve and primed samples, while early and late samples exhibit a negative correlation. The author should conduct further analysis to elucidate this species difference.

2. In extended data Fig 2a, it is surprising to see that there are more DEPs between naïve and primed H9 ESCs, or even between the two primed ESC lines, than between naïve ESCs and late stage gastruloids. But this pattern is not observed in the mouse data. The authors should re-examine their results and attempt to validate these distinctions using transcriptomic data.

Reviewer #3 (Remarks to the Author):

Review of NCB-RS57819: "The proteomic landscape and temporal dynamics of mammalian gastruloid development" by Garge et al.

Summary

The manuscript by Schweppe and colleagues presents an interesting, ambitious and technically impressive multi-omic atlas of human and mouse gastruloid development, with matched profiling of transcriptome, proteome, and phosphoproteome dynamics across key developmental stages. While it lacks cell-type resolution at the proteome level, the study is valuable as a rich resource and offers important cross-species comparisons. In its current form, however, the authors over-interpret correlation-based findings and do not distinguish descriptive results from mechanistic conclusions. The authors have compiled a compelling resource of clear value to the field, but should tone down their claims to more accurately reflect the limitations of the methodology and the inference nature of the work. So while the manuscript has the potential to make a meaningful contribution to the developmental biology, systems biology and stem cell communities, it would benefit from a clearer framing and more restrained interpretation of bulk co-expression data.

Major points

1. The manuscript lacks a clearly articulated central motivation. It straddles resource generation, hypothesis generation, and mechanistic inference without prioritizing one. In its current form, it is best categorized as a comprehensive exploratory dataset rather than a mechanistically resolved study. The authors should clarify if the primary goal was to provide foundational datasets or to make new biological discoveries about gastrulation. This could help focus the narrative and sharpen the conclusions and impact.

2. Interpretation of RNA-protein discordance as evidence of widespread post-transcriptional regulation is overstated. The observed modest correlation is consistent with a large body of existing literature and could result from multiple factors, including protein stability and cellular heterogeneity. Without direct evidence of regulation (e.g. degradation rates, translational profiling), these claims should be tempered.

3. Co-regulation network analysis is a strength of the study, but the claims are overstated, and conclusions must be treated cautiously. Correlation does not establish functional interaction or complex formation. Many inferred associations are speculative and should be explicitly presented as hypotheses rather than established mechanisms, especially when links to disease phenotypes.

4. Cross-species comparison is potentially valuable, but the finding that human primed ESCs resemble early mouse gastruloids could be explainable by cell line- or culture-specific metabolic differences rather than conserved developmental timing. Regardless, this section would benefit from a more critical discussion of potential confounders and more cautious language around evolutionary interpretation.

5. Bulk proteomic profiling, while well executed and valuable, of heterogeneous gastruloids limits the ability to assign observed molecular dynamics to specific cell types. While scRNA-seq data are used for inference, these mappings are indirect. Without cell-type-specific proteomic resolution, conclusions about lineage-specific programs should be presented more tentatively.

Minor points

-Key terms such as "cooperative proteins" and "discordance" should be clearly defined.

-Statistical thresholds (e.g. Pearson $r > 0.95$ for co-reg) should be justified, and the sensitivity of results to such choices briefly touched on.

**For Nature Portfolio general information and news for authors, see <http://npg.nature.com/authors>.

Version 1:

Decision Letter:

6th February 2026

Dear Devin,

Thank you for submitting your revised manuscript "The proteomic landscape and temporal dynamics of mammalian gastruloid development" (NCB-RS57819A-Z). Please note that the original reviewer #1 was unavailable to re-review and we therefore asked reviewer #2 to comment on how the original concerns by reviewer #1 have been addressed in this revision. Your manuscript has therefore been seen by the original referees #2 and #3 and their comments are below. Both reviewers find the paper has improved in revision, but reviewer #2 continues to raise a few concerns. Having carefully considered the manuscript and all referee comments in this and the previous round of review, we have decided that their concerns should be addressed textually, but it will not be necessary to address them with new data and analyses to proceed at this journal. Therefore, we'll be happy in principle to publish your manuscript in Nature Cell Biology, pending minor revisions to textually satisfy the referee's final requests and to comply with our editorial and formatting guidelines.

If the current version of your manuscript is in a PDF format, please email us a copy of the file in an editable format (Microsoft Word or LaTeX)-- we cannot proceed with PDFs at this stage.

We are now performing detailed checks on your paper and will send you a checklist detailing our editorial and formatting requirements in about 10 days. Please do not upload the final materials and make any revisions until you receive this additional information from us.

Thank you again for your interest in Nature Cell Biology. Please do not hesitate to contact me if you have any questions.

Best wishes,
Stelios

Dr. Stylianos Lefkopoulos
He/him/his
Senior Editor, Nature Cell Biology
Springer Nature
Heidelberger Platz 3, 14197 Berlin, Germany

E-mail: stylianos.lefkopoulos@springernature.com
Bluesky: [@slefkopoulos.bsky.social](https://bsky.app/profile/@slefkopoulos.bsky.social)
LinkedIn: www.linkedin.com/in/stylianos-lefkopoulos-81b007a0

Reviewer #2 (Remarks to the Author):

In this revision, the authors have adequately addressed all questions raised by Reviewer 1. However, only a subset of my previous concerns has been satisfactorily resolved. Overall, I consider this study to be promising and potentially suitable for publication, but the remaining issues should be addressed. From my perspective, this work has several notable strengths: 1) a deep and comprehensive proteomic dataset of mouse gastruloid development; 2) the first proteomic dataset characterizing human gastruloid development; 3) the co-regulation network analysis identified genes associated with the Commander complex which are required for proper gastruloid development. Despite these strengths, several important issues remain unresolved. A central concern is how the presented datasets inform our understanding of natural gastrulation. Addressing the points outlined below will be necessary before I can recommend this manuscript for publication in Nat Cell Biol.

Major issues:

1. Although recently published mouse gastruloid proteomic datasets reduce, to some extent, the novelty and conceptual advance of the present study, I am convinced that the mouse dataset presented here is of substantially higher depth and quality than the existing report. Consequently, the human gastruloid dataset represents the primary source of novelty. The high concordance observed between the gastruloid data and recently published proteomic datasets from natural mouse gastrula embryos is encouraging and supports the validity and biological relevance of the model. However, despite the inclusion of human gastruloid data, it remains unclear how these data advance our understanding of gastrulation or change existing conceptual frameworks. While the authors emphasize that mouse and human gastrulation differ in many respects, the biological implications of these differences are not sufficiently developed. As a result, the human dataset, although potentially impactful, does not yet deliver a clear conceptual advance.

2. The value of cross-species analysis lies in the careful interpretation of interspecies differences. In the present study, however, the analytical framework predominantly emphasizes similarities across embryonic models, while observed differences are largely dismissed or implicitly attributed to artifacts of the model systems.

The human gastruloid model used here is derived from the mouse system, with the addition of RA and Matrigel to CHIR treatment to optimize patterning. These methodological choices introduce substantial similarities between the two models, while simultaneously complicating the interpretation of observed discrepancies. If the authors attribute these discrepancies primarily to model-specific features (e.g., RA or Matrigel), an alternative strategy would be to employ a human gastruloid model that more closely resembles the mouse system (e.g., Moris et al., 2020, Nature), thereby reducing confounding methodological variation.

More broadly, the authors should explicitly address the inherent challenges of interpreting interspecies differences using embryoid models. At a minimum, key species-specific findings should be validated in stage-matched natural mouse embryos and followed by a careful assessment of their biological significance.

3. Regarding RNA-protein discordance, as mentioned by the authors, numerous previous studies have reported varying degrees of concordance between mRNA and protein levels across diverse biological contexts. Therefore, the observation of RNA-protein discordance in this study is not, in itself, unexpected. The key issue is how such discordance provides new insight into gastrulation that was previously unappreciated. I encourage the authors to expand their discussion on this point.

Minors:

1. When comparing with the existing mouse proteomic dataset, I recommend that the authors use the most recent study (PMID: 39855199), which reports a substantially larger number of detected proteins than the dataset currently used in this manuscript.

2. In this revision, one major improvement is the authors' analysis of co-regulation patterns among cooperatively associated proteins. However, it remains unclear to what extent this analysis specifically benefits from proteomic measurements. In principle, a similar co-regulation network might be inferred from transcriptomic data alone. I therefore encourage the authors to clarify the added value of using proteomic data for this analysis and to explain whether and how the conclusions would differ from those derived from RNA-based datasets.

3. To validate the reported downregulation of mitochondrial activity, the authors assess ATP5F1A protein levels by immunofluorescence. However, it is unclear how the fluorescence signal was normalized, in particular which condition (primed or early) was used as the baseline for comparison.

Reviewer #3 (Remarks to the Author):

The authors have addressed all key issues, the manuscript is now suitable for publication.

Version 2:

Decision Letter:

Dear Devin,

I am pleased to inform you that your manuscript, "The proteomic landscape and temporal dynamics of human and mouse gastruloid development", has now been accepted for publication in Nature Cell Biology. Congratulations!

Please note that *Nature Cell Biology* is a Transformative Journal (TJ). Authors may publish their research with us through the traditional subscription access route or make their paper immediately open access through payment of an article-processing charge (APC). Authors will not be required to make a final decision about access to their article until it has been accepted. <https://www.springernature.com/gp/open-research/transformative-journals> Find out more about Transformative Journals

Authors may need to take specific actions to achieve compliance with funder and institutional open access mandates. If your research is supported by a funder that requires immediate open access (e.g. according to [Plan S principles](https://www.springernature.com/gp/open-science/plan-s-compliance) or the [NIH public access policy](https://www.springernature.com/gp/open-science/us-federal-agency-compliance)) then you should select the gold OA route, and we will direct you to the compliant route where possible. Because authors warrant under our subscription licensing terms that they haven't committed to licensing any version of their article under a licence inconsistent with the terms of our agreement – including the applicable embargo period – publication under the subscription model isn't suitable for authors whose funders require no embargo.

If you have not already done so, we strongly recommend that you upload the step-by-step protocols used in this manuscript to protocols.io (<https://protocols.io>), an open online resource that allows researchers to share their detailed experimental know-how. All uploaded protocols are made freely available and are assigned DOIs for ease of citation. Protocols and Nature Portfolio journal papers in which they are used can be linked to one another, and this link is clearly and prominently visible in the online versions of both. Authors who performed the specific experiments can act as primary authors for the Protocol as they will be best placed to share the methodology details, but the Corresponding Author of the present research paper should be included as one of the authors. By uploading your Protocols onto protocols.io, you are enabling researchers to more readily reproduce or adapt the methodology you use, as well as increasing the visibility of your protocols and papers. You can also establish a dedicated workspace to collect your lab Protocols. Further information can be found at <https://www.protocols.io/help/publish-articles>.

Nature Cell Biology encourages authors presenting evidence for cell, biological, molecular, and genetic interactions to consider communicating these findings using Biofactoid (<https://biofactoid.org/>). This tool helps users share a searchable representation of interactions (e.g. binding, gene expression, post-translational modification) between genes, gene products, or chemicals. Information added to Biofactoid, with author attribution, is shared on social media and public databases, such as Pathway Commons, where it can be discovered and analyzed in the context of a large and growing corpus of knowledge.

With kind regards,
Stelios

Dr. Stylianos Lefkopoulos
He/him/his
Senior Editor, Nature Cell Biology
Springer Nature
Heidelberger Platz 3, 14197 Berlin, Germany

E-mail: stylianos.lefkopoulos@springernature.com
Bluesky: [@slefkopoulos.bsky.social](https://bsky.app/profile/slefkopoulos.bsky.social)
LinkedIn: www.linkedin.com/in/stylianos-lefkopoulos-81b007a0

** Visit the Springer Nature Editorial and Publishing website at http://editorial-jobs.springernature.com?utm_source=ejp_NCB_email&utm_medium=ejp_NCB_email&utm_campaign=ejp_NCB for more information about our career opportunities. If you have any questions please click [here](mailto:editorial.publishing.jobs@springernature.com).

Response to Reviewers

We thank the three reviewers for their close reading and constructive feedback. We were encouraged by the overall tone of the reviews, particularly the comments of Reviewer #1, who described the work as “**conducted to a high standard**”, “**a powerful reference dataset for the community**”, “**clearly written**”, and “**simple and accessible**”, and Reviewer #3, who described the work as “**valuable as a rich resource**”, containing “**important cross-species comparisons**” and “**a compelling resource of clear value to the field**”.

However, all three Reviewers also raised important concerns, including with respect to novelty, description vs. mechanism, articulation of limitations, as well as many specific points. In the revision, we have sought to address these comments through new experiments, new analyses, and textual clarifications. These include:

1. New comparisons of our data with other publicly available proteomics datasets profiling mouse gastruloids¹, mouse embryos^{1,2} and human embryos³, that highlight the greater depth, coverage and complementary value of our datasets.
2. New analyses of our multi-omic datasets exploring the shared biological relationships among upregulated targets of four transcription factors (Sox2, Sox3, Tfp2c, Gata6).
3. New immunofluorescence experiments and accompanying analyses demonstrating the downregulation of the OxPhos subunit ATP5F1A during the transition from the primed ESC to early gastruloid stage.
4. New immunofluorescence experiments validating the temporal trends of H2AX S140 observed in our phosphoproteomics dataset.
5. New experiments testing the hypothesis of predicted MAPKAPK2 kinase activity in gastruloid differentiation by chemical perturbation.
6. New experiments demonstrating the phenotypic effects of genetic perturbations to subunits of the Commander co-regulatory network, highlighting new biology for proteins within disease modules.
7. New experiments confirming that bulk proteomic observations are capable of elucidating cell type-specific expression trends.
8. Substantial revisions to the text that incorporate the aforementioned experimental data and analyses, while also providing a clearer framing of the data’s value and limitations.

Below we provide a detailed point-by-point response, with reviewer comments replicated in full in **blue text**, together with our responses inline in **black text**.

Reviewer #1 (Remarks to the Author):

In this manuscript, Garge et al. conduct a dual transcriptomics and (phospho)proteomics time course study in both human and mouse stem cell-based embryo models, from naïve through gastrulation stages. Using two primed human embryonic stem cell lines (H9 and RUES2-GLR), the authors also address differences in the proteomic state of different human ESC lines, finding the RUES2-GLR to be more enriched for proteins involved in respiration, including oxidative phosphorylation. Through co-regulation analysis, the authors recapitulate established protein-protein interactions and then begin to assign unknown proteins to complexes. The authors then ask whether transcript expression profiles can be predictive of protein abundance across different species and stages, finding that correlations (and anti-correlations) align with specific biological processes. Interestingly,

the authors then uncover that developmental transcription factor expression profiles correlates with only a very small number of total targets. Extending the investigation to phosphoproteomics, the authors were able to capture expected protein phosphorylation states at specific developmental stages, as well as potentially new kinase-substrate regulation e.g. MAPKAPK2 mediated phosphorylation of ZFP36L leading to NANOG degradation. Finally, focusing on different neurological syndromes, the authors recapitulate disease associated protein complexes whilst identifying potential additional complexes members.

Overall, the work is conducted to a high standard and this significant study provides a powerful reference dataset for the community, from early through to late gastrulation stage mammalian development. The text is clearly written and the data presented in a simple and accessible manner. My overall recommendation is “to publish” provided the following comments/concerns are addressed by the authors

Thank you for this accurate summary and positive assessment of the manuscript. We have sought to address your comments and concerns as outlined below.

1. While the manuscript is being evaluated as a “resource” article, it will definitely be strengthened if the authors emphasised the conceptual advances that were made through this work. Specifically there are two interesting results that authors should discuss more/elaborate on better.

1.1 One of the striking observations that the authors make is that there is surprisingly little discordance at the protein level between human and mouse gastruloids in this study, particularly in the WNT and MAPK signalling pathways and metabolic pathways. This is interesting because the mouse and human gastruloid protocols used have a 10-fold difference in starting cells number, ultimately generating gastruloids of substantially different sizes. Recent works (<https://doi.org/10.1242/dev.204870>) indicate that starting cell number is a major factor in the developmental trajectory of a gastruloid (doi.org/10.1101/2024.12.12.628151, doi.org/10.1101/2024.12.23.630037). Could the authors comment on this observation, or provide further analysis to investigate this further?

While we have not profiled human gastruloids from different starting cell numbers as was done in the cited papers, we note that the markers of key signaling pathways (e.g. WNT and MAPK signaling) and key developmental TFs (e.g. POU5F1, NANOG, SOX2, CDX2, ZEB2) that show particularly strong mouse-human concordance are well established as fundamental drivers of gastrulation. Furthermore, we do observe species-specific or protocol-specific differences where expected. For example, primed RUES2-GLR cells displayed elevated mitochondrial protein abundances compared to both primed H9 cells and mouse primed ESCs, suggesting cell line-specific metabolic states (see **Fig. 2a-d**; **Extended Data Fig. 2d-f**). In contrast, the scaling of gastruloids (*i.e.* more cells → larger gastruloids) may simply reflect an aspect of the protocol (whether within species or between species) that does not substantially alter the proportions of proteins.

However, we fully acknowledge that our study design is limited. The different numbers of cells (400 vs 4,000) and differential timing of CHIR addition (48 hrs vs 96 hrs post-aggregation) for mouse vs. human represent confounding variables that prevent us from unequivocally distinguishing species-specific vs. protocol-specific effects. To fully address this, future studies would need to vary parameters within species (e.g. varying the starting numbers of cells) and/or revise the protocols to make them fully concordant.

We have updated the Discussion section to emphasize this interpretation and these limitations. Also, we have added citations to the work referenced by this comment regarding the impact of cell number on gastruloid trajectories. The relevant passages now read as follows:

“Of note, conservation of these processes is evident despite substantial differences between the human and mouse gastruloid protocols (*e.g.* number of starting cells, timing of induction, etc.), suggesting that they correspond to robustly conserved features.”

“Fifth, the absence of standardized mammalian gastruloid culturing techniques makes it difficult to unambiguously distinguish species-specific vs. protocol-specific variation. Future studies performing multi-omic profiling of gastruloids of the same species prepared under different conditions (*e.g.* varying starting cell number^{134,135}) may be necessary to shed light on which aspects of the gastruloid protome are robust vs. sensitive to protocol variation, while species comparisons would benefit from greater harmonization of mouse and human gastruloid protocols.”

1.2 The finding that TF abundance correlates with only a small percentage of the total TF target abundance is very interesting. For the small number of targets that do correlate with a given TF - is it possible to identify any commonality between them? Are they in protein complexes, or associated complexes?

Thanks for these suggestions. We performed further analyses and found that there are indeed commonalities with respect to biological modules and physical interactions for the correlated subsets. These analyses are described in an updated Results section, reproduced below together with new **panels e-i** of **Extended Data Figure 8**:

“Gata6 targets were enriched for biological processes associated with SMAD signal transduction, heart development and embryonic morphogenesis, Sox2 targets for lysosome organization, autophagy and response to LIF, and Sox3 targets for mitochondrial translation and RNA processing (**Extended Data Fig. 8e**). Given that Sox2 exhibited elevated levels in naïve ESCs and early stage gastruloids, we asked how discrete these sets were and if the same downstream targets were upregulated at both stages. Out of the 245 naïve-stage Sox2 targets and 298 early-stage Sox2 targets, 69 were enriched in both stages (**Extended Data Fig. 8f**). GO analyses on the discrete Sox2 targets revealed that naïve-stage targets were enriched for response to LIF, while early-stage targets were enriched for processes associated with cell adhesion, placenta development and meiosis (**Extended Data Fig. 8g**). Downstream targets of these 4 TFs were also enriched for protein-protein interactions (**Extended Data Fig. 8h,i**). Thus, the temporal relationship between TFs and transcripts suggests that among the large number of putative targets⁸³, these subsets of transcripts would be good candidates for additional study in the context of differentiating gastruloids.”

Extended Data Figure 8 (panels e-i only). Sample-matched temporal multi-omic profiling of mouse gastruloid differentiation reveals stage-specific upregulation and biological commonalities of downstream transcription factor targets. (e) Dotplot highlighting the biological processes significantly enriched in downstream targets of Gata6, Sox2, and Sox3. Color scale indicates the p-value adjusted using the Benjamini-Hochberg procedure and sizes of dots indicate the number of genes detected within each term. **(f)** Scatterplot comparing levels of downstream Sox2 targets in naïve stage ESCs and early stage gastruloids. Naïve and early stage enriched targets colored in orange and blue respectively while brown points indicate enriched Sox2 targets upregulated in both. **(g)** Dotplot highlighting the biological processes significantly enriched in downstream targets of Sox2. Color scale indicates the p-value adjusted using the Benjamini-Hochberg procedure and sizes of dots indicate the number of genes detected within each term. **(h)** Enrichment of BioPlex protein-protein interactions across Sox2, Sox3, Tfp2c, and Gata6. Dotted line indicates the background rate **(i)** Network representation of protein-protein interactions in the enriched targets of Sox2, Sox3, Tfp2c, and Gata6.

2. While there is a generally high standard of validation through known transcript-protein expression profiles, could the authors comment on the low (<50% for mouse samples) unique mapping rate in the RNA-seq data.

We believe that this arises from running FeatureCounts under default parameters. While collapsing the reads to gene based counts, FeatureCounts estimates higher multi-mapping rate from reads spanning multiple exons. We inspected the alignment statistics from STAR and observed much higher unique mapping rates ranging between 71-75% for mouse samples and 77%-90% for human samples. We apologize for confusion and have amended the Methods section to now read:

“Human samples had an average unique mapping rate of 85.1% while those of mouse samples were 73.1%. Finally, count matrices for each species were then generated with the bam files using FeatureCounts with default parameters.”

3. The authors mention that “over human gastruloid development, proteins in the TCA cycle tend to be upregulated in primed ESCs as compared to late gastruloids, and late gastruloids display elevated levels of

glycolytic proteins.”. Can they discuss these findings in light of recent works on glycolytic activity in mouse gastruloids and embryos (DOI: 10.1016/j.stem.2025.04.005 and DOI: 10.1038/s41586-024-08044-1).

We thank the reviewer for pointing us to these studies, which find that upregulation of glycolysis is associated with somitic and mesodermal cell populations while elevated levels of TCA cycle is associated with neural tube formation. Our findings are consistent with these works, a point that we make in the revised Discussion that also now cites them:

“However, the downregulation of oxidative phosphorylation was observed in gastruloids generated from either RUES2-GLR or H9 cells, suggesting shifts in metabolism underlie the formation of early stage gastruloids. The elevated levels of glycolysis in later stages of gastruloid development are in line with previous studies demonstrating that elevated levels of glycolysis underlie somite formation and occurring in human RA-gastruloids from 96-120 hours after induction^{122–124}. Future efforts profiling in neural or somite organoid models may shed light on how metabolic states shape or are shaped by their differentiation.”

4. This study shows that the primed RUES2-GLR cells show an enrichment for oxidative phosphorylation components but this is specifically downregulated in early gastruloid stages. Can the authors rule out that this is not a result of the changing aggregate size, where larger gastruloids limit the diffusion of metabolites at the core (related to point 1.1 above). To address this, could the authors show (a) generate human gastruloids from H9 cells (which do not show the same enrichment for OxPhos proteins) and assess potential downregulation of oxidative phosphorylation between primed and early gastruloid stages(e.g. Via immunofluorescence or western blot); and/or c) generate RUES2-GLR gastruloids from 400 rather than 4000 cells to test if the downregulation of certain metabolic proteins is still observed. (d) Also whether the same down regulation occurs in mouse gastruloids which are seeded from 10x fewer cells?

Thank you for raising these important points. Towards addressing this comment, we performed:

- 1) Immunofluorescence staining of OxPhos protein ATP5F1A, (reviewer suggestion (a))
- 2) New analyses comparing OxPhos proteomic changes in our human gastruloid dataset with our mouse gastruloid dataset (reviewer suggestion (d))

For #1 (reviewer suggestion (a)), we performed immunofluorescence staining on ATP5F1A in human gastruloids generated from H9 and RUES2 cells. We observe that gastruloids generated from both human cell line backgrounds undergo downregulation of oxidative phosphorylation as they transition from primed ESCs to 24 hour gastruloid stages. These new results are represented in the following updated section of the Results and new **panels e-f of Extended Data Fig. 3**, reproduced below.

“Downregulation of mitochondrial activity was also observed in H9 early gastruloids despite lower levels of OxPhos protein levels in the H9 primed ESCs (**Extended Fig. 3e,f**).”

For #2 (reviewer suggestion (d)), in the original submission, we had already shown that our mouse gastruloids also demonstrate downregulation of OxPhos proteins (see **Extended Figure 3h**) but the point was not stated as clearly as it might have been. To remedy this as well as to go a bit further, we performed two new analyses.

First, we asked if there was systematic downregulation of mouse OxPhos proteins in our datasets by performing differential expression testing and GO analyses comparing early vs. late gastruloid timepoints. We observed that this was indeed the case with OxPhos subunits being downregulated in late gastruloids with GO terms mapping

to OxPhos, mitochondrial organization, and respiration being downregulated in late gastruloids. These results are presented in **panels a-b** of newly added **Extended Data Fig. 4** (reproduced below).

Second, we compared our mouse gastruloid data with independent work from Stelloo *et al.*¹. We found consistent temporal expression trends in terms of OxPhos downregulation in their independently generated data, further supporting the interpretation that the downregulation is independent of the cell number. These results are presented in **panels c-f** of newly added **Extended Data Fig. 4** (reproduced below).

These analyses are referenced in the following new passage in the Results:

“As such, despite species-specific aspects of the protocols, e.g. the ten-fold lower number of starting cells for mouse vs. human gastruloids, the downregulation of oxidative phosphorylation observed in the transition from the primed state to early human gastruloids is mirrored by a similar downregulation in early to late mouse gastruloids (**Extended Data Fig. 4a-b**). Furthermore, these trends in early vs. late mouse gastruloids reproduce (*i.e.* independent confirmation), extend (*i.e.* by showing homologous patterns in human gastruloids) and add resolution to (*i.e.* by profiling more proteins) similar observations by Stelloo *et al.*⁴² in mouse gastruloids (**Extended Data Fig. 4c-e**).”

Extended Data Figure 3 (panels e-f only). Mapping differentially expressed biological processes across gastruloid development. (e) Representative ATP5F1A fluorescence images of RUES2 and H9 primed ESCs (left) and early gastruloids (right). Blue channel indicates DAPI, Scale bar: 25 μ m. **(f)** Boxplots of normalized ATP5F1A immunofluorescence intensity.

Extended Data Figure 4. Temporal protein profiling recapitulates systematic downregulation of OxPhos over mouse gastruloid differentiation. (a) Volcano plots depicting the DEPs between early (blue) and late (red) mouse gastruloids, where x-axis represents the \log_2 fold change between the two timepoints and y-axis represents the negative \log_{10} of the BH-adjusted p-value. Labels indicate OxPhos subunits. (b) Dot plots indicating the GO terms enriched in DEPs between early and late mouse gastruloids. Color scales for dot plots indicate the BH-adjusted p-value and sizes of dots indicate the number of genes detected within each term. (c) Schematic of OxPhos complexes. (d) Number of significantly changing OxPhos proteins in this study and Stelloo et al.⁴² (e) Comparison of the temporal dynamics of OxPhos proteins in mouse gastruloid development between this study (left) and Stelloo et al.⁴² (right).

5. Could the authors speculate on why early gastruloid mouse samples specifically showed low RNA-protein expression profile correlation?

We speculate that at the protein level, genes associated with aerobic respiration and mitochondrial activity drive the observed discordance, as evidenced by the GO term enrichment shown in **Fig. 4h**. We adjusted the text in Results and Discussion sections to better articulate this speculation:

“Overall, we observed varying profiles of discordance across mouse gastruloid development (**Fig. 4g**; **Extended Data Fig. 7e**) and applied GO enrichment analysis to genes with absolute discordance ratios greater than 1 (*i.e.* protein either highly more or less abundant than expected, given RNA levels) across

each developmental stage. In early mouse gastruloids, we observed that the discordance tended to be driven by mitochondrial and metabolic processes (**Fig. 4h; Supplementary Table 9**).”

“When comparing protein abundances to their corresponding transcripts, we observe modest correlation ($r_{\text{Pearson}} = 0.39$) with a clear discordance between mitochondrial proteins and transcripts including proteins underlying key metabolic pathways such as oxidative phosphorylation being anticorrelated. However this was not the case for other pathways such as WNT signaling, steroid biosynthesis and glycolysis. Our findings are in broad agreement with other studies mapping RNA-protein relationships in developmental contexts across a range of organisms^{16,19–23} and highlight the need to study multiple layers of biomolecular composition of organisms during development. However, our study is limited to profiling at the transcript and protein levels. Future studies, applying ribosome profiling¹²⁶ and single cell multi-omics of the proteome and transcriptome¹²⁷ may inform the translation and turnover rates of proteins involved in discordant mitochondrial and metabolic processes across developmental stages.”

6. Along the lines of point 1.2 above, in Extended Data Fig. 5h, Sox2 displays two peaks in abundance, during both naïve and early gastruloid stages. Do the same Sox2 targets correlate with TF abundance at these developmental distinct stages?

Thank you for this suggestion. We compared Sox2 target transcripts that were upregulated (≥ 2 fold change to the mean abundance) between the naïve and early stages. We identified 229 targets upregulated specifically in the naïve stage, 176 targets specifically in the early stage, and 69 targets enriched in both stages. Subsequent GO analysis revealed that the early-stage targets were associated with processes including regulation of cell adhesion, meiosis, placenta development, and transposition. In contrast, naïve-stage targets were enriched primarily for genes involved in response to leukemia inhibitory factor (LIF), which aligns well with literature and the fact that naïve mESCs are cultured in the presence of LIF. We now describe this in the Results and added plots in **Extended Data Fig. 8f-g**, reproduced below (also shown above in response to your comment #1)

“Out of the 245 naïve-stage Sox2 targets and 298 early-stage Sox2 targets, 69 were enriched in both stages (**Extended Data Fig. 8f**). GO analyses on the discrete Sox2 targets revealed that naïve-stage targets were enriched for response to LIF, while early-stage targets were enriched for processes associated with cell adhesion, placenta development and meiosis (**Extended Data Fig. 8g**).”

Extended Data Figure 8 (panels f-g only). Sample-matched temporal multi-omic profiling of mouse gastruloid differentiation reveals stage-specific upregulation and biological commonalities of downstream transcription factor targets. (f) Scatterplot comparing levels of downstream Sox2 targets in naïve stage ESCs and early stage gastruloids. Naïve and early stage enriched targets colored in orange and blue respectively while brown points indicate enriched Sox2 targets upregulated in both. (g) Dotplot highlighting the biological processes significantly enriched in

downstream targets of Sox2. Color scale indicates the p-value adjusted using the Benjamini-Hochberg procedure and sizes of dots indicate the number of genes detected within each term.

Reviewer #1 (Remarks on code availability):

DOI or URL: <https://github.com/bbi-lab/Temporal-Gastrulomics>

I have not reviewed the code myself but have accessed the authors Github page. The authors have done a terrific task at making the data and the codes readily accessible to the community

We thank the reviewer for these kind comments.

Reviewer #2 (Remarks to the Author):

In this study, Garge et al. profiled the proteomic and phosphoproteomic landscapes of mammalian gastruloid development. They generated proteomic and phosphoproteomic data and integrated them with publicly available scRNA-seq datasets to characterize the progression from naïve ESCs to late stage gastruloids in both human and mouse models. Their results highlight dynamic changes at both the protein and phosphorylation levels during gastruloid formation, reveal RNA–protein discordance, and identify conserved phosphorylation sites between species. However, given the limited novelty relative to recent publications and the insufficient insights currently offered by the study, I cannot support the publication of this manuscript in Nat Cell Biol.

We thank the reviewer for their comments. The primary concern of this reviewer, as we understand it, relates to novelty and insights relative to Stelloo *et al.*, which focused on proteomic profiling of mouse gastruloids. We emphasize that the two studies were initiated independently (Stelloo *et al.* appeared while we were writing our manuscript), and, as often happens in science, this independence naturally yields both points of convergence (*i.e.* where similar aspects of study design provide independent replication of observed trends) as well as points of divergence (*i.e.* where differences in model systems, experimental design, and analytical depth generate new insights). Across several analyses, the convergences support the strength of both studies' conclusions (*e.g.* **Extended Data Fig. 2a–d**, **Extended Data Fig. 3h**, **Extended Data Fig. 4b–c** of the revised manuscript). Replication of conclusions by independent studies is an important facet of scientific progress given widespread concerns about irreproducibility (particularly relevant to stem cell and organoid research), and we argue should not be too easily dismissed, particularly in a new area such as gastruloid proteomics.

However, the divergences, where our study contributes beyond Stelloo *et al.*, span even more dimensions of the work, particularly so with the revisions. We cover these in detail in response to specific points, but summarize them here: (i) substantially deeper proteome coverage in both species, enabling network-level analyses not possible previously; (ii) incorporation of human RA-gastruloids that reach advanced morphological and lineage states, providing the first proteomic and phosphoproteomic maps of human post-implantation embryo models; (iii) an experimentally matched, multimodal dataset (RNA, protein, phosphostate) that enables quantification of pathway-specific and developmental stage-specific RNA–protein discordance; (iv) Experimental validation of phosphosite dynamics of H2AX S140 as well as the inference of kinase activities which we validate through MAPKAPK2 inhibition; (v) comparison of conserved versus species-specific temporal programs; and (vi) construction of a co-regulatory protein network whose predictions we validate through CRISPR-Cas9

perturbations of Commander subunits (COMMD9 and COMMD10) and their cooperative proteins (DPYSL4 and PRKACB).

In the revision, we have clarified this context and more explicitly articulated how our experiments and data both align with and extend beyond the recently reported work from Stelloo *et al.*

Major issues:

1. Stelloo *et al.* (PMID: 38754429) recently profiled the gastrulation process in mouse embryos and gastruloids using multilayered proteomics. Their study combined bulk proteomics, phosphoproteomics, and single-cell proteomics, and included more detailed time points throughout gastrulation. Although the authors integrated transcriptomic data and included human gastruloids, these additions are incremental and do not offer substantial new insights. Overall, I do not find that this work surpasses the previous one either technically or conceptually.

While we agree with the reviewer that Stelloo *et al.* performed multilayered proteomics, we disagree with the reviewer that our dataset fails to surpass this previous work. Technically, using TMT-based quantitative proteomic workflows, we sample 2,303 more human and 3,290 more mouse proteins, representing a 46% and 65% increase in the resolution and depth of the proteome sampled over the course of gastruloid differentiation. We have amended the Results section and added a new **Extended Data Fig. 2** summarizing these comparisons to Stelloo *et al.*, as reproduced below:

“We quantified 3,290 more mouse proteins (65% increase) and 2,303 more homologous human proteins (46% increase) compared to recent mouse gastruloid proteomics datasets⁴² (**Extended Data Fig. 2a,b**). Strong overlap with human and mouse embryonic proteome datasets support the interpretation that we sampled biologically relevant temporal protein changes (**Extended Data Fig. 2c,d**). The increased depth of the proteome sampled over the course of gastruloid differentiation also enabled temporal coregulatory analysis at the level of proteins, complexes, and phosphosignaling (**Extended Data Fig. 2a,b**). Taken together, these data along with the dedicated web application enable exploration and new understanding of the temporal dynamics of RNA, protein, and cell signaling models of human and mouse gastrulation.”

Extended Data Figure 2. Quantitative proteomics of human gastruloids expands protein coverage and recapitulates temporal trends observed during gastruloid differentiation. (a) Barplot of the number of proteins quantified in human and mouse gastruloids [this study] vs. mouse gastruloids [Stelloo et al.⁴²]. (b) Venn diagram showing intersection of proteins detected in human gastruloids [this study], mouse gastruloids [this study], and mouse gastruloids [Stelloo et al.⁴²]. (c) Proportion of the 7,352 proteins detected in human gastruloids [this study] also detected in published human and mouse embryo proteomics datasets^{42–44}. (d) Left: Venn diagram showing intersection of proteins detected in human gastruloids [this study], mouse gastruloids [Stelloo et al.⁴²], and mouse embryos [Stelloo et al.⁴²]. Right: Comparison of temporal trends of selected proteins in mouse embryos [Stelloo et al.⁴²] vs. human gastruloids [this study].

Beyond depth of proteomic profiling, our work goes beyond Stelloo *et al.* along multiple axes, many of which are strengthened in the revision. **First**, we profile not only mouse gastruloids but also human gastruloids, enabling cross-species comparisons not possible in a single-species study. In particular, we use a human RA-gastruloid model that we recently reported⁴. Human RA-gastruloids reach more advanced developmental stages (roughly comparable to E9.5 mouse embryos) and encompass a broader range of lineages (neural, somitic, renal, cardiac, gut) than conventional mouse or human gastruloids⁴, providing (to our knowledge) the first proteomic and phosphoproteomic maps of human post-implantation embryo models.

Second, our design is explicitly multi-omic and sample-matched (RNA, protein, phosphosite), which allows us to quantify pathway-specific RNA–protein discordance, identify stage- and pathway-specific patterns in metabolic rewiring, and infer kinase activities across developmental time. In the revision, we not only report these inferences but also validate them experimentally, for example by testing predicted MAPKAPK2 activity with chemical inhibition and validating specific phosphosite dynamics (e.g. H2AX S140) by immunofluorescence. We also expand on metabolic findings by demonstrating conserved downregulation of oxidative phosphorylation in both human and mouse gastruloids and across cell lines, including new ATP5F1A immunofluorescence in RUES2 and H9 gastruloids.

Third, the increased coverage and temporal resolution enable network-level analyses that go beyond those in Stelloo *et al.* We construct a co-regulatory protein network spanning complexes and pathways, use it to nominate disease-associated modules, and then test these predictions through genetic perturbation. In particular, we perturb Commander subunits (COMMD9 and COMMD10) and their cooperative partners (DPYSL4 and PRKACB) by CRISPR-Cas9 and demonstrate predicted phenotypic consequences, thereby linking the resource to concrete mechanistic insights rather than leaving it as a purely descriptive atlas.

The reviewer-only figure below illustrates how greater depth and species-breadth enables both recovery of established developmental regulators and discovery of previously unappreciated candidates. For instance, we detect 176 human and 300 mouse transcription factors not observed by Stelloo *et al.*, including regulators of pluripotency (NANOG), neurogenesis (SOX1, SOX6, SOX9), neural crest (MSX1), endoderm (CDX1), somitogenesis (PAX3, DACH2), EMT (SNAI1), and cardiac development (FOXP1) (**Reviewer Fig. 1a-b**). Their temporal dynamics in our dataset align with the expected appearance of advanced lineages in human RA-gastruloids, confirming a broader developmental spectrum than conventional mouse or human gastruloids.

Reviewer Figure 1. Enhanced proteomic coverage of gastruloid differentiation enhances coverage of developmentally regulated transcription factors. (a) Venn diagram showing intersection of the 617 transcription factors detected in our study with those detected in Stelloo *et al.*, (b-c) Heatmaps of known (b) and uncharacterized (c) transcription factors. (d) BioPlex interaction network of ZNF446, ZNF316, and ZNF358.

Beyond known factors, the increased depth enables nomination of understudied genes for roles in early development. For example, ZNF316, ZNF446, and ZNF358—transcription factors with almost no functional characterization and undetected by Stelloo *et al.*—show tightly correlated expression profiles across human gastruloid development (**Reviewer Fig. 1c**). BioPlex data further support that these factors participate in a shared interaction neighborhood (NOTCH2, FGF13, SCRIB, EFEMP1, LTBP2, FBLN2, VWCE), all of which have links to neural development (**Reviewer Fig. 1d**). This provides concrete hypotheses that these ZNFs contribute to neural development and illustrates how the expanded proteomic landscape can seed mechanistic follow-up on poorly characterized genes. While we have not incorporated this specific example into the revised manuscript for reasons of narrative flow, we include it here to underscore the types of questions that our dataset enables but are inaccessible from the Stelloo *et al.* resource alone.

More broadly, the multi-omic and cross-species nature of our study, combined with the additional experimental work introduced in revision (new immunofluorescence, kinase inhibition, and genetic perturbations), means that our dataset does not simply parallel Stelloo *et al.* in a different format, but both independently replicates key trends and opens up new territory, both conceptually and as a human+mouse resource.

2. The scRNA-seq data lack coverage of the naïve and primed stages, which limits the ability to perform comprehensive co-analysis of the transcriptome and proteome. As a result, only two gastruloid stages are available for comparison.

The reviewer is correct that we have not performed scRNA-seq on naïve and primed ESCs. However, to be clear, the comparison of bulk proteomics data with scRNA-seq data was performed solely to demonstrate that despite being bulk proteomic observations, our multi-omic workflow was able to detect temporally driven, cell type-specific proteomic changes. For most of the comparisons, we are using **matched bulk** RNA-seq data for proteome vs. transcriptome comparisons, **at all stages and in both human and mice**. We consider the matching to be key, given heterogeneity across labs but also across experiments and users within a single lab. As single cell proteomic technologies progress, similarly matched scRNA-seq + sc-Proteome datasets may be

possible in the future. Our comparison of non-matched scRNA-seq vs. bulk-proteome data had a limited purpose and was not the primary means by which we built our multi-omic resource.

3. As a developmental stage during which the anteroposterior axis is established, bulk proteomic profiling may obscure critical spatial information related to body axis formation.

We agree with the reviewer—both the non-spatial and non-single cell aspects of bulk proteomic profiling are less than ideal. However, just as co-analysis of bulk proteomic + scRNA-seq data has the potential to enhance the interpretation of bulk proteomic data, the same may be possible by combining bulk proteomic data with targeted spatial profiling. To highlight one example, we observe that SOX2 and ZIC2 were highly correlated in our dataset at the protein level, and their scRNA-seq profiles suggested both were expressed in neural cell types. Upon performing immunostaining, we observed ZIC2 in nuclear bodies as previously reported⁵ and colocalized with SOX2 to neural cell types. We incorporated this specific case as new **panel d** in **Extended Data Fig. 5** as an illustration of how temporal trends in bulk proteomic data, when intersected with other datasets, can nominate verifiable hypotheses with respect to cell type-specificity (and possibly spatial specificity, *e.g.* one could imagine analogous intersections with gastruloid-wide spatial transcriptomic datasets). We amended the Results section to clarify this point as follows:

“To highlight one example of cell type specific protein expression trends, SOX2 and ZIC2 were highly correlated in our dataset at the protein level, and their scRNA-seq profiles suggested both were expressed in neural cell types. Upon immunostaining, we observed ZIC2 in nuclear bodies as previously reported³⁰ and colocalized with SOX2 to neural cell types. Thus, although limited by the bulk nature of the proteomic measurements, these observations suggest that our data captures at least some cell type-specific expression patterns for major lineages.”

Extended Data Figure 5 (panel d only). Mapping cell types contributing to bulk proteomic observations. (d) Immunostaining of 120 hour RA-gastruloids reveals co-expression of SOX2 (green) and ZIC2 (red) in neural tube cells. Blue channel indicates DAPI stain. Scale bar 50 μ m.

4. Although gastruloids are valuable *in vitro* models for studying mammalian gastrulation, this study relied exclusively on them to investigate normal embryonic development, without incorporating natural embryos as authentic references (at least in the case of mouse) or *in vivo* data to validate the findings.

Owing to the obvious ethical and experimental limitations associated with studying human embryos around gastrulation, and the substantial technical constraints on obtaining sufficient material from mouse embryos for deep, time-resolved multi-omic profiling, we deliberately focused this study on human and mouse gastruloids. That said, we agree that anchoring gastruloid data to *in vivo* references is important, and in the revision we have sought to make this connection more explicit. Proteomic studies in human embryos³ profile preimplantation stages, and mouse embryo proteomics^{1,2} cover developmental windows that only partially overlap with our gastruloid time course.

In the revised manuscript, we compare our datasets directly to these embryo proteomes (see **Extended Data Fig. 2c-d**, reproduced on page 11 above in response to Reviewer #2, Comment #1). We recapitulate >75% of proteins profiled in these studies and observe that temporal expression trends for key developmental and cell-state markers, e.g. DNMT3B (stem cell differentiation), EPCAM and VIM (epithelial-to-mesenchymal transition), NCAM1 and GLI3 (neural development) and TBX6 (somitogenesis), are mirrored in our human gastruloid

dataset. Moreover, we find that temporal profiles of human proteins spanning diverse cellular processes in our human RA-gastruloids are recapitulated in mouse embryo proteomics.

Together, these analyses show that although our primary material is *in vitro* mouse and human gastruloids, the proteomic and phosphoproteomic dynamics we report are closely aligned with available *in vivo* data and therefore provide a relevant reference for normal mammalian embryonic development.

5. The authors identified numerous phosphosites on key proteins that are dynamically regulated during gastruloid formation. It would be valuable to validate some functionally relevant phosphosites, as this could guide future investigations in the field.

In response to this point, we performed two types of validation experiments. **First**, we validated the temporal dynamics of H2AX S140 using immunofluorescence, comparing both H9 and RUES2-GLR primed ESCs states as well as early and late gastruloids. We observed that H2AX S140's phosphostate was in concordance with our bulk phosphoproteomics data, with its levels elevated in primed RUES2-GLR ESCs and late gastruloids.

Second, we reasoned that the kinase activity prediction from our phosphoproteomics datasets offers starting hypotheses of kinases that may be important for gastruloid development. From our phosphoproteomics datasets, we predicted that MAPKAPK2 activity increases during gastruloid development. This prediction was in line with observed MAPKAPK2 protein profiles during gastruloid development. Further, we found that its downstream target, ZFP36L1 S92, was correlated with its upstream kinase levels. These observations, coupled with previous reports of ZFP36L1's roles in development, suggested that MAPKAPK2 may play crucial roles in gastruloid development. Upon chemically perturbing MAPKAPK2 during gastruloid development, we observed that gastruloids displayed abnormal morphologies, including multi-axis, lack of patterning, and gastruloid cells tending to express SOX2. These data suggest that MAPKAPK2 indeed plays an important role in regulating cell type differentiation in gastruloids. While this is an indirect validation of its MAPKAPK2's downstream phosphorylation effects, it highlights the value of our resource in nominating kinases (and their targets) for functional dissection and hope that these data can inspire similar efforts in the future.

We modified the Results section and added new panels to **Extended Data Figs. 9 (panels e,f and j-m)** and **10 (panel i)** with these new experiments/analyses, as reproduced below.

"Immunofluorescence confirmed that H2AX S139/S140 phosphorylation dynamically changes across human gastruloid development (**Extended Data Fig. 9e**). H2AX S139/S140 phosphorylation was highest in RUES2 primed ESCs, lower in H9 primed ESCs, and markedly reduced in early gastruloids before increasing again in late-stage gastruloids (**Extended Data Fig. 9f**)."

"Given this observation and ZFP36L1's roles in embryonic development and implication in developmental defects¹⁰¹, we hypothesized that MAPKAPK2 may play functional roles in symmetry breaking and body axis formation. In the presence of the MAPKAPK2 inhibitor MK2in1 (**Extended Data Fig. 10i**), gastruloids failed to elongate and displayed multi-axis morphology with the majority of gastruloid cells expressing SOX2 at 120 hours of induction (**Extended Data Fig. 9j-m**). The elevation of SOX2 levels began after 48 hours of induction (**Extended Data Fig. 10i**) and continued until the end of gastruloid induction."

Extended Data Figure 9 (panels e-f, j-m only). Quantitative phosphoproteomics reveals kinase activities across gastruloid development. **(e)** Bar plots of scaled H2AX pS139/pS140 abundance changes across human ESCs and gastruloid developmental stages. **(f)** Validation of differential phosphorylation state of pS139/pS140 (red) in Primed H9 (left) and RUES2-GLR (right) ESCs. Blue channel indicates DAPI, Scale bar: 25 μm . **(j)** Representative images of 120 hour gastruloids cultured with DMSO (left) and MAPKAPK2 inhibitor, MK2in1 (right). Fluorescent images depict SOX2-mCitrine expression. Scale bar: 200 μm . **(k)** Fraction of multi-axis gastruloids when treated with DMSO (control) and MK2in1 (MAPKAPK2 inhibitor). Fractions calculated from 16 gastruloid observations for each condition. **(l)** Boxplots depicting the differences in gastruloid area **(l)** and fraction of SOX2+ cells **(m)** when treated with DMSO and 10 μM MK2in1.

Extended Data Figure 10 (panel i only). Mapping phosphorylation states across gastruloid development. **(i)** Protocol and timecourse of RA-gastruloid development when treated with DMSO (top row) and MAPKAPK2 inhibitor, MK2in1 (bottom row). Fluorescence images indicate the expression of SOX2-mCitrine. Ess6, Essential 6 media; CHIR9, CHIR99021; RA, Retinoic Acid. Scale bar: 200 μm .

Minors:

1. In extended data Fig 1d and 1e revealed a negative correlation between naïve and primed H9 ESCs, as well as a positive correlation between early and late samples in human proteomic and phosphoproteomic data. In contrast, mouse data showed a positive correlation between naïve and primed samples, while early and late samples exhibit a negative correlation. The author should conduct further analysis to elucidate this species difference.

Thank you for raising these important points.

Re: “a negative correlation between naïve and primed H9 ESCs”, we note that the proteomes of naïve H9 ESCs are generally outliers in our data, as evidenced by poor correlation and the high number of DEPs observed when comparing it with other timepoints. This is discussed further in our response to the next point raised by the reviewer.

Re: the “negative correlation between early and late mouse gastruloids”, our analyses show this is driven by metabolic state, particularly the mitochondrial proteome (**Fig. 4g,h**). We have updated the Results to clarify this point in the following passage:

“As such, despite species-specific aspects of the protocols, *e.g.* the ten-fold lower number of starting cells for mouse vs. human gastruloids, the downregulation of oxidative phosphorylation observed in the transition from the primed state to early human gastruloids is mirrored by a similar downregulation in early to late mouse gastruloids (**Extended Data Fig. 4a-b**).”

These analyses make us reluctant to claim any species difference based on these patterns. We would further note that: (i) our analyses directed at aligning the stages found that the human primed RUES2-GLR cells’ proteome resembles the early mouse gastruloids’ proteome (see **Fig. 2a,b** and **Extended Data Fig. 2h**); and (ii) despite stage-matching of both the human and mouse gastruloid timepoints, the differences in how mammalian gastruloid protocols are implemented (not just by us, but by the field) include species-specific aspects that may be impossible to circumvent.

Although we obviously believe in the power of gastruloids for investigating both early development both within and across species, these limitations are important, and we have amended the Discussion to include the following passage noting them:

“Fifth, the absence of standardized mammalian gastruloid culturing techniques makes it difficult to unambiguously distinguish species-specific vs. protocol-specific variation. Future studies performing multi-omic profiling of gastruloids of the same species prepared under different conditions (*e.g.* varying starting cell number^{134,135}) may be necessary to shed light on which aspects of the gastruloid proteome are robust vs. sensitive to protocol variation, while species comparisons would benefit from greater harmonization of mouse and human gastruloid protocols.”

2. In extended data Fig 2a, it is surprising to see that there are more DEPs between naïve and primed H9 ESCs, or even between the two primed ESC lines, than between naïve ESCs and late stage gastruloids. But this pattern is not observed in the mouse data. The authors should re-examine their results and attempt to validate these distinctions using transcriptomic data.

Thanks for raising this. We refrained from emphasizing comparisons between H9 and RUES2-derived samples from different stages because any stage-based differences might be confounded by cell line differences.

However, there are indeed a large number of DEPs between these naïve H9 cells and other samples. As suggested, we examined the corresponding transcriptome data and observed similar trends. In particular, naïve H9 cells have an elevated number of differentially expressed transcripts (DETs) when compared to all other samples. Below we plot a heatmap of DEP and DETs to convey this point (now incorporated to **Extended Data Fig. 3a**). As discussed above, the elevated number of DEPs observed between the Primed RUES2 vs. other timepoints is driven by the upregulation of mitochondrial processes. The panel is cited in the following passage:

“Owing to cell line differences, we refrained from directly comparing naïve H9 cells to the other stages. However, naïve H9 cells tended to exhibit a high number of both DEPs and differentially expressed transcripts (DETs) when compared to other samples/timepoints (**Extended Data Fig. 3a**)”

Extended Data Figure 3 (panel a only). Mapping differentially expressed biological processes across gastruloid development. (a) Heatmaps indicating the number of differentially expressed proteins (left) and transcripts (right) between pairs of human samples. DE proteins (DEPs) and transcripts (DETs) determined by those having absolute log2 fold change ≥ 1 and BH-adjusted p-value < 0.05 .

Reviewer #3 (Remarks to the Author):

Review of NCB-RS57819: “The proteomic landscape and temporal dynamics of mammalian gastruloid development” by Garge et al.

Summary

The manuscript by Schweppe and colleagues presents an interesting, ambitious and technically impressive multi-omic atlas of human and mouse gastruloid development, with matched profiling of transcriptome, proteome, and phosphoproteome dynamics across key developmental stages. While it lacks cell-type resolution at the proteome level, the study is valuable as a rich resource and offers important cross-species comparisons. In its current form, however, the authors over-interpret correlation-based findings and do not distinguish descriptive results from mechanistic conclusions. The authors have compiled a compelling resource of clear value to the field, but should tone down their claims to more accurately reflect the limitations of the methodology and the inference nature of the work. So while the manuscript has the potential to make a meaningful contribution to the developmental biology, systems biology and stem cell communities, it would benefit from a clearer framing and more restrained interpretation of bulk co-expression data.

We thank the reviewer for characterizing our study as a rich, compelling and valuable resource, as well as for calling out potential overstatements in the original submission. In the revision, we have toned down the claims by: (i) clarifying in the main text the methodology and limitations of our approach, and (ii) amending the Results to incorporate new experiments to test hypotheses generated through the resource.

Major points

1. The manuscript lacks a clearly articulated central motivation. It straddles resource generation, hypothesis generation, and mechanistic inference without prioritizing one. In its current form, it is best categorized as a comprehensive exploratory dataset rather than a mechanistically resolved study. The authors should clarify if the primary goal was to provide foundational datasets or to make new biological discoveries about gastrulation. This could help focus the narrative and sharpen the conclusions and impact.

We apologize for the lack of clarity regarding the central motivation. The primary goal was indeed to generate a foundational dataset, with a secondary goal of demonstrating how such data could be leveraged to make new discoveries (with the latter now strengthened through additional experiments added during revision). The introduction has been revised to sharpen our articulation of these points:

“The primary motivation of this study was to generate a foundational proteomic resource to understand the temporal dynamics of gastrulation. To this end, we applied high-throughput quantitative mass spectrometry to quantify proteins and phosphosites across four key stages of gastruloid differentiation. We demonstrate the utility of the resulting data by mapping the dynamics of hundreds of known protein complexes, while also identifying additional proteins whose temporal profiles correlate with specific complexes, suggesting cooperative relationships during early development.”

2. Interpretation of RNA-protein discordance as evidence of widespread post-transcriptional regulation is overstated. The observed modest correlation is consistent with a large body of existing literature and could result from multiple factors, including protein stability and cellular heterogeneity. Without direct evidence of regulation (e.g. degradation rates, translational profiling), these claims should be tempered.

Thanks for raising this. We fully agree that factors such as differential rates of translation and degradation surely contribute to RNA-protein discordance. We reviewed the manuscript to correct potential overstatements with respect to post-transcriptional regulation, and identified and sought to correct two overstatements that we believe this comment is referencing. First, in the Results, we deleted “...consistent with post-transcriptional and post-translational control of mitochondrial protein levels during development” in the context of describing GO enrichments. Second, in the Discussion, we deleted “...and highlight the roles of post-transcriptional regulation in gastruloid development...”. We also adjusted the following passage of the Discussion section to tone this down:

“However, our study is limited to profiling at the transcript and protein levels. Future studies, applying ribosome profiling¹²⁶ and single cell multi-omics of the proteome and transcriptome¹²⁷ may inform the translation and turnover rates of proteins involved in discordant mitochondrial and metabolic processes across developmental stages.”

3. Co-regulation network analysis is a strength of the study, but the claims are overstated, and conclusions must be treated cautiously. Correlation does not establish functional interaction or complex formation. Many inferred associations are speculative and should be explicitly presented as hypotheses rather than established mechanisms, especially when links to disease phenotypes.

We agree with the reviewer that correlation does not establish physical/functional interactions, and apologize for any overstatements. To address this, in the Results section, we have deleted a sentence in the original submission that read: “In summary, cooperative protein analysis recovered both known physical interactions as well as potentially novel associations between complexes and cooperative proteins.”

We also modified a related sentence to now read:

“Although correlative, functional proteomics can be a powerful approach to nominate molecular functions for disease-associated genes, as well as to nominate new disease candidate genes in developmental contexts^{106,107}.”

To be clear, we agree that the proteomics or multi-omics alone can at best provide strong starting hypotheses that future studies can leverage to narrow down the possible sets of proteins that share physical interactions, participate in biochemical pathways, and/or share cellular roles in this developmental context. In the revision, we sought to better showcase how such nominations can be further evaluated.

Specifically, we curated genes from the Commander co-regulatory network, and then perturbed two Commander subunits (COMMD9 and COMMD10) and 2 cooperative subunits (PRKACB and DPYSL4) in gastruloids. We observed that perturbed gastruloids were generally stunted in growth and displayed abnormal neural tube morphology. We measured the morphometric features of the neural tube (SOX2⁺) cells and observed that COMMD10, COMMD9, and DPYSL4 had reduced area and shorter major axes with low eccentricity (Major:Minor axis ratio). We updated the Results added new figures to show these results, namely **panels f, g of Fig. 4** and **panel d of Extended Data Fig. 11**:

“We hypothesized that the Commander co-regulatory network could identify genes associated with developmental disorders. To test this, we perturbed two Commander subunits (COMMD9 and COMMD10) and two co-regulatory proteins (DPYSL4 and PRKACB) in human ESCs and generated gastruloids from these knockout lines. Knockout of both Commander subunits, COMMD9 and COMMD10 failed to elongate, resulting in abnormal neural tube morphology (**Fig. 5f**). Perturbation of DPYSL4 resulted in similar abnormal neural tube morphologies, phenocopying the knockouts of the Commander subunits (**Fig. 5f**). Across knockouts, perturbed gastruloids had reduced areas and a pronounced reduction in the length of major axes (**Fig. 5g; Extended Data Fig. 11b**). Perturbation of PRKACB also resulted in gastruloids with reduced areas, though the reduction in major axis length was less pronounced.”

Figure 5. Co-regulatory networks of protein dynamics in gastruloids link to shared phenotypes and developmental disorders (panels f-g only). (f) Representative images of WT (wild-type), COMMD10 KO, COMMD9 KO, DPYSL4 KO, PRKACB KO gastruloids. n= 48 gastruloids per genotype, Scale bar: 200 µm (g) Boxplots comparing the area and major axis length of wild-type vs. genetically perturbed gastruloids. Significance determined using standard t-test.

Extended Data Figure 11 (panels d only). Mining co-regulatory protein networks to nominate new disease gene candidates. (d) Boxplots comparing the distributions of minor axis length(left) and major-to-minor axis ratio (Eccentricity, right) of wild-type and perturbed gastruloids. Significance determined using standard t-test.

We also updated the Discussion to emphasize the role of co-regulatory networks as hypotheses generators and their limitations for causal inference:

“Our work highlights the potential of co-regulatory networks as hypotheses generators to uncover the roles of understudied genes in development contexts, particularly those related to disease. As an illustration of this potential, we focused on subunits of the Commander complex, which are associated with Ritscher-Schinzel syndrome. In our data, the co-regulatory network containing the Commander complex consisted of 2 of the 4 subunits strongly linked to Ritscher-Schinzel syndrome^{128,129}. The cooperative network containing this complex consisted of 2 of the 4 genes strongly linked to Ritscher-Schinzel syndrome. Perturbations to proteins co-regulated with the Commander complex, including DPYSL4 and PRKACB, led to similar morphological phenotypes as observed when perturbing Commander subunits in gastruloids. DPYSL4 has been associated with neural functions including neurite initiation and dendrite growth of hippocampal neurons^{130,131} and PRKACB has been associated with neural tube defects¹³² while their molecular mechanisms and temporal roles during early embryogenesis remain understudied. These results support the value of network-based predictions as powerful starting points to better understand the roles of genes in gastrulation.”

“Fourth, although co-regulatory protein networks provide a strong starting point for inferring functions of understudied genes in development, they are correlative and lack molecular and biochemical resolution. Integrating structural modeling and interactome-mapping may help improve the quality of hypotheses nominated for functional validation and mechanistic investigation.”

We highlight another example here demonstrating the utility of our co-regulatory network. We found that TMC01 and LRRC59 were strongly correlated in our datasets and it was previously observed that these proteins co-elute in biochemical separations in a targeted study⁶ to map endosomal protein-protein interaction networks. We used this orthogonal evidence and hypothesized that these 2 proteins may interact and used AlphaFold3 (Multimer) to predict binding interfaces and indeed found that contacts between both proteins were predicted with high PAE scores. While we have not included this analysis and figure in the paper, we’d like to share this in the response document to highlight the utility of our dataset to narrow down hypotheses of novel protein-protein interactions in developmental contexts.

Reviewer Figure 2. Structural insights into LRRC59 and TMCO1 from gastruloid proteomics datasets. (a) Heatmap depicting the temporal dynamics of LRRC59 and TMCO1 protein expression across human gastruloid differentiation samples and replicates. **(b)** AlphaFold3 model and **(c)** predicted pLDDT scores for LRRC59 (red) and TMCO1 (blue). **(d)** Predicted aligned error scores for LRRC59 and TMCO1.

4. Cross-species comparison is potentially valuable, but the finding that human primed ESCs resemble early mouse gastruloids could be explainable by cell line- or culture-specific metabolic differences rather than conserved developmental timing. Regardless, this section would benefit from a more critical discussion of potential confounders and more cautious language around evolutionary interpretation.

We fully agree that there are potential confounders and caution is important. In the revision, we have sought to be more cautious in our language both the corresponding Results and Discussions sections, and have added the following passage to the Discussion that directly speaks to this point:

“Fifth, the absence of standardized mammalian gastruloid culturing techniques makes it difficult to unambiguously distinguish species-specific vs. protocol-specific variation. Future studies performing multi-omic profiling of gastruloids of the same species prepared under different conditions (e.g. varying starting cell number^{134,135}) may be necessary to shed light on which aspects of the gastruloid proteome are robust vs. sensitive to protocol variation, while species comparisons would benefit from greater harmonization of mouse and human gastruloid protocols.”

5. Bulk proteomic profiling, while well executed and valuable, of heterogeneous gastruloids limits the ability to assign observed molecular dynamics to specific cell types. While scRNA-seq data are used for inference, these mappings are indirect. Without cell-type-specific proteomic resolution, conclusions about lineage-specific programs should be presented more tentatively.

We fully agree. A sentence in the Results has been adjusted to now read:

“Thus, although limited by the bulk nature of the proteomic measurements, these observations suggest that our data captures at least some cell type-specific expression patterns for major lineages.”

A relevant passage in the Discussion has been adjusted to now read:

“Third, gastruloids consist of diverse cell types arising from all three germ layers. Our approaches were all bulk measurements and lacked cell-type-specific resolution, such that any lineage-specific claims remain speculative. Additional characterization of separate cell types with fluorescence activated cell sorting (FACS)⁴², or via emerging single cell proteomic technologies, could help to overcome this limitation.”

Minor points

-Key terms such as “cooperative proteins” and “discordance” should be clearly defined.

The main text now includes what we hope are clearer definitions of these terms, as well as reference to the Methods section. Relevant passages include:

“Drawing from previous high-throughput proteomics studies^{52,65}, we defined a protein cooperativity metric to enrich for first-degree neighbors of known complexes and pathways, termed “cooperative edges” (**Extended Data Fig. 6h**). We define cooperative proteins as those participating in a cooperative edge (**Methods**).”

“We defined a metric of discordance between RNA and protein measurements—the log₂ transformed ratio of the average fold change of a protein to its corresponding RNA—at a given stage of gastruloid development (**Methods**). Thus, discordance values close to 0 signify comparable levels of RNA and protein, positive discordance implies that the protein is more abundant than its corresponding transcript and vice versa.”

We have also added detail to the corresponding Methods sections:

“Bioinformatic identification of cooperative protein interactions

We searched all nodes in our correlation network against known complexes and pathways which consisted of at least 3 subunits. We adapted a previously described approach⁶⁵ and employed a Fisher’s exact test to compute statistical enrichment of cooperative complexes with established modules. For each protein complex or pathway module, we tested its neighboring proteins (first-degree edges) for significant association with a particular module and termed those as cooperative proteins. For each protein tested, we first counted the number of edges that it shared with the established module, second we counted the number of edges that linked the module to other proteins (excluding the candidate protein) in the network, third we counted the number of edges the candidate protein had to rest of the correlation network (*i.e.* excluding the module of interest) and finally, we counted the number of edges that were not associated with the candidate protein nor the module of interest. These edge counts were used to compute statistical significance using Fisher’s exact test. We independently repeated this test for all 6,261 proteins against 1,357 known protein complexes and select metabolic pathways. The p-values obtained were adjusted for multiple hypothesis testing using the BH procedure and only cooperative proteins with adjusted p-value < 0.05 were considered significant.”

“Comparison of RNA and protein abundance analysis

Global RNA-protein correlations were calculated using all 9 observations of transcripts and proteins across mouse and human gastruloid development. To ensure stringent analysis, we filtered for genes detected in both species for the downstream analysis. Pseudocounts of 1 were added to filtered count matrices and were converted to transcripts per million (TPM). Mean transcript and protein abundances were converted to log₂ fold change ratios to their respective species geometric mean. For every gene, we calculated the per-gene RNA-protein correlation (r_{Pearson}) using a vector of abundances across 9 samples. GO term enrichment of biological processes in correlated and anticorrelated genes was performed using ClusterProfiler¹⁵⁶. We intersected the 6010 genes detected across both datasets with Human Protein Atlas³⁰

for subcellular locations, CORUM⁶⁶ and ComplexPortal⁶⁷ for protein complexes and KEGG for biochemical pathways⁶². To measure the extent of correlation of transcripts and RNAs within mouse timepoints we calculated the ratio of protein to RNA mean fold changes across each timepoint. In summary, a discordance of 0 implied that the protein and RNAs were highly correlated while discordance less than 0 implied that the RNAs were more abundant than protein levels and vice versa. Discordance scores for protein complexes was calculated by taking the median protein-RNA correlation across constituent members. To prevent averaging pairs of proteins, we only considered complexes where more than 2 proteins were detected in our data. Transcriptional signatures of stage specific mouse transcription factors were detected as follows. First, we calculated the Pearson correlation comparing transcription factor protein abundances to all observed transcripts. We subset the resulting correlation matrix to identify protein-transcript pairs with high correlation ($r_{\text{Pearson}} \geq 0.9$) and used TFLink⁸³ to select only transcripts that were annotated as targets of specific transcription factors. We confirmed the identified transcription factor targets displayed similar temporal regulation to their upstream transcription factor by comparing target transcript abundance at each stage to determine the maximum transcript abundance.”

-Statistical thresholds (e.g. Pearson $r > 0.95$ for co-reg) should be justified, and the sensitivity of results to such choices briefly touched on.

We have modified the text in the Methods section to clearly define the above mentioned terms as follows:

“Correlation network construction and network analysis

We first intersected the human and mouse protein datasets and used 6,261 proteins that were observed across the shared timepoints within a cell line *i.e.* primed ESCs, early and late gastruloids. We normalized each protein’s abundance in given replicate to its respective species geometric mean and \log_2 transformed values for subsequent analysis unless otherwise stated. To construct our correlation network, we first calculated the Pearson correlation coefficients (r_{Pearson}) across all 19,596,930 possible pairs of proteins. Since we already calculated r_{Pearson} across all possible pairs of proteins, we permuted sample labels across our dataset to generate the null distribution of correlation coefficients. Given the relatively lower number of timepoints sampled and the strong bimodal distribution of Pearson correlation coefficients we stringently filtered the network edges with Benjamini-Hochberg (BH) adjusted p-values < 0.01 and absolute $r_{\text{Pearson}} \geq 0.95$. This step filtered the network down to 489,417 (301,561 correlated and 187,856 anticorrelated) pairs but was strongly enriched for protein-protein interactions, macromolecular complexes, and biochemical pathways and was used for subsequent network analysis.”

References cited

1. Stelloo, S. *et al.* Deciphering lineage specification during early embryogenesis in mouse gastruloids using multilayered proteomics. *Cell Stem Cell* <https://doi.org/10.1016/j.stem.2024.04.017> (2024)
2. Gao, Y. *et al.* Protein Expression Landscape of Mouse Embryos during Pre-implantation Development. *Cell Rep* **21**, 3957–3969 (2017).
3. Zhu, W. *et al.* Comparative proteomic landscapes elucidate human preimplantation development and failure. *Cell* **188**, 814–831.e21 (2025).
4. Hamazaki, N. *et al.* Retinoic acid induces human gastruloids with posterior embryo-like structures. *Nat Cell Biol* **26**, 1790–1803 (2024).
5. Uhlén, M. *et al.* Tissue-based map of the human proteome. *Science* **347**, 1260419 (2015).
6. Gonzalez-Lozano, M. A. *et al.* EndoMAP.v1 charts the structural landscape of human early endosome complexes. *Nature* **643**, 252–261 (2025).

POINT-BY-POINT RESPONSE

We thank the reviewers for their careful review of the revised manuscript. Below we describe how we have addressed the remaining concerns raised by Reviewer #2 through textual clarifications and rewording of the manuscript text, per editorial guidance.

Reviewer #2 (Remarks to the Author):

In this revision, the authors have adequately addressed all questions raised by Reviewer 1. However, only a subset of my previous concerns has been satisfactorily resolved. Overall, I consider this study to be promising and potentially suitable for publication, but the remaining issues should be addressed. From my perspective, this work has several notable strengths: 1) a deep and comprehensive proteomic dataset of mouse gastruloid development; 2) the first proteomic dataset characterizing human gastruloid development; 3) the co-regulation network analysis identified genes associated with the Commander complex which are required for proper gastruloid development. Despite these strengths, several important issues remain unresolved. A central concern is how the presented datasets inform our understanding of natural gastrulation. Addressing the points outlined below will be necessary before I can recommend this manuscript for publication in Nat Cell Biol.

Major issues:

1. Although recently published mouse gastruloid proteomic datasets reduce, to some extent, the novelty and conceptual advance of the present study, I am convinced that the mouse dataset presented here is of substantially higher depth and quality than the existing report. Consequently, the human gastruloid dataset represents the primary source of novelty. The high concordance observed between the gastruloid data and recently published proteomic datasets from natural mouse gastrula embryos is encouraging and supports the validity and biological relevance of the model.

However, despite the inclusion of human gastruloid data, it remains unclear how these data advance our understanding of gastrulation or change existing conceptual frameworks. While the authors emphasize that mouse and human gastrulation differ in many respects, the biological implications of these differences are not sufficiently developed. As a result, the human dataset, although potentially impactful, does not yet deliver a clear conceptual advance.

We thank the reviewer for noting that our mouse dataset is of higher depth and quality than the cited report. We agree that the human dataset presents a key source of novelty, but argue that that is not the only aspect of impact, particularly with the experiments included in the revisions. The primary goal of this work was to generate a comprehensive resource of stage- and experimentally-matched datasets of RNA, protein, and phosphorylation states across gastruloid development. As described in the previous rebuttal, we also demonstrate that this comprehensive phenotyping can generate new insights that cannot be derived from single-cell or bulk transcriptomics alone. Critically, we demonstrate that this multi-modal approach yields biological insights inaccessible from transcriptomics alone, and that the human data is not merely additive but is in several instances indispensable.

1. **Species-specific differences in developmental staging:** By profiling both species, we found that proteomes of primed human gastruloids more closely resemble those of early mouse gastruloids, a difference driven by mitochondrial gene expression (**Extended Data Fig. 3h**). This suggests that human and mouse gastruloids may not be directly stage-equivalent at matched timepoints, which has important implications for how cross-species comparisons are interpreted in the broader gastruloid field. This observation would not have been possible without the human dataset. This point was emphasized in the following passage of the revised Discussion, such that further textual changes to incorporate are

not needed:

“Surprisingly, primed RUES2-GLR proteomes were most similar to early mouse gastruloids, driven by mitochondrial protein upregulation suggesting RUES2-GLR cells may already be primed towards gastrulation at the protein level. This highlights potential species-specific differences in staging. While our study is a starting point for cross-species comparisons, more work is needed to understand the extent of cell line-specific and species-specific differences, i.e., through more continuous temporal sampling and computational staging between species¹⁰.”

- 2. Post-transcriptional regulation of oxidative phosphorylation:** We observe systematic downregulation of oxidative phosphorylation proteins across gastruloid development that is not captured by RNA-seq data underscoring the necessity of proteomic profiling to accurately characterize metabolic transitions during gastrulation. Mitochondrial gene sets are among the most transcriptome-proteome discordant across both human and mouse gastruloid development, highlighting evolutionarily conserved post-transcriptional regulation during mammalian gastrulation. We have modified the Discussion to make these points in the following further revised passage:

“The discordance of oxidative phosphorylation genes in both human and mouse gastruloid development suggests that post-transcriptional regulation of metabolic machinery is evolutionarily conserved during early lineage specification and the heightened discordance at the earliest stages points to a developmental window of active proteome remodeling during cell fate transitions.”

- 3. Nomination of a developmentally important kinase from human phosphoproteomics:** Using kinase-substrate enrichment analysis of the human phosphoproteome, we nominated MAPKAPK2 as a regulator of early development, a role that, while established in cancer biology, has not previously been characterized in this context. This nomination was driven by the enrichment of its downstream substrates including ZFP36L1 S92 observed only in the human phosphoproteomics dataset. Functional validation confirmed that perturbation of MAPKAPK2 in developing gastruloids produces striking morphological defects, including multi-axis formations, loss of patterning, and upregulation of SOX2. This mechanistic insight connecting a specific kinase to axis patterning and pluripotency exit during gastrulation represents a direct conceptual advance made possible exclusively by the human phosphoproteomics dataset. We have modified the Discussion to include this point in the following further revised passage:

“Using phosphoproteomics, we identified MAPKAPK2 as a regulator of human gastruloid development, a role not previously characterized outside of cancer and stress response contexts. Our results implicate this kinase in symmetry breaking and pluripotency exit during human gastrulation and highlight how phosphoproteomics data can reveal post-translational regulators of early human development that are invisible to transcriptomic approaches alone.”

2. The value of cross-species analysis lies in the careful interpretation of interspecies differences. In the present study, however, the analytical framework predominantly emphasizes similarities across embryonic models, while observed differences are largely dismissed or implicitly attributed to artifacts of the model systems.

The human gastruloid model used here is derived from the mouse system, with the addition of RA and Matrigel to CHIR treatment to optimize patterning. These methodological choices introduce substantial similarities between the two models, while simultaneously complicating the interpretation of observed discrepancies. If the authors attribute these discrepancies primarily to model-specific features (e.g., RA or Matrigel), an alternative strategy would be to employ a human gastruloid model that more closely resembles the mouse system (e.g., Moris et al., 2020, Nature), thereby reducing confounding methodological variation.

More broadly, the authors should explicitly address the inherent challenges of interpreting interspecies differences using embryoid models. At a minimum, key species-specific findings should be validated in stage-matched natural mouse embryos and followed by a careful assessment of their biological significance.

In response to the first sub-point, we agree with the reviewer that cross species comparisons require careful interpretation because protocols are species-specific. As noted in our rebuttal, we chose the RA-gastruloid model as it produced the most developmentally advanced cell types for this window of human development. We agree that comparing the proteomes and dynamics across conventional human and mouse gastruloids can shed light on the extent of differences arising from protocols as opposed to true evolutionary differences. Furthermore, while we agree that additional independent validation in stage-matched embryos would definitely strengthen our observations, we compare our datasets with existing stage-matched mouse embryo datasets (Stelloo et al., 2024, Cell Stem Cell, PMID: 38754429) and observe concordant trends in both human and mouse gastruloids. The Discussion has been further modified to make these points more clearly:

“Profiling of gastruloids under different conditions (e.g. varying starting cell numbers^{131,132}) and benchmarking against in vivo models may be necessary to identify effects of protocol variation and establish the physiological significance of gastruloid-derived proteomic signatures. Species comparisons would further benefit from harmonization of mouse and human protocols.”

3. Regarding RNA-protein discordance, as mentioned by the authors, numerous previous studies have reported varying degrees of concordance between mRNA and protein levels across diverse biological contexts. Therefore, the observation of RNA-protein discordance in this study is not, in itself, unexpected. The key issue is how such discordance provides new insight into gastrulation that was previously unappreciated. I encourage the authors to expand their discussion on this point.

We agree with the reviewer about previous reports of RNA-protein discordance and have sought to eliminate any claims or implication of novelty with regard to discordance. Instead, we argue that the extent of discordance has not been cataloged in development, let alone early embryonic stages including gastrulation. As stated above, what is unique to our dataset is the ability to identify the processes that tend to discordant across gastruloid development (i.e. mitochondrial genes) and the stages of gastruloid development that exhibit the most discordance (e.g. early stage mouse gastruloids) because of experimentally matched sampling of transcripts and proteins. We have further modified the future directions and opportunities passage of the Discussion to better articulate this point:

“In gastruloids, we observe moderate correlation between transcript and protein abundances with a clear discordance in mitochondrial oxidative phosphorylation genes but not for WNT signaling and steroid biosynthesis. Our findings align with studies mapping RNA-protein relationships in developmental contexts^{16,19–23} and highlight the need to study multiple biomolecular layers—e.g., the transcriptome, proteome, and their interactions—during development. The discordance of oxidative phosphorylation genes in both human and mouse gastruloid development suggests that post-transcriptional regulation of metabolic machinery is evolutionarily conserved during early lineage specification and the heightened discordance at the earliest stages points to a developmental window of active proteome remodeling during cell fate transitions. Applying ribosome profiling¹²³ could disentangle translational control from protein turnover as a driver of these discordances, and matched single cell proteomics and transcriptomics¹²⁴ would enable cell type resolution of these effects.”

Minors:

1. When comparing with the existing mouse proteomic dataset, I recommend that the authors use the most recent study (PMID: 39855199), which reports a substantially larger number of detected proteins than the dataset currently used in this manuscript.

We thank the reviewer for pointing us to this study and are aware that it reports a substantially larger number of detected proteins. We updated **Extended Data Figure 2c** (reproduced below) to incorporate the mouse embryo comparison. We refrained from making comparisons of temporal protein dynamics with this study because it samples stages of development from meiosis II and up to blastocyst. We now cite the referenced study as follows:

“Strong overlap with gastruloid and embryonic proteome datasets^{42–44} support the interpretation that we sampled biologically relevant temporal protein changes (**Extended Data Fig. 2c,d**).”

Extended Data Figure 2 (panel c only). Quantitative proteomics of human gastruloids expands protein coverage and recapitulates temporal trends observed during gastruloid differentiation. (c) Proportion of the 7,352 proteins detected in human gastruloids [this study] also detected in published human and mouse embryo proteomics datasets^{42–44}.

2. In this revision, one major improvement is the authors' analysis of co-regulation patterns among cooperatively associated proteins. However, it remains unclear to what extent this analysis specifically benefits from proteomic measurements. In principle, a similar co-regulation network might be inferred from transcriptomic data alone. I therefore encourage the authors to clarify the added value of using proteomic data for this analysis and to explain whether and how the conclusions would differ from those derived from RNA-based datasets.

We thank the reviewer for raising this important point. As noted in our manuscript and echoed by the reviewer in Major Point #3, numerous studies have documented substantial RNA–protein discordance, demonstrating that transcript abundance alone is often a poor predictor of protein abundance, particularly when inferring functional relationships among co-regulated genes. Indeed, systematic comparisons of mRNA and protein coexpression networks have demonstrated that proteome profiling outperforms transcriptome profiling for coexpression-based gene function prediction (Wang et al., *Mol. Cell Proteomics*. 2017, PMID: 27836980).

To illustrate the added value of proteomic measurements for our co-regulation analysis, we computed pairwise Pearson correlations of either transcript or protein levels across two well-characterized metabolic pathways: glycolysis and the TCA cycle (added as **Extended Data Fig. 6a**, reproduced below). At the protein level, we observe a striking and biologically coherent pattern: genes within each pathway are strongly positively correlated with one another, while genes across the two pathways are anticorrelated. This is consistent with the known metabolic reciprocity between glycolysis and the TCA cycle and suggests that protein-level measurements faithfully capture coordinated pathway regulation. In contrast, the RNA-level correlations do not recapitulate this pattern; intra-pathway and inter-pathway correlations are less clearly delineated, and the anticorrelation between the two pathways is largely absent.

This example underscores that, at least in some cases, constructing a co-regulation network from transcriptomic data alone would yield fundamentally different and noisier connectivity patterns that obscure biologically meaningful relationships. Post-transcriptional regulatory mechanisms, including differential mRNA

stability, translational control, and protein turnover, likely contribute to this discordance and are only captured at the protein level. Thus, our proteomic-based co-regulation analysis provides a more functionally relevant representation of coordinated gene activity than would be achievable from RNA measurements alone.

We have modified the Results section to cite the aforementioned study and to show the matrix of correlations that illustrates this example (as a new panel in **Extended Data Figure 6**) as follows:

“Proteome-based coexpression has been shown to outperform transcriptome-based coexpression for predicting gene function⁵⁷. Consistent with this, pairwise correlations of glycolysis and TCA cycle genes in our data revealed coherent intra-pathway correlations and inter-pathway anticorrelations at the protein level that were not recapitulated at the RNA level (**Extended Data Fig. 6a**).”

Extended Data Figure 6 (panel a only). Protein-level co-regulation reveals distinct pathway-specific correlation not observed at the RNA level. Heatmap displaying pairwise Pearson correlation coefficients among glycolysis (purple) and TCA cycle (green) genes. The upper triangle represents protein-level correlations and the lower triangle represents RNA-level correlations. Black cells denote self-comparisons along the diagonal.

3. To validate the reported downregulation of mitochondrial activity, the authors assess ATP5F1A protein levels by immunofluorescence. However, it is unclear how the fluorescence signal was normalized, in particular which condition (primed or early) was used as the baseline for comparison.

We thank the reviewer for flagging this. We normalized ATP5F1A pixel intensities to that of DAPI and define this as normalized fluorescence. We plot this ratio across H9 and RUES2 cells and gastruloids and have clarified this now in the Methods section as follows:

“When comparing pixel intensities across images we normalized fluorescence intensities (e.g. antibody) to that of DAPI (defined as normalized fluorescence).”